# ARTICLES

# Differences in CD80 and CD86 transendocytosis reveal CD86 as a key target for CTLA-4 immune regulation

Alan Kennedy [1,13], Erin Waters[1,13], Behzad Rowshanravan[1,13], Claudia Hinze [1], Cayman Williams[1], Daniel Janman[1], Thomas A. Fox [1], Claire Booth[2], Anne M. Pesenacker [1], Neil Halliday [1], Blagoje Soskic [1], Satdip Kaur[3], Omar S. Qureshi[4], Emma C. Morris [1], Shinji Ikemizu[5], Christopher Paluch [6], Jiandong Huo[7,8,9,10], Simon J. Davis [6,11], Emmanuel Boucrot[12], Lucy S. K. Walker [1] and David M. Sansom [1 ✉]

**CD28 and CTLA-4 (CD152) play essential roles in regulating T cell immunity, balancing the activation and inhibition of T cell responses, respectively. Although both receptors share the same ligands, CD80 and CD86, the specific requirement for two distinct ligands remains obscure. In the present study, we demonstrate that, although CTLA-4 targets both CD80 and CD86 for destruction via transendocytosis, this process results in separate fates for CTLA-4 itself. In the presence of CD80, CTLA-4 remained ligand bound, and was ubiquitylated and trafficked via late endosomes and lysosomes. In contrast, in the presence of CD86, CTLA-4 detached in a pH-dependent manner and recycled back to the cell surface to permit further transendocytosis. Furthermore, we identified clinically relevant mutations that cause autoimmune disease, which selectively disrupted CD86 transendocytosis, by affecting either CTLA-4 recycling or CD86 binding. These observations provide a rationale for two distinct ligands and show that defects in CTLA-4-mediated transendocytosis of CD86 are associated with autoimmunity.**

The CD28–CTLA-4 system is a critical regulator of the immune response. CD28 is required to stimulate T cell responses, enhance T cell differentiation, support cytokine production and regulate metabolism[1]. In the absence of CD28 engagement, T cell responses are suboptimal in naive, memory and regulatory T (T_reg) cell compartments[2–5]. CD28 defects also indirectly affect the B cell compartment, resulting in a lack of high-affinity, class-switched antibodies due to poor development of follicular helper T cells and consequent failure of T cell help[6]. Accordingly, CD28 stimulation is critical to adaptive immunity and blockade of CD28 co-stimulation is used therapeutically as an immune suppressive modality[7].

In direct contrast to CD28, CTLA-4 functions as a negative regulator of T cell responses. CTLA-4-deficient mice die early in life, exhibiting defective T_reg cell function[8–10] and increased T cell responses to self-antigens[11]. Humans with heterozygous gene defects in *CTLA4* have been described and exhibit a range of autoimmune features consistent with T_reg cell defects[12,13]. In addition, the heightened anti-tumor responses seen after CTLA-4 blockade are often accompanied by substantial autoimmune side effects[14]. Intriguingly, the opposing functions of CD28 and CTLA-4 are connected by the fact that they share the same two ligands (CD80 and CD86)[15], with immune dysregulation resulting from defective CTLA-4 seemingly due to excessive, ligand-driven stimulation of CD28.

The distinct roles of CD80 and CD86 remain elusive, with expression of the ligands largely confined to immune cells, such as dendritic cells (DCs) and B cells[16], consistent with both ligands providing CD28 co-stimulation. However, expression levels are highly variable depending on cell type, activation state and cytokine environment. Although CD80 and CD86 can be co-expressed, there is clear evidence of differential expression on monocytes, memory B cells, DCs, T cells and T_reg cells[16–19].

Although considered as immunologically similar, CD80 and CD86 have diverged considerably and possess surprisingly low sequence homology (~25%). The two ligands have distinct biophysical characteristics, with CD80 having a higher monomeric affinity for both CD28 and CTLA-4 (CD80–CD28 ~4 μM and CD80–CTLA-4 ~0.2 μM)[20]. In contrast, CD86 has much weaker interactions with both receptors (CD86–CD28 ~20 μM and CD86–CTLA-4 ~2 μM)[20,21]. Further differences in the dimerization state of ligands and receptors results in additional avidity effects, with CD80 and CTLA-4 forming an unusually high-avidity, dimer–dimer interaction, compared with the lower-affinity monomeric interaction of CD86 with CTLA-4 (refs. [20,22]). Recent data have further reinforced the concept of differential ligand functions, revealing that CD80 physically interacts with programmed cell death protein 1 (PD-L1) in *cis*, thereby preventing binding to PD-1 (ref. [23]). Despite these

[1]UCL Institute of Immunity and Transplantation, London, UK. [2]Molecular and Cellular Immunology Section, UCL Great Ormond Street Institute of Child Health, London, UK. [3]School of Immunity and Infection, Institute of Biomedical Research, University of Birmingham Medical School, Birmingham, UK. [4]Celentyx Ltd, Birmingham, UK. [5]Division of Structural Biology, Graduate School of Pharmaceutical Sciences, Kumamoto University, Kumamoto, Japan. [6]Medical Research Council Human Immunology Unit, John Radcliffe Hospital, University of Oxford, Oxford, UK. [7]Structural Biology, The Rosalind Franklin Institute, Didcot, UK. [8]Division of Structural Biology, University of Oxford, Oxford, UK. [9]Wellcome Trust Centre for Human Genetics, Oxford, UK. [10]Protein Production UK, The Rosalind Franklin Institute—Diamond Light Source, The Research Complex at Harwell, Didcot, UK. [11]Radcliffe Department of Medicine, John Radcliffe Hospital, University of Oxford, Oxford, UK. [12]Institute of Structural and Molecular Biology, University College London, London, UK. [13]These authors contributed equally: Alan Kennedy, Erin Waters, Behzad Rowshanravan. ✉e-mail: d.sansom@ucl.ac.uk

obvious differences, the impact of two ligands within the CD28–CTLA-4 system remains unclear.

Understanding CTLA-4 interactions with its ligands has been hindered by long-standing issues over the precise role of CTLA-4 itself. Several cell-intrinsic and cell-extrinsic inhibitory models have been proposed, including inhibitory signaling, $T_{reg}$ cell function, adhesion effects, inhibitory cytokine production and ligand competition[10,24–28]. Previously, we identified that CTLA-4 possesses an unusual property known as transendocytosis (TE)[29]. In this process, CTLA-4 physically captures its ligands from opposing cells and targets their destruction via T cell-mediated endocytosis and lysosomal degradation. We have found that patients with mutations in *CTLA4* have defects in TE[12] and that TE occurs in vivo between CTLA-4+ $T_{reg}$ cells and migratory DCs[30]. CTLA-4-mediated TE therefore exploits the rapid endocytic and recycling behavior of CTLA-4 (ref. [31]) to limit ligand availability for CD28 engagement[32], thereby providing an explanation for why CD28 and CTLA-4 share ligands despite their opposing functions.

Although TE is a plausible mechanism for CTLA-4 function, it does not explain why two distinct ligands exist, because such an inhibitory mechanism could operate with only a single shared ligand. In the present study, we directly compared the two ligands and their interactions with CTLA-4 during TE. We reveal that CD80 remained bound to CTLA-4 after TE, accompanied by the ubiquitylation of CTLA-4 and its trafficking toward late endosomes and lysosomes. In contrast, CD86 readily dissociated after internalization, leaving CTLA-4 unmodified and permitting its recycling for further ligand capture. Furthermore, we observed that disease-related defects in CTLA-4 recycling due to deficiency in a key regulator of protein trafficking (lipopolysaccharide-responsive beige-like anchor (LRBA) protein)[33,34] affected TE of CD86 more than CD80. We also identified disease-associated missense mutations in CTLA-4 that specifically compromised TE of CD86. Taken together, these data support a model whereby CD86 is effectively controlled by CTLA-4, despite its weak affinity, due to efficiency gains in TE resulting from CTLA-4 recycling. In contrast, CD80 functionally disables CTLA-4 by remaining bound, causing ubiquitylation of CTLA-4 and inhibiting its recycling.

## Results

**TE of CD80 and CD86 show distinct features.** To study CD80 and CD86 TE, we carried out experiments using a variety of CTLA-4-expressing cells (Chinese Hamster Ovary (CHO) cells, Jurkat T cells and primary human $T_{reg}$ cells) to capture fluorescent ligands (either green fluorescent protein (GFP) or mCherry tagged) from opposing cells during CTLA-4–ligand interactions. Consistent with previous reports[29,35], CTLA-4-expressing cells robustly captured both

ligands by TE resulting in their transfer to the recipient CTLA-4+ cell. To ensure that we measured ligand transfer only due to TE and not to cell doublets, we labeled ligand donor cells with CellTrace Violet (CTV) and monitored fluorescent ligand transfer into CTLA-4+ cells by flow cytometry (Fig. 1a–c). Loss of ligand was evident from the CTV+ donor cells (Fig. 1b,c, upper quadrants), indicating that both CD80 and CD86 were effectively removed by TE. Removal of ligand from donor cells and uptake by CTLA-4 recipient cells was time dependent, indicating that this was not a trogocytosis event, but rather sustained removal over time (Extended Data Fig. 1a,b). Ligand acquisition by CTLA-4+ cells was evident as fluorescent protein uptake in the lower right quadrants, which was readily visible with CTLA-4+ cells acquiring CD80 (Fig. 1b, lower right gray quadrant). In contrast, CD86 uptake was consistently more difficult to detect (Fig. 1b, lower right blue quadrant). We therefore utilized a lysosomal acidification inhibitor, $NH_4Cl$, to examine whether CD86 detection was sensitive to pH because this is important for endolysosomal fusion and acidification of lysosomes. In the presence of $NH_4Cl$, CD86 capture was now more readily observed (Fig. 1b, right column, lower right quadrants). These data raised the possibility that, after TE, CD80 and CD86 were subject to different intracellular processing and that CD86 detection within recipient cells was more sensitive to pH. Similar observations were made in CTLA-4+ Jurkat T cells (Extended Data Fig. 1c) as well as using unmanipulated CTLA-4+ human $T_{reg}$ cells, where we used bafilomycin A (BafA) rather than $NH_4Cl$ to neutralize the pH (Fig. 1c).

To verify that transferred ligands were actually inside CTLA-4+ recipient cells, we used confocal microscopy, which revealed further significant differences in ligand behavior after TE. By exploiting the distinct cytoplasm of CHO cells, we observed that CD80 accumulated in large vesicles together with CTLA-4, whereas CD86 was typically found in smaller vesicles and was significantly less co-localized with CTLA-4 (Fig. 1d). Similar results were obtained using unmanipulated CTLA-4+Foxp3+ human $T_{reg}$ cells, which again showed a greater degree of co-localization between CTLA-4 and CD80 than CTLA-4 and CD86 (Fig. 1e). Taken together, we observed distinct characteristics for each ligand in terms of uptake, vesicle size and degree of co-localization with CTLA-4, consistent with the idea that, after TE, CD86 separated from CTLA-4 in a pH-dependent manner, whereas CD80 remained strongly associated with CTLA-4.

**CD80 engagement induces CTLA-4 ubiquitylation.** To study these ligand-dependent differences in further detail, we examined the fate of CTLA-4 in CHO cells after TE using immunoblot analysis. Engagement by CD80 resulted in an increased molecular mass of the CTLA-4 band (seen as upward smearing (Fig. 2a, boxed)),

**Fig. 1 | TE of CD80 and CD86 reveals distinct ligand characteristics. a**, Cartoon representing TE assay. Ligand donor cells expressing CD80 or CD86 proteins with mCherry or GFP fusion tags (red plasma membrane) are labeled with CTV (CTV+) and mixed with CTLA-4-expressing (blue dots) recipient cells (CTV−). During TE, plasma membrane-expressed ligands are removed from donor cells (reduced red plasma membrane signal) and the fluorescent ligand is now detected in CTLA-4-expressing recipient cells. Internalized ligands either separate from CTLA-4 (red dots) or remain co-localized (blue dots with red outline). **b**, Flow cytometric analysis of the TE assay described in **a** for CD80 and CD86 (16 h) showing loss of ligand–GFP from CTV+ donor CHO cells (cells move from the top right to the top left quadrant with ligand loss) and ligand gain by CTLA-4-expressing recipient (CTV−) CHO cells (cells move from the bottom left quadrant to the bottom right). $NH_4Cl$ was added to inhibit lysosomal activity. Detection of CD80 and CD86 acquisition is highlighted in the gray- and blue-shaded quadrants, respectively. **c**, TE by human $T_{reg}$ cells showing CD80–mCherry or CD86–mCherry ligands captured from DG-75 B cells. BafA was added to inhibit lysosomal activity. Detection of CD80 and CD86 acquisition is highlighted in the gray- and blue-shaded quadrants, respectively. **d**, Confocal analysis of overnight TE in CHO cells showing CTLA-4 (green), co-localization of CTLA-4 and ligand (yellow) and CD80 or CD86 (red). Scale bar, 10 μm. Graphs show the percentage co-localization between CTLA-4 and ligand and the average size of co-localized vesicles. Statistical significance was determined by a Mann–Whitney, two-tailed, unpaired test: *$P < 0.05$, ***$P < 0.001$. All data are presented as mean ± s.d. and show individual data points ($n = 37$–40 cells from one experiment, representing three independent experiments). **e**, Confocal analysis of 6-h TE in $T_{reg}$ cells showing CTLA-4 alone (green) or co-localization (yellow) with CD80 or CD86 ligand (red) acquired from DG-75 B cells (gray). Scale bar, 5 μm. Graphs show the percentage co-localization between CTLA-4 and ligand and the average size of co-localized vesicles. Statistical significance was determined by a Mann–Whitney, two-tailed, unpaired test: ***$P < 0.001$, ****$P < 0.0001$. All data are presented as mean ± s.d. and show individual data points ($n = 33$ fields of view examined over three independent experiments). All confocal analysis was performed in CellProfiler.

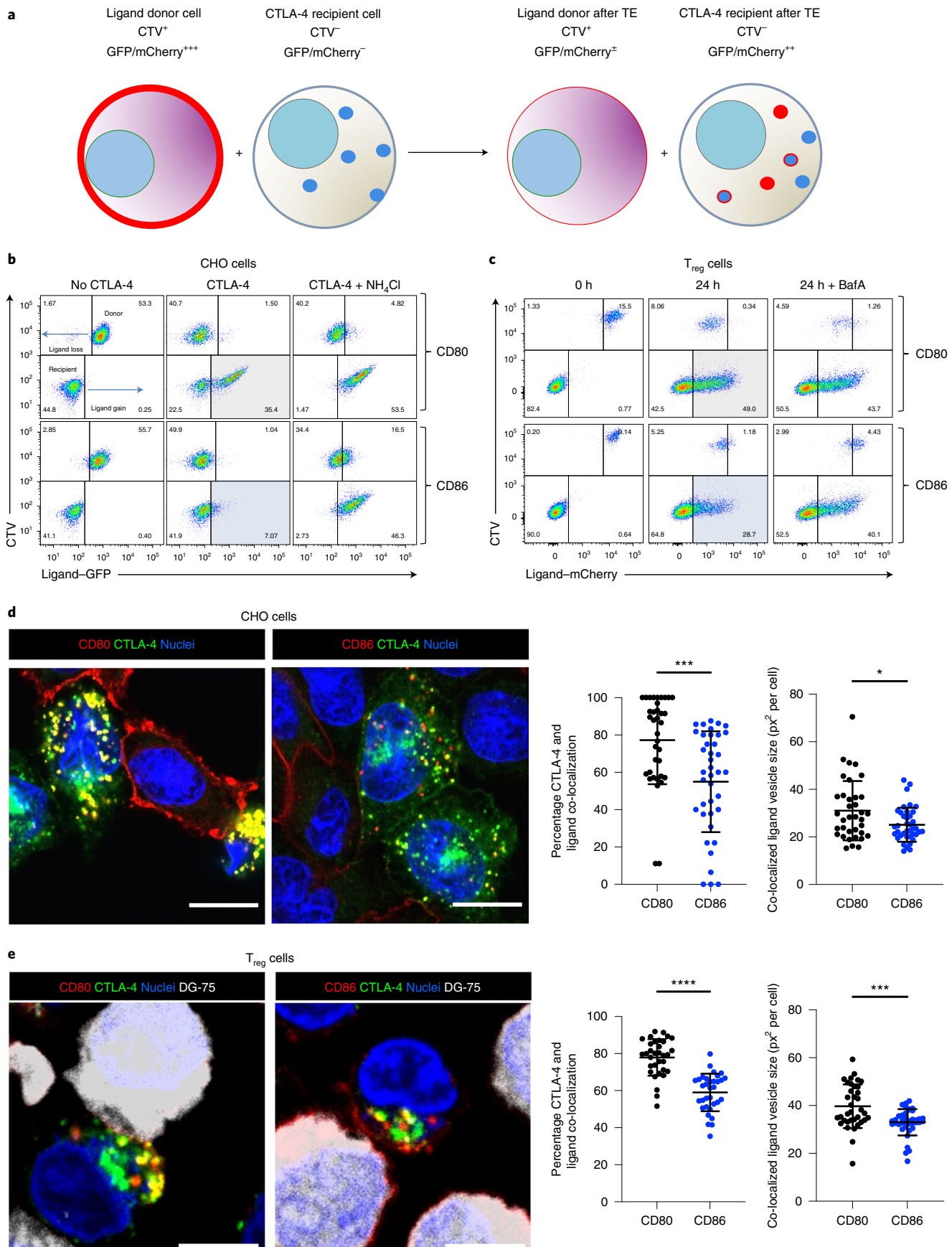

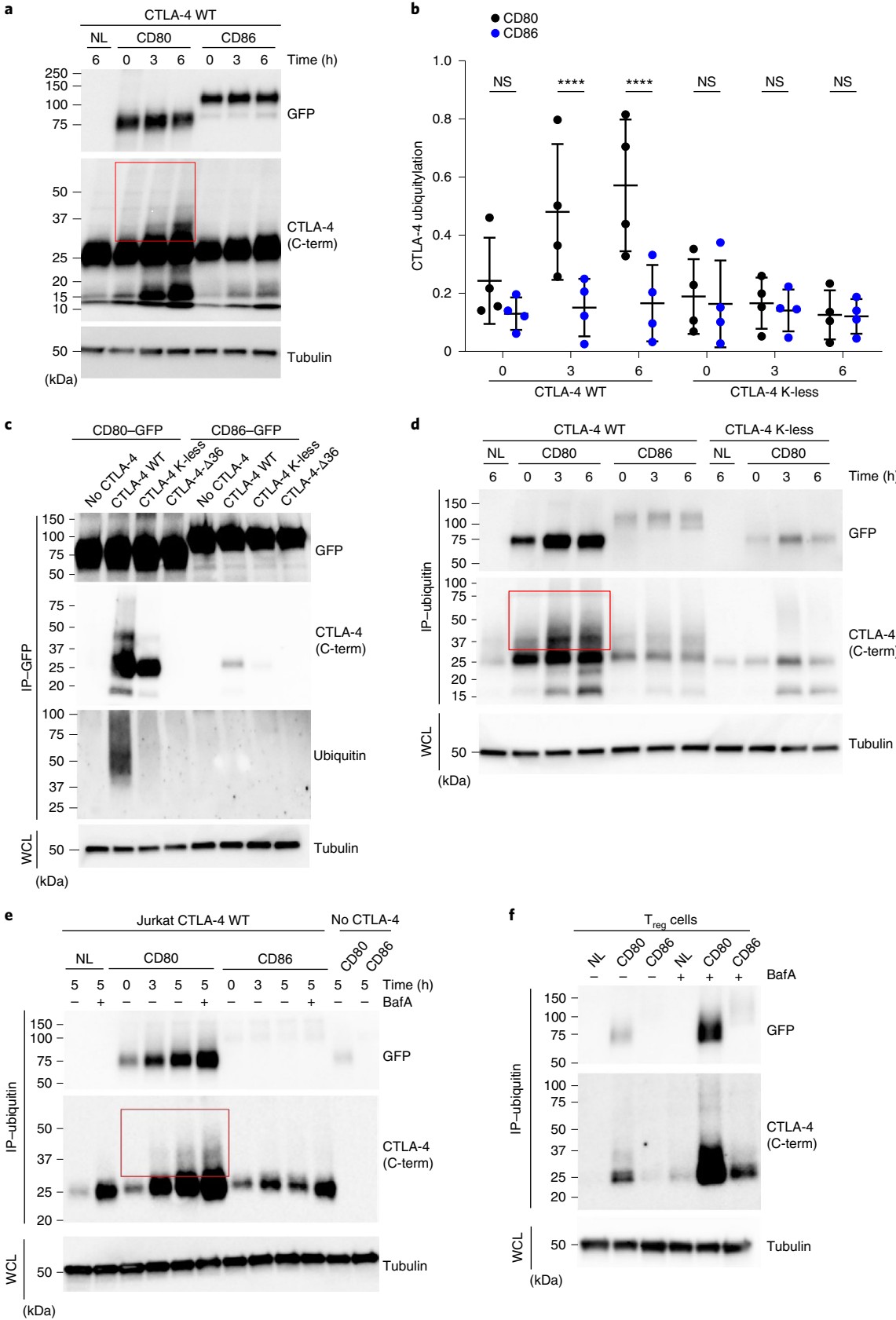

indicative of post-translational modification, whereas this was not observed after engagement with CD86 (Fig. 2a,b). The inclusion of the proteasome inhibitor MG132, which depletes free ubiquitin, prevented this change in protein mobility, supporting the possibility

that CTLA-4 was ubiquitylated (Extended Data Fig. 2a). In addition, a mutant CTLA-4 protein lacking lysine residues in the cytoplasmic domain (CTLA-4 K-less) did not show similar increases in molecular mass ($M_r$) on CD80 TE (Fig. 2b and Extended Data

**Fig. 2 | CD80 engagement drives CTLA-4 ubiquitylation. a**, Immunoblot analysis of TE experiments using CD80–GFP- and CD86–GFP-expressing CHO cells compared with cells with no ligand (NL). After TE for the times shown, whole-cell lysates (WCL) from combined CTLA-4 and ligand-expressing cells were directly blotted using anti-CTLA-4 (C-term) and anti-GFP (ligand) antibodies. Lysates were also blotted for tubulin as a sample-processing control. **b**, Quantification of anti-CTLA-4 (C-term) blots showing >25-kDa smear density relative to the 25-kDa band after CD80 or CD86 TE. The graph compares CTLA-4 WT or CTLA-4 lacking cytoplasmic lysine residues (K-less). Analysis was performed in Image Lab (BioRad) from four independent experiments. Statistical significance was calculated using two-way ANOVA with Sidak's multiple comparison correction. All data are presented as mean ± s.d. and show individual data points. $^{****}P < 0.0001$. **c**, CHO TE experiments performed for 4 h and IP carried out via CD80–GFP or CD86–GFP. Precipitates from WT CTLA-4, K-less and CTLA-4 lacking a cytoplasmic domain (Δ36) were blotted for ubiquitin, CTLA-4 (C-term) and ligand (GFP). **d**, TE using CHO–CD80/86-CTLA-4 (WT/K-less) was carried out for the times indicated, subjected to ubiquitin IP, followed by blotting for CTLA-4 (C-term) and CD80/86 (GFP). **e**, CD80 and CD86 TE experiments performed with CTLA-4-expressing Jurkat T cells and DG-75 B cells expressing CD80–GFP or CD86–GFP at a 1:1 ratio. At the times indicated, ubiquitin was precipitated and then blotted for CTLA-4 and ligand–GFP. **f**, CD80 and CD86 TE experiments carried out using DG-75–GFP ligand-expressing cells and unmodified primary human $T_{reg}$ cells for 5 h followed by ubiquitin precipitation and blotting for CTLA-4 and ligand. All data represent a minimum of three similar experiments. The increased $M_r$ of CTLA-4 is highlighted by a red box. For IPs, WCLs from all experiments were blotted for tubulin to control for protein loading.

Fig. 2b), consistent with the requirement for lysine residues in protein ubiquitylation.

To confirm that CD80 TE induced ubiquitylation of CTLA-4, we performed immunoprecipitation (IP) experiments pulling down CTLA-4 via bound CD80–GFP or CD86–GFP and immunoblotting for ubiquitin. Wild-type (WT) CTLA-4 was ubiquitylated in the presence of CD80, but not the CTLA-4 K-less mutant, or another CTLA-4 mutant that lacked the cytoplasmic domain (Δ36) (Fig. 2c). Although CD86–GFP was readily precipitated in these experiments, very little CTLA-4 was co-precipitated in keeping with this lower avidity interaction. To overcome this issue, we immunoprecipitated all ubiquitin-associated proteins after CD80 and CD86 TE and then immunoblotted for CTLA-4. This revealed a CD80-dependent increase in $M_r$ of CTLA-4 WT over time (red box), which was not observed with CD86 or seen after CD80 interaction with CTLA-4 K-less (Fig. 2d). Experiments using CTLA-4-expressing Jurkat T cells confirmed the difference between CD80 and CD86 in inducing ubiquitylation of CTLA-4 (Fig. 2e) and that CTLA-4 K-less was not ubiquitylated (Extended Data Fig. 2c). Precipitation via ubiquitin also revealed the co-precipitation of CD80–GFP but very little CD86–GFP (Fig. 2e), supporting the notion that internalized CD80 remained bound to CTLA-4, but that CD86 did not. We finally verified these observations using unmanipulated primary human $T_{reg}$ cells expressing endogenous CTLA-4, with similar results (Fig. 2f). Ubiquitin IP also pulled down native $M_r$ CTLA-4 (~25 kDa), suggesting that nonubiquitylated CTLA-4 might be associated with a ubiquitylated partner, even in the unligated state. Nonetheless, CD80 ligation stimulated an increase in this association and in the

ubiquitylation of CTLA-4 itself, resulting in precipitation of higher $M_r$ CTLA-4 (Fig. 2d–f). In all systems, we observed a background of CTLA-4 ubiquitylation, even in the absence of ligand, revealed after treatment with BafA (Fig. 2e,f). We also observed a low $M_r$ degradation fragment of CTLA-4, at ~15 kDa. Although increases in this fragment occurred after CD80 engagement (Fig. 2a,d), it was also seen to some extent in the absence of ligand and with nonubiquitylated (K-less) CTLA-4 molecules, indicating that it was derived from both ubiquitin-dependent and ubiquitin-independent pathways. Overall, we concluded that CTLA-4 undergoes a constitutive level of degradation, but that ubiquitylation of CTLA-4 was specifically stimulated by ligation with CD80 but not CD86.

**CD80 and CD86 differentially control CTLA-4 traffic.** To visualize the association between CTLA-4 and ubiquitin in a cellular context, we performed a proximity ligation assay (PLA). Accordingly, when antibodies to ubiquitin and CTLA-4 are within <40-nm proximity, a PLA signal is generated. We compared the level of CTLA-4 ubiquitylation resulting from CD80 and CD86 TE within the extensive cytoplasm of CHO cells (Fig. 3a). Consistent with biochemical data, although background ubiquitylated CTLA-4 was evident in the presence of CD86 and the absence of ligand, CD80 significantly increased the number of PLA puncta, representing an increase in the association between ubiquitin and CTLA-4 (Fig. 3a). We also carried out additional analysis of the intracellular locations in which CTLA-4 and ligand were found. This revealed that CD80–CTLA-4 interactions were significantly more likely to be observed in low pH compartments such as late endosomes (Rab7+) and lysosomes

**Fig. 3 | CD80 and CD86 direct CTLA-4 trafficking via different intracellular compartments. a**, PLAs in CHO cells showing the association between CTLA-4 and ubiquitin (red) and their co-localization (yellow) with CD80 or CD86 (green) after overnight TE with CTLA-4+ CHO cells labeled with CTFR and CD80/CD86–GFP CHO cells or CHO cells expressing NL at a 1:1 ratio in the presence of $NH_4Cl$. Images were acquired by confocal microscopy and cells quantified for the number of PLA puncta (CTLA-4+Ubq+). The significance was calculated using the Mann–Whitney $U$-test: $^{**}P < 0.01$ All data are presented as mean ± s.d. and show individual data points ($n = 30$ cells from one experiment representing three independent experiments). **b**, Co-localization of CTLA-4, ligand–GFP and markers of intracellular compartments in CTLA-4-expressing HeLa cells (human cells with appropriate morphology) and quantified using CellProfiler. The statistical significance was determined by two-way ANOVA with Sidak's multiple comparison correction: $^{*}P < 0.05$, $^{**}P < 0.01$. All data are presented as mean ± s.d. and show individual data points ($n = 8$ fields of view examined over two independent experiments). **c**, CHO cells expressing CD80 or CD86 surface stained using CTLA-4–Ig and then washed at the pH indicated. CTLA-4–Ig remaining bound was detected by Immunoblot using anti-IgG. Lysates were immunoblotted for tubulin as a sample-processing control. Data represent four individual experiments. **d**, Impact of $NH_4Cl$ on CD80 and CD86 TE over time. CHO cells expressing GFP–ligands were labeled with CTV and co-incubated with CTLA-4-expressing CHO cells for 8 h (**d**) or the times shown (**e**) in the presence of 25 mM $NH_4Cl$, and analyzed by flow cytometry. Detection of CD80 and CD86 acquisition is highlighted in the gray- and blue-shaded quadrants, respectively. All data are presented as mean ± s.d. ($n = 4$ independent experiments). **f**, Detection of available CTLA-4 in Jurkat T cells after overnight TE of DG-75 B cells expressing CD80, CD86 or without ligand ('no TE'). Histograms show available CTLA-4 measured using anti-CTLA-4 antibody at 37 °C for 60 min and MFI of CTLA-4 staining quantified. The statistical significance was determined by two-way ANOVA with Sidak's multiple comparison correction: $^{****}P < 0.0001$. All data are presented as mean ± s.d. and show individual data points. **g**, The TE experiment in **f** was repeated using CTLA-4+CD4+CD25+ human $T_{reg}$ cells. The statistical significance was determined by two-way ANOVA with Sidak's multiple comparison correction: $^{****}P < 0.0001$. All data are presented as mean ± s.d. and show individual data points from four independent experiments.

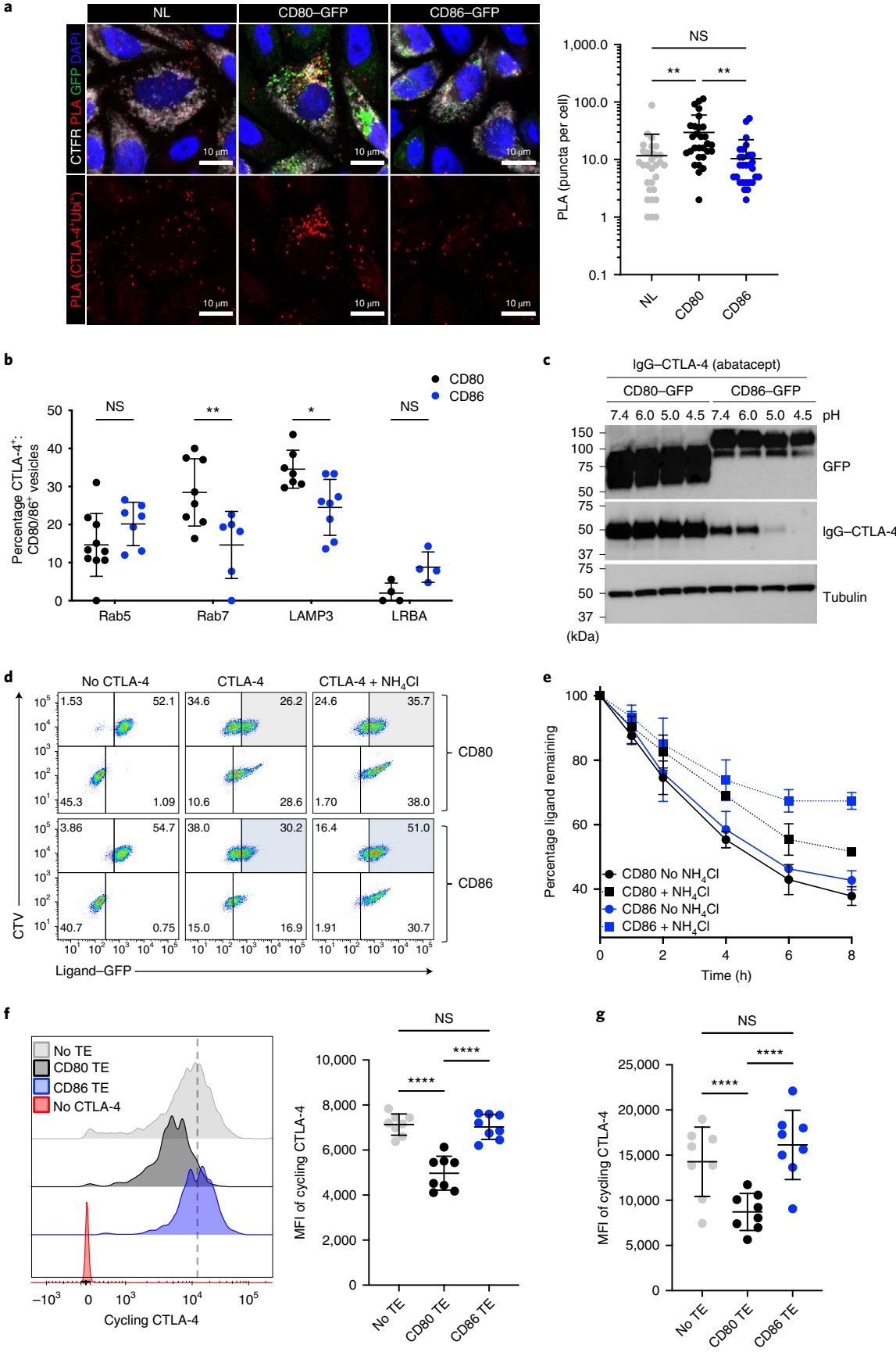

(LAMP3[+]) than CD86–CTLA-4 interactions (Fig. 3b and Extended Data Fig. 3). There was also a suggestion (albeit not statistically significant) that there was more CD86–CTLA-4 co-localization than CD80–CTLA-4 co-localization in LRBA[+] compartments, which are thought to be important compartments for CTLA-4 recycling[34,36,37].

As pH decreases because proteins traffic toward later endosomal compartments, we addressed how pH affected the interaction between CTLA-4 and its ligands. CD80- or CD86-expressing cells were bound by soluble CTLA-4 (CTLA-4-Ig) and then washed at decreasing pH. This revealed that CTLA-4 remained bound to CD80 even at low pH, whereas it readily dissociated from CD86 under acidic conditions, pH < 6 (Fig. 3c). We further tested the impact of pH-driven dissociation of CD86 on CTLA-4 function. TE of CD80 or CD86 was carried out in the presence or absence of 25 mM NH$_4$Cl to neutralize intracellular pH. We hypothesized that, if low pH affected CD86–CTLA-4 dissociation, then neutralization should impact the ability of CTLA-4 to remove CD86 from donor cells. We observed that the addition of NH$_4$Cl had a substantially greater impact on the ability of TE to downregulate CD86 compared with CD80, with the percentage of donor cells that remained CD86[+] going from ~30% to 51% in the presence of NH$_4$Cl (Fig. 3d). The inhibitory effect on CD86 TE also became more pronounced over time, with removal of CD86 reaching a plateau by 6 h, whereas there was continued downregulation of CD80 even in the presence of NH$_4$Cl (Fig. 3e). Accordingly, these data were consistent with pH-driven dissociation being a feature of efficient CD86 TE.

Next, we performed experiments in Jurkat cells to detect available CTLA-4 after TE. We hypothesized that TE of CD80 should reduce subsequent CTLA-4 detection by remaining bound and driving traffic toward late endosomes. In contrast, if CD86 dissociated, this should allow unoccupied CTLA-4 to recycle to the cell surface after TE. We tested this concept by performing TE assays with either CD80 or CD86 overnight and then staining with anti-CTLA-4. This revealed that detection of CTLA-4 was significantly impaired after TE of CD80 (Fig. 3f). In contrast, CD86 TE did not affect subsequent detection of CTLA-4, consistent with the recycling of empty CTLA-4. These experiments were also performed using human T$_{reg}$ cells with similar results (Fig. 3g).

Overall, these data supported an emerging model whereby, after TE, CTLA-4 remained bound to CD80 and was ubiquitylated, whereas, in the presence of CD86, CTLA-4 remained unmodified and dissociated from CD86 in a pH-dependent manner. Moreover, CD80–CTLA-4 complexes were rarely observed in LRBA[+] recycling compartments, raising the possibility that CD80-mediated ubiquitylation diverted CTLA-4 away from the recycling pathway. Together these data revealed clear differences in the ability of CD80 and CD86 to regulate the pool of functionally available CTLA-4 at the cell surface after TE.

**CTLA-4 recycling affects TE of CD86.** To directly address the importance of CTLA-4 recycling during TE we tested the impact of manipulating the Rab11 GTPase, which is known to affect protein recycling in general and has been implicated in CTLA-4

trafficking[34,37]. We therefore transfected CTLA-4[+] Hela cells with dominant-negative (DN) Rab11 constructs to inhibit recycling, and then measured the impact on CD80 and CD86 TE. We observed that DN Rab11 significantly inhibited CD86 (but not CD80) TE, consistent with recycling activity being of greater importance for efficient CD86 TE (Extended Data Fig. 4a).

To further explore the impact of recycling on ligand uptake, we carried out TE assays using Jurkat cells expressing the CTLA-4 K-less mutant or using LRBA-deficient cells that expressed WT CTLA-4. The clustered regularly interspaced short palindromic repeats (CRISPR)–Cas9 knockout of LRBA increased CTLA-4 degradation (Extended Data Fig. 4b), consistent with defective recycling and in keeping with previous studies[34,37,38]. Conversely, the K-less CTLA-4 mutant, which does not undergo ubiquitylation, showed reduced degradation and enhanced recycling capabilities (Extended Data Figs. 4b and 5a,b). We therefore tested the impact of both the K-less mutation and LRBA knockout (KO) on CTLA-4 TE. Analysis of LRBA-deficient cells revealed that, although TE of CD80 was largely intact, acquisition of CD86 was strongly impaired (Fig. 4a,b, shaded quadrants). Conversely, K-less CTLA-4 exhibited better acquisition of CD86, increasing ligand capture relative to CD80 (Fig. 4b). Kinetic analysis of ligand uptake revealed that the improved ability of K-less to capture CD86 emerged over time, as would be expected from an improved recycling phenotype, whereas LRBA-deficient cells lacked this capability (Fig. 4c). Finally, to verify the K-less phenotype in primary human CD4 T cells, we adopted a CRISPR–Cas9/AAV6 knock-in strategy. *CTLA4* gene expression was replaced with either WT or K-less CTLA-4 complementary DNA expressed under the endogenous CTLA-4 promoter. After gene editing, edited T cells (GFP[+]) were compared for uptake of CD80 or CD86 by TE (Fig. 4d). This revealed that both WT and K-less-expressing T cells retained the same capacity to perform CD80 TE, as shown by mCherry ligand uptake, despite the increased recycling capacity of the K-less CTLA-4 mutant (Extended Data Fig. 5c). In contrast, CTLA-4 K-less conferred a clear advantage for TE of CD86 (Fig. 4d), again highlighting the importance of CTLA-4 recycling to CD86 uptake.

Taken together these data suggested that recycling of CTLA-4 was critical to maintain the efficient uptake of CD86 and that perturbations in recycling caused by three independent manipulations (DN Rab11, LRBA and CTLA-4 lysine mutations) all selectively influenced CD86 TE, while having limited impact on CD80 TE. Given that LRBA mutations are known to be pathogenic, this suggested that impaired control of CD86 might be important in disease development.

**Loss of CD86 TE is associated with autoimmunity.** Given that a growing number of CTLA-4 mutations have been found in patients with autoimmunity[39], we screened these for mutations affecting ligand binding. Based on the crystal structure of CTLA-4 (Fig. 5a), we identified mutations seen in patients (highlighted in red) that were ligand facing and likely to influence ligand binding. We initially expressed these CTLA-4 mutants in CHO cells and tested them for their ability to bind to soluble ligands (CD80–Ig and

**Fig. 4 | Altering CTLA-4 recycling affects TE of CD86 but not CD80. a**, TE assays carried out for 16 h using Jurkat T cells at a ratio of three donors (DG-75 mCherry ligand):one recipient to provide excess ligand. Recipient Jurkat T cells either lacked CTLA-4 ('no CTLA-4') or expressed CTLA-4 WT or CTLA-4 K-less variants. CTLA-4 WT was also expressed in Jurkat cells lacking LRBA after CRISPR–Cas9 targeting (LRBA KD). Representative FACS plots are shown in **a** and quantified in **b** for the amount of ligand acquired by the CTLA-4-expressing cell (shaded quadrant: gray CD80 and blue CD86). The far right-hand graph shows the uptake of CD86 relative to CD80 for each condition. The statistical significance was determined by the paired Student's *t*-test: **P < 0.01. All data are presented as mean ± s.d. and show individual data points from three independent experiments. **c**, Kinetic analysis of the TE assay used in **b**. All data are presented as mean ± s.d. from four independent experiments. **d**, The impact of the K-less mutation on TE by human T cells. K-less or WT CTLA-4 was knocked into the endogenous CTLA-4 locus by homology-directed repair (HDR) using a CRISPR–Cas9/AAV6 system. Knock-in cells were detected using a GFP reporter. GFP[+] T cells expressing WT or K-less cDNA, were analyzed for their ability to capture CD80–mCherry (gray quadrants) or CD86–mCherry (blue quadrants) from DG-75 B cells. The statistical significance was calculated using two-way ANOVA with Sidak's multiple comparison correction: **P < 0.01. All data are presented as mean ± s.d. and show individual data points from three biologically independent samples.

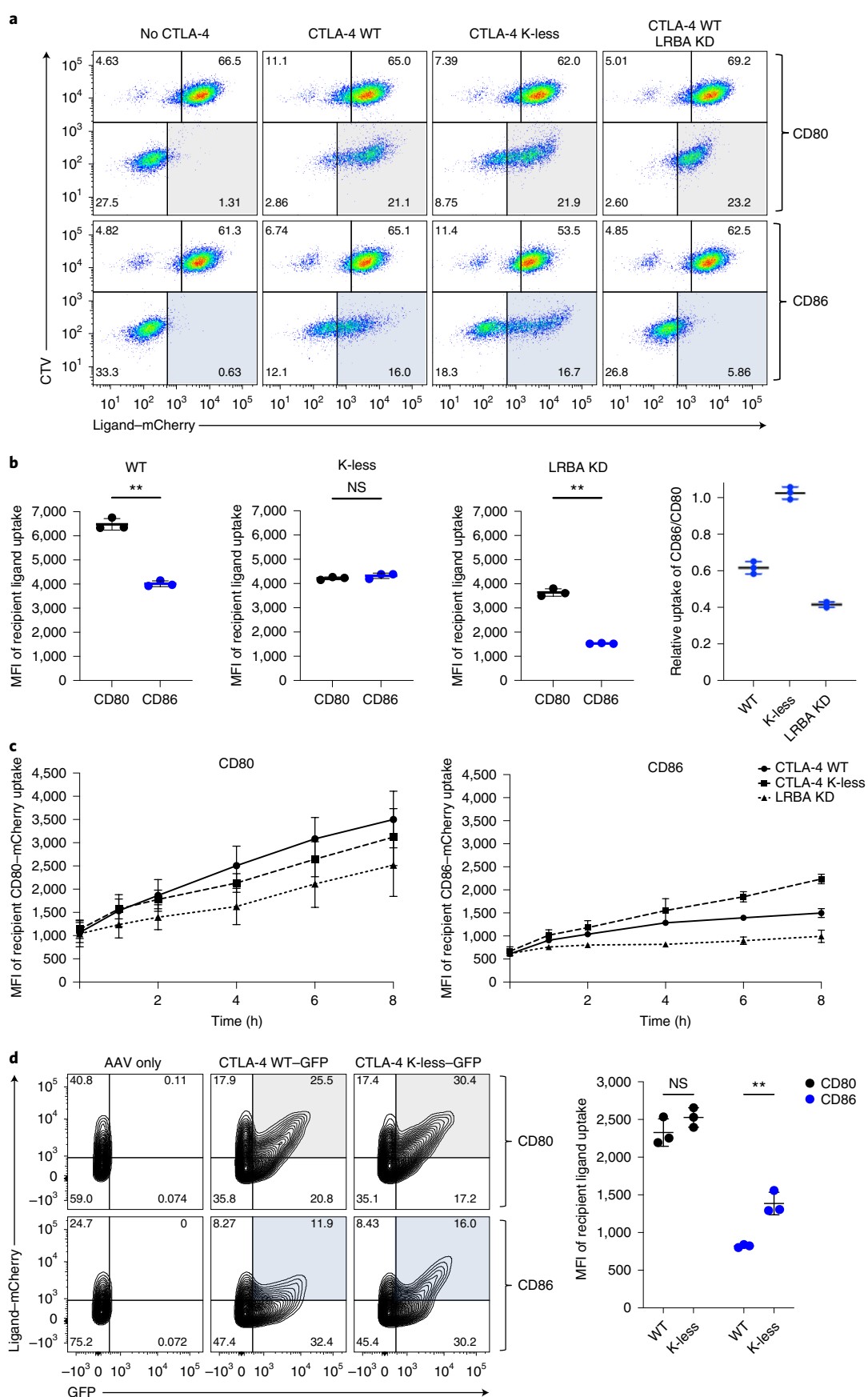

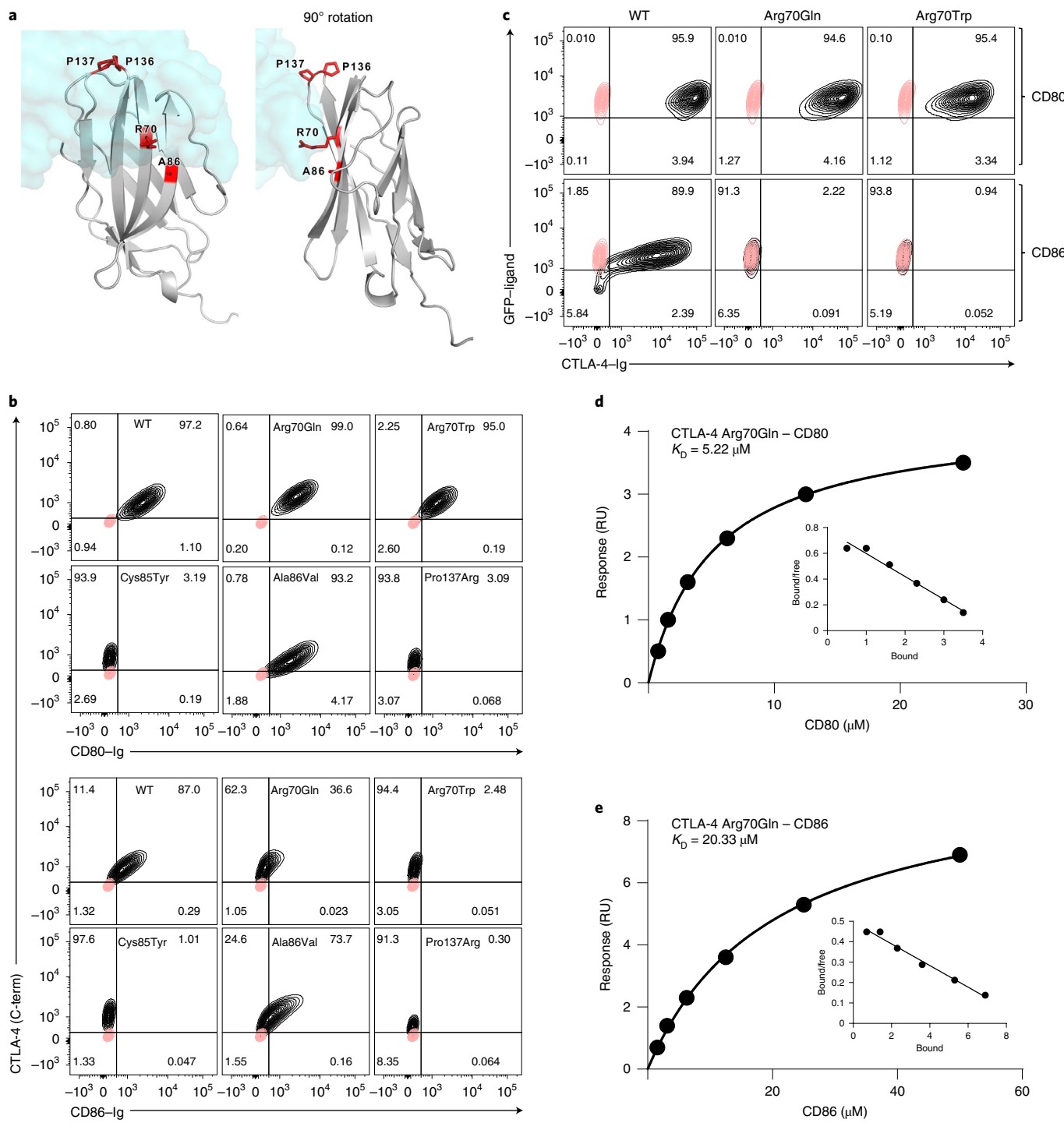

**Fig. 5 | CTLA-4 Arg70 mutations allow binding to CD80 but not CD86. a**, Patient-identified CTLA-4 mutations resulting in ligand-facing amino acid changes (red) mapped to CTLA-4 (ribbon structure) and the location of bound CD80 (blue space-filling structure). The right-hand structure shows a view after 90° rotation. **b**, FACS analysis of CD80–Ig binding and CD86–Ig binding to CTLA-4 WT or mutant proteins (Arg70Gln, Arg70Trp, Cys85Tyr, Ala86Val and Pro137Arg) expressed in CHO cells. Binding of Ig fusions to CTLA-4 (*x* axis) at 37 °C for 1 h is plotted against a co-stain for total CTLA-4 (*y* axis) using a cytoplasmic (C-term) domain antibody (C-19). Staining of CTLA-4⁻ control cells is shown in the red contours. **c**, Mutant CTLA-4 Ig proteins (Arg70Gln, Arg70Trp) or CTLA-4 WT Ig was used to stain CD80–GFP- and CD86–GFP-expressing CHO cells. Red contours show anti-Ig control staining in the absence of CTLA-4 Ig. **d**, Calculated monomeric affinity of the CD80–CTLA-4-Arg70Gln interaction, based on binding of soluble monomeric CD80 to immobilized Arg70Gln–Ig on the sensor. **e**, Calculated monomeric affinity of the CD86–CTLA-4-Arg70Gln interaction, based on binding of soluble monomeric CD86 to immobilized Arg70Gln–Ig on the sensor.

CD86–Ig). As expected, Pro137Arg mutations, in the established ligand-binding site 'MYPP¹³⁷PYY' motif, ablated binding to both CD80 and CD86, as did a control patient mutation Cys85Tyr which

disrupted disulfide bonds important for the Ig fold (Fig. 5b). In contrast, the Ala86Val mutation did not appear to dramatically alter ligand binding. However, two independent mutations at arginine

70 (Arg70Gln/Trp) both revealed that, although binding to CD80–Ig was readily detectable, binding of CD86–Ig was lost, suggesting that this mutation may result in a CTLA-4 protein that selectively interacts with CD80 (Fig. 5b). Staining using a carboxy-terminal anti-CTLA-4 antibody was used to control for the expression level of each mutant. To confirm these observations and to control for the impact of Ig fusion on CD86 valency, we repeated these experiments in reverse using CTLA-4–Ig (WT or Arg70Gln/Trp–Ig) binding to CD80- or CD86-expressing cells. This also showed that soluble Arg70Gln and Arg70Trp CTLA-4–Ig fusion proteins bound to CD80 but not CD86, in a setting where CD80 and CD86 ligands were expressed in their correct cellular configurations and where natural avidities are preserved (Fig. 5c). We therefore concluded that Arg70 mutations resulted in a loss of detectable binding to CD86 but not CD80.

To quantify the binding of Arg70Gln further, we performed surface plasmon resonance-based binding experiments to measure interaction affinity (Fig. 5d,e). This showed that Arg70Gln–CTLA-4 bound to CD80 with an affinity of ~5 μM, which, although reduced compared with WT CTLA-4 (~0.2 μM), is in a range similar to the known CD86–CTLA-4 affinity (~2 μM)[20]. In contrast, CD86–CTLA-4–Arg70Gln interactions were found to be much weaker, at ~20 μM. Importantly, the 5-μM monomeric affinity of Arg70Gln–CD80 is avidity enhanced in cellular settings because CD80 is a dimer. This indicated that, although Arg70Gln–CTLA-4 had reduced affinity, it binds to CD80 more strongly than CTLA-4 normally binds to CD86.

We therefore tested the function of CTLA-4 Arg70 mutants in TE assays using Jurkat T cells. In line with the above binding data, analysis in this functional assay revealed robust removal of CD80 by Arg70 mutants; however, CD86 TE was abrogated (Fig. 6a,b). Although CD80 was similarly downregulated by both the Arg70 mutants (Fig. 6b), we did not observe downregulation of CD86 by either CTLA-4 Arg70 mutant (Fig. 6a,b). Intriguingly, CD80 detection inside the mutant CTLA-4-expressing cells was now more sensitive to pH neutralization by BafA (Fig. 6c). Indeed, several features of CTLA-4 Arg70–CD80 interactions now resembled lower-affinity CTLA-4–CD86 behavior (Extended Data Fig. 6), including less robust co-localization between CTLA-4 Arg70 mutants and CD80 and enhanced pH dissociation.

To formally establish the impact on TE in primary $T_{reg}$ cells, we again used the CRISPR–Cas9/AAV6 knock-in strategy to replace the endogenous *CTLA4* gene in $T_{reg}$ cells with either WT or Arg70Gln cDNA. This allowed us to express Arg70Gln under the endogenous CTLA-4 promoter in its normal physiological cell type and then assess TE. After gene editing, Arg70Gln-edited cells (GFP+) clearly retained the capacity to perform CD80 TE as shown by mCherry ligand uptake (Fig. 6d, gray-shaded quadrants), but were defective in CD86 TE (Fig. 6d,e, blue-shaded quadrants). Moreover, the levels of CD80 uptake in both edited (Arg70Gln GFP+) and unedited (WT GFP−) $T_{reg}$ cells in the same assay were remarkably similar, reinforcing the functional capability of Arg70Gln despite its reduced CD80 affinity. In contrast, control targeting of the CTLA-4 locus with a

WT CTLA-4 cDNA template continued to allow robust TE of both CD80 and CD86 by the edited GFP+ population (Fig. 6d), confirming that the knock-in strategy itself did not affect the function of CTLA-4. These results indicated that a clinically significant mutation at Arg70, expressed from its endogenous locus in human $T_{reg}$ cells, caused a specific defect in CD86 TE.

**CTLA-4 Arg70Gln cannot regulate a CD86-driven T cell response.** Finally, we tested the functional consequence of mutations at Arg70, for the control of human T cell proliferation. Human CD4 T cells were stimulated using DG-75 B cells expressing either CD80 or CD86 in a co-stimulation assay. The B cells were pre-exposed to Jurkat cells expressing either WT CTLA-4 or the Arg70Gln variant to allow TE to occur, before fixation to preserve ligand expression. This revealed that the level of CD86–GFP on the B cells was effectively reduced by WT CTLA-4 TE but not by Arg70Gln (Fig. 7a, left column). In contrast, Arg70Gln cells effectively downregulated CD80 to a similar extent to WT-expressing cells (Fig. 7b, left column). Accordingly, T cells were able to respond to CD86 co-stimulation from B cells exposed to Arg70Gln, whereas CD80-driven T cell responses were suppressed (Fig. 7b, right column). Measuring the frequency of T cells responding to stimulation, in multiple independent donors, showed that control of CD86-driven T cell responses by Arg70Gln was completely defective, whereas CD80 control by Arg70Gln was virtually indistinguishable from WT (Fig. 7c). Overall, we concluded that CTLA-4 Arg70Gln lacks the ability to control CD86-driven T cell responses due to a selective defect in CD86 TE. Taken together, the above data identify a number of features that suggest a new model for differential functions of CD80 and CD86 (Extended Data Fig. 7).

## Discussion

The CD28–CTLA-4 pathway is an integrated system for regulating T cell activation and preventing autoimmunity, based on the balance of four competing interactions with both stimulatory and inhibitory outcomes[15,24]. Although the stimulatory impact of CD28 and the inhibitory action of CTLA-4 on T cells are well described, the effects of the two ligands are enigmatic. Both ligands are capable of CD28 co-stimulation[40–42]; however, their effects on CTLA-4 behavior have not been well characterized.

Our data highlight a number of key differences between CD80 and CD86 in their interactions with CTLA-4. The stable dimer–dimer interaction between CD80 and CTLA-4 results in ubiquitylation of CTLA-4 and targeting of the CTLA-4–CD80 complex toward lysosomes. Due to continued occupancy by CD80 and routing of the complex via the ubiquitin pathway toward late endosomes (Rab7+ and LAMP3+), CTLA-4 molecules appear to be disabled by CD80 binding. In contrast, we did not observe these features for CD86, which rapidly dissociates from CTLA-4 and did not trigger CTLA-4 ubiquitylation. As, by default, unligated CTLA-4 recycles to the plasma membrane[31], CD86 disengagement releases unmodified CTLA-4 for recycling, whereas CD86 itself is rapidly degraded. The concept that recycling of CTLA-4 is critical in human immune

**Fig. 6 | CTLA-4 Arg70 mutants are defective in CD86 TE. a,** CTLA-4 WT or mutant proteins (Arg70Gln and Arg70Trp) expressed in Jurkat T cells tested for TE of CD80–mCherry and CD86–mCherry from ligand-expressing DG-75 B cells. CTV-labeled, ligand-expressing cells were incubated with CTLA-4-expressing cells (CTV−) and assessed for TE overnight. Detection of CD80 and CD86 acquisition is highlighted in the gray- and blue-shaded quadrants, respectively. **b,** Quantification of ligand remaining on the donor cell relative to no CTLA-4 controls. **c,** The amount of CD80–mCherry ligand detected inside CTLA-4+ recipient cells shown for the CTLA-4 mutants. BafA was added to evaluate the impact of lysosome blockade. **d,** The impact of Arg70Gln mutation on TE by human $T_{reg}$ cells. Arg70Gln or WT CTLA-4 was knocked into the endogenous CTLA-4 locus by HDR using a CRISPR–Cas9/AAV6 system. Knock-in cells were detected using a GFP reporter. GFP+ $T_{reg}$ cells expressing Arg70Gln mutant cDNA were compared with endogenous CTLA-4 (GFP−) or GFP+-expressing $T_{reg}$ cells containing WT cDNA for their ability to capture CD80 (gray quadrant) or CD86 (blue quadrant) from DG-75 B cells. **e,** Quantification of the experiment shown in **d**, using data from three independent samples. The statistical significance was determined by two-way ANOVA with Sidak's multiple comparison correction (**b** and **c**) or two-tailed, unpaired Student's *t*-test (**e**): **P < 0.01, ***P < 0.001, ****P < 0.0001. All data are presented as mean ± s.d. from three independent experiments (**b** and **c**) or three biologically independent samples (**e**).

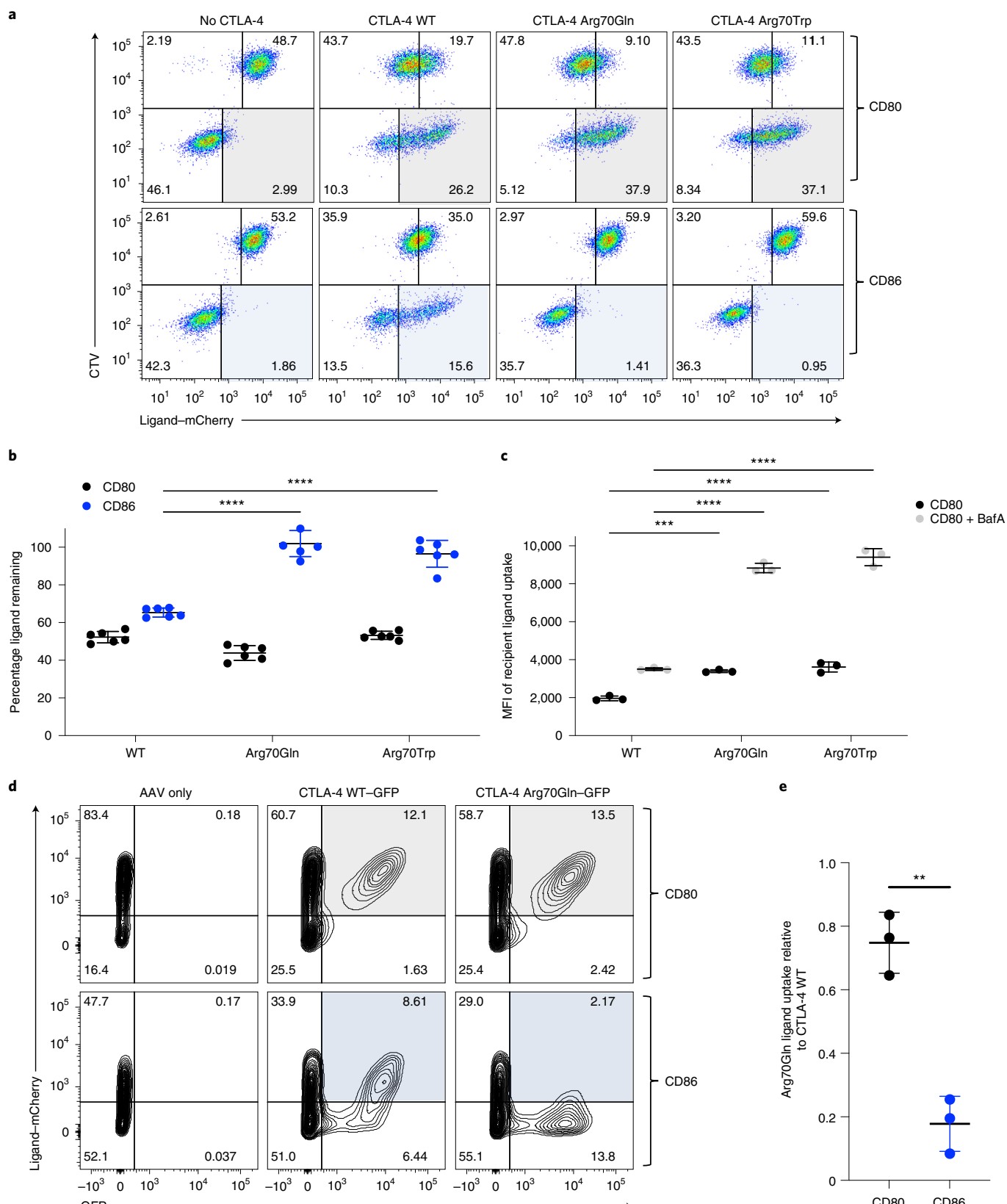

function is supported by the observation that defects in this pathway due to LRBA deficiency are associated with autoimmunity[34,36].

The observed differences between CD80 and CD86 behavior fit well with their known biophysical characteristics. CD80 has approximately tenfold higher monomeric affinity for CTLA-4 than CD86 and CD80 is a homodimer with the potential to form an avidity-enhanced lattice with CTLA-4 (refs. [20,22]). Such extended dimer–dimer interactions may also have the capacity to generate highly concentrated regions of CD80 and CTLA-4 at the cell–cell interface, which could facilitate the activation of

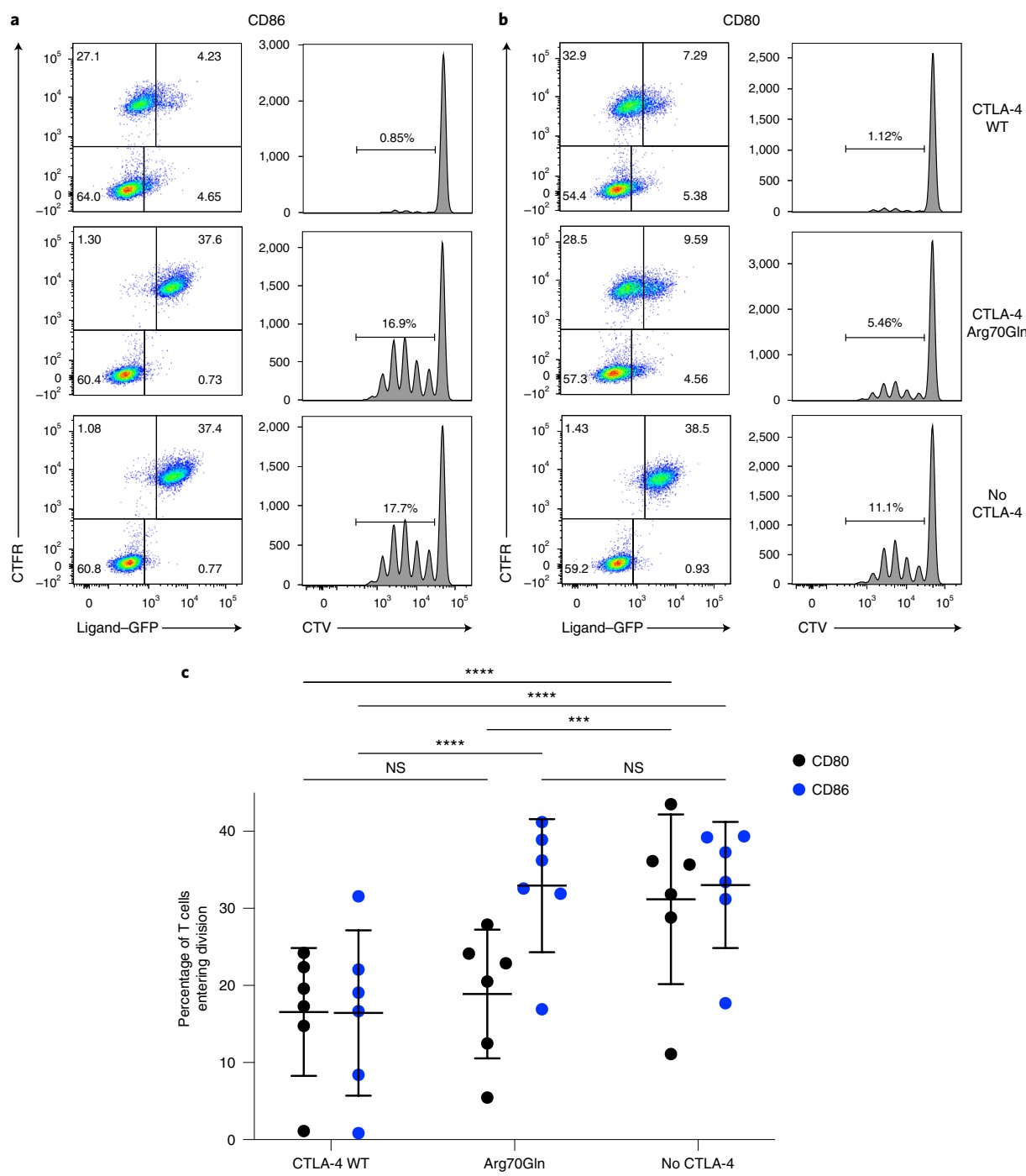

**Fig. 7 | CTLA-4 Arg70Gln is unable to regulate a CD86-driven T cell response. a,b,** CTFR-labeled CD86-expressing (**a**) or CD80-expressing (**b**) DG-75 B cells exposed to Jurkat T cells expressing no CTLA-4, CTLA-4 Arg70Gln or CTLA-4 WT overnight in a TE assay. TE of GFP–ligand is shown in the left-hand column. After TE, B cells were then used to co-stimulate T cell proliferation of CTV-labeled human CD4+CD25− T cells (representative histograms). T cell receptor stimulation was provided by soluble anti-CD3 antibody and proliferative responses measured by flow cytometry at day 5 for CTV dilution. **c,** Quantification of the experiment shown in **a** and **b**, using data from six individual donors, showing the percentage of T cells in the initial culture that entered cell division. The statistical significance was determined by two-way ANOVA with Sidak's multiple comparison correction: ***$P < 0.001$, ****$P < 0.0001$ All data are presented as mean ± s.d. and show individual data points from six biologically independent samples.

ubiquitin machinery. It is of interest that substrate clustering is known to be able to activate some ubiquitin ligases[43]. We also show directly that removal of the lysine residues within the CTLA-4 cytoplasmic domain prevented ubiquitylation, increased protein stability and enhanced its recycling. These concepts share obvious parallels with other trafficking receptors such as epidermal growth factor receptor where different ligands can profoundly influence dimerization, internalization, ubiquitylation and receptor recycling[44]. The fact that we did not readily observe CD80–CTLA-4 complexes co-localizing with LRBA suggests that ubiquitylation targets CTLA-4 away from recycling, as seen for other receptors[44].

In contrast to the high-avidity CD80–CTLA-4 interaction, CD86 is a monomeric ligand with relatively weak affinity for CTLA-4. Consistent with these features, we observed that CD86 readily dissociates from CTLA-4 in a pH-sensitive manner, allowing us to observe 'free' CD86 within the cell. It is interesting that the CD86–GFP signal itself was more sensitive to lysosomal pH inhibitors than CD80–GFP, indicating potentially different degradation pathways and kinetics for the two ligands within the cell.

Several lines of evidence suggest that CD86 permits a recycling itinerary for CTLA-4. CD86–CTLA-4 could be observed in LRBA+ compartments and loss of LRBA dramatically affected CD86 TE. Previous data have shown that LRBA deficiency leads to substantial immune dysregulation due to defects in CTLA-4 function[33,34,38]. One possibility is that LRBA is responsible for an early sorting event directing CTLA-4 into the recycling pathway[34,36]. Indeed, our own recent data indicate that LRBA is probably upstream of Rab11 and required for CTLA-4 entry into this pathway[37]. Therefore, in the absence of LRBA, sorting of CTLA-4 via the recycling pathway is impaired and CTLA-4 degradation is increased, consistent with data from patient T cells[34,38]. It is noteworthy that the reduced levels of CTLA-4 expression seen in LRBA-deficient $T_{reg}$ cells are substantially reversed on activation, as new CTLA-4 synthesis is upregulated[38]. Given that $T_{reg}$ cells are thought to be dependent on activation, a simple defect in CTLA-4 expression might therefore be insufficient to explain disease. Instead, our data support the possibility that TE of CD86 remains compromised in LRBA-deficient individuals due to impaired CTLA-4 recycling, even if CTLA-4 expression is substantially corrected by activation. A further line of evidence implicating the importance of CD86 regulation comes from CTLA-4 mutations found in patients with immune dysregulation[39]. These individuals suffer from a variety of autoimmune features[12,13,39,45] and we show in the present study that patient-derived CTLA-4 Arg70Gln mutations associated with autoimmunity lose the ability to transendocytose CD86.

Given the very large avidity advantage of CD80 over CD86 (ref. [20]), the fact that capture of CD86 by TE is at all comparable to CD80 is remarkable. Our data suggest that CD86 exploits a different approach, where its low-affinity monomeric interaction becomes advantageous. After TE, CD86 and CTLA-4 dissociate, permitting unmodified CTLA-4 to be recycled and compensating for the intrinsically weak interaction. In contrast, high-affinity crosslinking by CD80 drives long-lived binding and ubiquitylation of CTLA-4, thereby preventing useful recycling. It is interesting that this concept fits with our previous mathematical modeling of the CTLA-4 system, which predicts that optimal TE rates are not based on the highest possible affinity[46]. Therefore, from a biological standpoint these features might allow CD86 to be continuously and efficiently removed during interactions between $T_{reg}$ cells and antigen-presenting cells (APCs) where the contact times may be limited.

Given that, in two independent disease-related settings (LRBA KO and CTLA-4 Arg70 mutation), we identify selective defects in CD86 TE, this suggests that CD86 is an important target for immune regulation and control of CD28 co-stimulation. In mice, deficiency of B7-2 (CD86) has broader impacts on antibody class switching compared with CD80, particularly in the absence of adjuvant[40]. It is interesting that we have observed that antibody blockade experiments show a more inhibited phenotype when CD86 is blocked compared with CD80, even when both ligands are co-expressed. In addition, we have observed that CD86 is a more effective CD28 ligand for stimulating activated T cells and $T_{reg}$ cells in the presence of high levels of CTLA-4, where it has an advantage in sustaining extended cell division and higher levels of CD25, inducible co-stimulator and CD40L[42,47]. Thus, in an activated immune system where CTLA-4 is expressed on all activated T cells, CD86 is likely to be the more important CD28 ligand, capable of sustaining stimulation in the face of CTLA-4 competition.

Finally, our data suggest that CD80 may act as an attenuator of CTLA-4 function by remaining bound and altering its trafficking. Given its greater affinity for CTLA-4, CD80 could act as a regulator, protecting CD86 from CTLA-4 TE, and thereby increasing CD86–CD28 co-stimulation. Such a function could occur only when both ligands are present and would therefore not be seen in settings of single ligand deficiency that occur in KO mice. This predicts that, in some settings, inhibitory effects of CD80 blockade might be due to release of CTLA-4, allowing it to inhibit CD86–CD28 co-stimulation. Overall, our data reveal clear functional differences between CD80 and CD86 and their interactions with CTLA-4, providing a new framework for understanding this critical immunoregulatory system.

## Online content

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

## Methods

**Tissue culture and cell lines.** All cell lines were grown at 37 °C in 5% $CO_2$ in a humidified atmosphere. Jurkat E6.1T cells (American Type Culture Collection (ATCC), catalog no. TIB-152) and DG-75 B cells (ATCC, catalog no. CRL-2625) were grown in complete RPMI (Roswell Park Memorial Institute) 1640 medium supplemented with 10% fetal bovine serum (FBS), 2 mM L-glutamine, 100 U ml$^{-1}$ of penicillin and 100 mg ml$^{-1}$ of streptomycin (all from Life Technologies, Gibco). CHO-K1 (ATCC, catalog no. CCL-61) and HeLa adherent cell lines (ATCC, catalog no. CRL-1958) were maintained in complete Dulbecco's modified Eagle's medium (supplemented with 10% FBS, 2 mM L-glutamine, 100 U ml$^{-1}$ of penicillin and 100 mg ml$^{-1}$ of streptomycin—all from Life Technologies, Gibco). Cells were routinely detached using trypsin–EDTA and passaged 1 in 10.

**$T_{reg}$ cell isolation and expansion.** For $T_{reg}$ cell isolation, CD4$^+$ T cells were enriched by dilution of leukocyte cones (1:5) with phosphate-buffered saline (PBS), before addition of RosetteSep Human CD4$^+$ T Cell Enrichment Cocktail (STEMCELL Technologies) as per the manufacturer's instructions. Blood was layered over Ficoll-Paque PLUS (GE Healthcare) and centrifuged at 1,200$g$ for 25 min with slow acceleration and no brake. The CD4-enriched layer was collected and washed twice in PBS by centrifugation for 10 min at 300$g$ with gentle braking. CD4$^+$CD25$^+$ cells were then isolated by immunomagnetic-positive selection using human CD25 MicroBeads II (Miltenyi Biotec) according to the manufacturer's instructions. CD4$^+$CD25$^-$ cells were retained for CRISPR–Cas9 HDR editing in some subsequent experiments, following activation. Enriched CD4$^+$CD25$^+$ cells were stained using an antibody cocktail (anti-CD4 (RPA-T4), anti-CD25 (3G10) and anti-CD127 (A019D5)) and FACSAria sorting was used to sort CD4$^+$CD25$^+$CD127lo $T_{reg}$ cells.

For $T_{reg}$ cell expansion, DG-75 cell lines stably expressing CD86 were irradiated at 7,500 rad. Sorted $T_{reg}$ cell populations were plated at a 1:1 ratio with irradiated DG-75, with 100 ng ml$^{-1}$ of anti-human-CD3 (OKT3, BioLegend) and 1,000 IU ml$^{-1}$ of interleukin (IL)-2 (PeproTech). All $T_{reg}$ cells were maintained in complete OpTimizer Medium, supplemented with OpTimizer T-Cell Expansion Supplement, 2 mM L-glutamine, 100 U ml$^{-1}$ of penicillin and 100 mg ml$^{-1}$ of streptomycin (all from Life Technologies, Gibco). IL-2 was replenished every 2 d and $T_{reg}$ cells were restimulated every 7 d.

**Cell-line engineering.** Transduced cell lines contained stable integrations of transgenes that were generated using CTLA-4, CD80 or CD86 fusion proteins cloned into the MP71 retroviral vector. Retroviral supernatants were obtained by transfection of Phoenix-Amphoteric packaging cells (ATCC, catalog no. CRL-3213) in combination with the plasmid pVSV, using the FUGENE HD transfection reagent (Roche Molecular Biochemical). Retroviral supernatants were harvested 24 h post-transfection and used for transduction of CHO, Jurkat (CTLA-4) or DG-75 (CD80 and CD86) cell lines. For transduction, nontissue culture-treated, 24-well plates were coated with RetroNectin (TaKaRa) overnight at 30 mg ml$^{-1}$; $5 \times 10^5$ cells were added to 1 ml of retroviral supernatants in the RetroNectin pre-coated wells and centrifuged at 1,000$g$ and 32 °C for 2 h. Then, 24 h post-infection, the medium was changed to fresh medium appropriate to the cell type and 3 d post-transduction cells were screened by staining for transduced protein expression and analyzed by flow cytometry.

**TE assays.** For TE assays by flow cytometry: ligand donor cells were either CHO or DG-75 B cells expressing CD80 or CD86 molecules C-terminally tagged with either GFP or mCherry. Donor cells were labeled with CTV or Far Red labeling kits (Thermo Fisher Scientific) according to the manufacturer's instructions. CHO, Jurkat or $T_{reg}$ cells expressing CTLA-4 WT or mutants thereof were used as recipient cells. Donor and recipient cells or CTLA-4$^-$ control cells were plated in round-bottomed, 96-well plates at 37 °C at the ratio of donor:recipient cells and the times indicated in the figure legend. Typically, TE was observed between 1 h and 6 h depending on conditions. Where indicated, $NH_4Cl$ or BafA (Sigma-Aldrich) was used at 20 mM and 20 nM, respectively, to inhibit lysosomal degradation of transferred ligand, unless otherwise stated. BafA was generally used in assays with T cells whereas $NH_4Cl$ was used in CHO CTLA-4 cells due to better tolerability in each cell type.

For staining of CTLA-4 post-TE, cells were disaggregated by pipetting, before fixation in 4% paraformaldehyde (PFA) for 20 min at 22 °C. Cells were then permeabilized in 0.1% saponin for 10 min and stained for total CTLA-4 using antibodies to the cytoplasmic domain (F8-647 or C-19, Santa Cruz). After staining, singlets were gated, with total CTLA-4 level and GFP or mCherry uptake measured in recipient cells (CTV$^-$) by flow cytometry.

**Immunoblotting.** TE assays were carried out as above and, where indicated, both donor and recipient cells were treated with either $NH_4Cl$ or BafA as lysosomal acidification inhibitors or 10 μM MG132 (Sigma-Aldrich) proteasome inhibitor. Cells were subsequently washed with $Ca^{2+}$- and $Mg^{2+}$-free PBS (Thermo Fisher Scientific) and lysed with ice-cold lysis buffer (20 mM Tris-HCl, pH 7.5–8.0, 2 mM EDTA, 150 mM NaCl, 10% (v:v) glycerol, 1% (v:v) Triton X-100, 0.5% (w:v) CHAPS (3-((3-cholamidopropyl) dimethylammonio)-1-propanesulfonate), 1 mM dithiothreitol, 1 mM phenylmethylsulfonyl fluoride, 1 mM iodoacetamide, 2 mM

$N$-ethylmaleimide, 1× EDTA-free cOmplete protease cocktail inhibitor (Roche)) on ice for 30 min. Lysates were spun at 10,000$g$ for 10 min at 4 °C to clear the detergent insoluble fraction, before measuring total protein concentration using Precision Red protein assay. Samples were snap-frozen in liquid nitrogen and stored at −20 °C.

Lysates were separated by sodium dodecylfulfate (SDS)–polyacrylamide gel electrophoresis and transferred on to poly(vinylidene difluoride) membranes. Membranes were subsequently stained with anti-GFP (clones 7.1 and 13.1, Sigma-Aldrich) at 1:1,000 dilution. Anti-red fluorescent protein (RFP) mouse monoclonal (catalog no. 6G6, Chromotek), anti-CTLA-4 goat polyclonal (catalog no. C-19, Santa Cruz Biotechnology), anti-CTLA-4 rabbit monoclonal (catalog no. EPR1476, Abcam) and anti-tubulin mouse monoclonal (catalog no. DM1A, Sigma-Aldrich) were all used at 1:1,000 dilution for 12–16 h at 4 °C. Membranes were washed 3× for 10 min in Tris-buffered saline with 0.1% Tween (TBST) and stained with secondary antibody using either anti-mouse IgG horseradish peroxidase (HRP)-linked (Cell Signaling) antibody, goat anti-rabbit IgG HRP-linked (Cell Signaling) antibody or monoclonal (catalog no. GT-34) anti-goat/sheep IgG HRP-linked (Sigma-Aldrich) antibody, all at 1:1,000 dilution for 1 h at 22 °C. Membranes were washed and incubated with Clarity western ECL HRP-substrate (BioRad) for 3 min at 22 °C. The HRP signal was captured with the Chemidoc Touch system (BioRad) and visualized using Image Lab v.5.2.1 build 11 (BioRad). Membranes were stripped and probed for housekeeping gene tubulin by incubating in stripping buffer (1% Tween, 200 mM glycine, 0.1% SDS (w:v), pH adjusted to 2.2) for 10 min twice. Membranes were washed twice for 10 min in PBS and then washed twice in TBST for 5 min before continuing with antibody staining as above.

**Immunoprecipitation.** Cleared cell lysates were incubated with 10–20 μl of GFP–Trap or RFP–Trap agarose affinity peptide beads (Chromotek) for 2 h at 4 °C. Beads were washed 5× with wash buffer (20 mM Tris-HCl, pH 7.5–8.0, 2 mM EDTA, 150 mM NaCl, 10% (v:v) glycerol, 1% (v:v) Triton X-100, 0.5% (w:v) CHAPS and 1× EDTA-free cOmplete protease cocktail inhibitor). The beads were heated in NuPAGE LDS sample buffer (Thermo Fisher Scientific) plus NuPAGE sample reducing agent (Thermo Fisher Scientific) for 10 min at 100 °C, before being spun at 17,000$g$ for 1 min at 4 °C. For ubiquitin immunoprecipitation, cleared lysates were incubated with ubiquitin–Trap beads (catalog no. UBA01, Cytoskeleton) for 2 h at 4 °C. Beads were washed 5× and incubated with 4× NuPAGE LDS sample buffer at 22 °C for 5 min and spun at 17,000$g$ for 1 min.

**CTLA-4–ligand pH-dependent binding assay.** CHO cells expressing surface CD80–GFP or CD86–GFP were washed in ice-cold PBS and incubated with 18 μg ml$^{-1}$ of soluble IgG1–CTLA-4 (abatacept, Bristol-Myers Squibb) for 1 h at 4 °C. Cells were then washed with PBS once to eliminate excess abatacept before washing 3× with stripping buffer (150 mM NaCl, 100 mM glycine and 5 mM KCl) with pH adjusted to 7.4, 6.0, 5.0 and 4.5. After a 2 min incubation of cells with ice-cold stripping buffer, cells were spun at 500$g$ for 3 min and washed twice with ice-cold PBS, before lysing. The remaining abatacept bound to cells was assessed with immunoblotting using goat anti-human IgG HRP-linked (Thermo Fisher Scientific) antibody at 1:1,000 dilution.

**CRISPR for generation of cell lines.** CRISPR–Cas9 targeting was used for the generation of LRBA-deficient, CTLA-4 Jurkat T cell lines and the initial generation of CD80- and CD86-deficient DG-75 lines before transduction with tagged ligands. CRISPR–Cas9 target sites were identified using CHOPCHOP (https://chopchop.cbu.uib.no/) and in vitro single guide (sg)RNA syntheses containing the relevant target site were performed using the EnGen sgRNA Synthesis Kit, *Streptococcus pyogenes* (New England Biolabs) according to the manufacturer's instructions. Transcribed sgRNAs were purified using the RNA Clean & Concentrator kit (Zymo Research) following the manufacturer's instructions.

For generation of cell lines, 500 ng of sgRNA and 2 μg of Cas9 protein (TrueCut Cas9 Protein v.2) were electroporated into $2 \times 10^5$ target cells using the Neon Transfection System (Thermo Fisher Scientific) under the following conditions: voltage (1,600 V), width (10 ms), pulses (three), 10 μl tip and buffer R. Cells were allowed to recover for 3–5 d before screening for KO by flow cytometry. This approach generally yielded KO of the target gene in 70–95% of the cells. These cell populations were then sorted based on loss of expression of the target.

**CRISPR and homologous recombination in primary T cells.** CRISPR guide RNAs (gRNAs) were designed using the Benchling online tool (https://www.benchling.com/crispr). NGG protospacer adjacent motif sequences were identified toward the 3′-end of the first intron of *CTLA4* and assessed in silico for on-target and off-target activity using the Benchling online tool. A donor template was designed to incorporate a *CTLA4* cDNA followed by P2A sequence and GFP in front of the first CTLA-4 exon. This insert was then cloned into an adeno-associated virus type 6 (AAV6) vector. Site-directed mutagenesis was performed to create Arg70Gln or K-less mutant donor templates using the QuickChange II XL Site-Directed Mutagenesis Kit.

Recombinant (r)AAV vectors were produced with a standard double-transfection method that introduces an inverted terminal repeat-containing

transfer plasmid and a single helper plasmid, pDGM6 (obtained from the Russell laboratory at the University of Washington with permission), which contains the AAV2 rep and AAV6 cap proteins and the adenoviral proteins and RNA required for helper functions. Protocols for the production and purification of rAAVs were broadly as have been described[48–50]. In brief, vector production took place in HEK293T cells and were purified by iodixanol density gradient and ultracentrifugation. AAV6 particles were extracted using a needle and syringe between the 40% and 60% gradient interface and dialyzed 3× in 1× PBS (Thermo Fisher Scientific) with 5% sorbitol (Sigma-Aldrich) in the third step using 10K MWCO Slide-A-Lyzer Dialysis Cassettes (Thermo Fisher Scientific). Titration was performed using Quick Titre AAV Quantification Kit (Cell Biolabs) before aliquoting and storage at −80 °C before use.

**Electroporation and transduction of primary T cells.** Cas9 protein was purchased from New England Biolabs and synthetic gRNAs were custom-made by Synthego. Cas9 and gRNA were mixed at a 1:3 molar ratio and incubated at 25 °C for 30 min to form ribonucleoproteins (RNPs). A Lonza Nucleofector 4D was used for nucleofection (program EO-115) with a P3 Primary Cell 4D-Nucleofector Kit (Lonza). Then, $1 \times 10^6$ T$_{reg}$ cells per reaction were washed twice in PBS and resuspended in 15 µl per reaction of P3 nucleofector solution. Cells were mixed 1:1 with their respective RNP solution and transferred to the nucleofector strip. Immediately after nucleofection, 80 µl of warmed TexMACs (Milltenyi Biotech) medium was added to the cells and then 80 µl was transferred from the nucleofector strip to a 24-well plate containing artificial APCs in 920 µl of warmed TexMACs medium with IL-2 (100 units µl⁻¹) and aCD3 (100 ng ml⁻¹). AAV6 templates were added at 13,000 multiplicity of infection (MOI) (vector genomes per cell) within 15 min of nucleofection and incubated for 24 h. After 24 h, cell density was adjusted to $0.5 \times 10^6$ ml⁻¹ using TexMACs medium (with IL-2 100 units µl⁻¹). Cells were phenotyped >48 h after editing with FACS to assess editing efficiency.

**Transient expression of DN Rab11.** HeLa cells, $1 \times 10^6$, were transfected with 1.4 µg of plasmid DNA (DN Rab11–GFP or GFP vector backbone) using Lipofectamine 2000 (Thermo Fisher Scientific) according to the manufacturer's instructions. Cells were grown for 24 h to allow expression and then subjected to TE assays as above. Cells were placed on ice and analyzed by flow cytometry gating for medium DN Rab11 expression levels based on GFP expression.

**Confocal microscopy.** Cells expressing CTLA-4 were plated on to 13-mm diameter glass coverslips with CHO CD80/86–GFP at a 1:2 ratio for a total of $1.5 \times 10^5$ CHO cells per coverslip and left for 4 h to permit TE. Adhered cells were fixed and permeabilized in methanol for 10 min at −20 °C. Cells were then washed 3× in PBS and blocked with blocking buffer (PBS with 2% bovine serum albumin) for 1 h at 22 °C. Coverslips were stained with 1:250 dilution of goat monoclonal anti-CTLA-4 (C-19) primary antibody in blocking buffer for 1 h at 22 °C with agitation. Cells were washed and stained with donkey anti-goat Alexa Fluor-546 for 1 h at 22 °C in the dark. After multiple PBS washes, coverslips were mounted on to slides using Fluorofield mounting medium with DAPI.

T$_{reg}$ cells were stained with CTV and DG-75 CD80/86–mCherry B cells with CTFR and TE performed in suspension at a 2:1 ratio T$_{reg}$ cells:DG-75 in the presence of 1 µg ml⁻¹ of anti-CD3 (OKT3) and 10 ng ml⁻¹ of IL-2 for 6 h. Cells were then washed in ice-cold PBS and $2 \times 10^5$ cells transferred into a well of a 0.01% poly(L-lysine) (Sigma-Aldrich)-coated, 96-well plate on ice. Ice-cold 8% PFA in PBS was added 1:1 (v:v) and cells were fixed on to the fibronectin by centrifuging for 20 min at 500g. After sequential washes of 2% FBS in PBS, PBS and 0.1% saponin, cells were blocked, then stained with a 1:800 dilution of anti-CTLA-4 (C-19) primary antibody at 4 °C overnight on a rocker, washed and stained with donkey anti-goat Alexa Fluor-488 and 2 µg ml⁻¹ of DAPI for 45 min at 22 °C in the dark. After sequential 0.1% saponin, PBS and deionized water washes, cells were mounted in Mowiol with 2.5% DABCO (1,4-diazabicyclo(2.2.2)octane).

All confocal data were collected using an inverted Nikon Eclipse Ti equipped with a ×60 oil immersion objective. Constant laser powers and acquisition parameters were maintained throughout. Digital images and scale bars were prepared using Fiji. All images were analyzed using CellProfiler analysis software.

**Proximity Ligation Assay (PLA).** CHO cells expressing CTLA-4 were plated on to 13-mm diameter glass coverslips with CHO CD80/86–GFP at a 1:1 ratio for a total of $0.2 \times 10^6$ cells per condition and left for 18 h to permit adhesion and TE. All conditions contained 5 mM NH$_4$Cl to prevent ligand degradation. Cells were stained using the Duolink PLA reagents supplied in the In Situ Red Starter Kit for Mouse/Goat antibodies, according to the manufacturer's protocol. In brief, cells were fixed and permeabilized in methanol for 10 min at −20 °C. Cells were then washed in PBS 3× and blocked in blocking buffer at 22 °C for 1 h. Cells were stained with primary antibodies against CTLA-4 (C-19, Santa Cruz) and mouse monoclonal anti-pan ubiquitin (FK2) diluted in Duolink antibody diluent at a 1:500 dilution. Coverslips were washed and incubated with PLA PLUS and MINUS probes against mouse and goat antibodies, respectively, for 1 h at 37 °C. The oligonucleotides conjugated within the probes were ligated for 30 min at 37 °C using Duolink ligase and amplified for 100 min at 37 °C with Duolink polymerase,

which encodes a red fluorophore permitting visualization of the PLA signal. The PLA signal should be limited to where the primary antibodies are within 40 nm proximity. Coverslips were washed twice in Duolink wash buffer B before mounting on to slides using Duolink In Situ Mounting Medium with DAPI and left for at least 15 min before analysis by confocal microscopy.

**T cell proliferation assays.** To test the functional competence of CD80 and CD86 ligands after TE, DG-75 B cells (CellTrace Far Red (CTFR) labeled) expressing either CD80–GFP or CD86–GFP were cultured in the presence of Jurkat T cells expressing CTLA-4 WT, CTLA4 Arg70Gln or no CTLA4 at a Jurkat:DG-75 ratio of 1:1 for 21 h to promote ligand downregulation. These cells were then fixed in 0.025% glutaraldehyde (Sigma-Aldrich) for 3 min to preserve ligand expression levels and used as APCs.

CD4⁺CD25⁻ T cells were isolated from freshly derived peripheral whole blood. Peripheral blood monouclear cells were separated from whole blood by Ficoll gradient centrifugation (GM Healthcare) and T cells purified by negative selection using the Custom EasySep Human CD4⁺CD25⁻ T cell enrichment kit (STEMCELL Technologies). T cells were CTV stained and cultured for 5 d in the presence of 1 µg ml⁻¹ of soluble anti-CD3 (Clone OKT3, BioXCell) and fixed DG-75 cells in complete RPMI. Assays were incubated at 37 °C in 5% CO$_2$, and samples were acquired on a BD LSRFortessaII flow cytometer and analyzed using FlowJo (TreeStar).

To quantify suppression of T cell responses resulting from ligand TE, a known number of AccuCheck Counting Beads (Thermo Fisher Scientific) were added to each sample immediately before acquisition. Based on CTV dilution, the absolute number of cells within each division was calculated, based on the number of cells in each peak and adjusting for the number of beads acquired as a fraction of the total. The total number of cells from the original inoculum that committed to divide was then determined by dividing the absolute number of cells in each division by $2^i$, where '$i$' represents the division number. The percentage of T cells entering division was calculated from the total number of precursors that committed to division, divided by the total number of cells in the initial sample ×100. Changes in percentage of T cells entering division therefore reveal suppression of T cell responses based on the absolute cell numbers.

**Surface plasmon resonance.** Surface plasmon resonance experiments were performed using a Biacore T200 system (GE Healthcare). All assays were performed using a Sensor Chip Protein A (GE Healthcare), with a running buffer of 10 mM Hepes sodium salt, 150 mM NaCl, 0.05% sodium azide, pH 7.4, at 25 °C.

To determine the binding affinity between CD80 and CTLA-4 Arg70Gln, CTLA-4–Arg70Gln-Fc was immobilized on to the sample flow cell of the sensor chip, to the level of ~2,500 relative units (RU). The reference flow cell was immobilized with anti-SARS-CoV-2 antibody CR3022-Fc to the level of ~2,300 RU. Soluble CD80 was injected over the two flow cells at a range of eight concentrations prepared by serial twofold dilutions from 208.5 µM, at a flow rate of 20 µl min⁻¹, with an association time of 60 s. Running buffer was also injected using the same program for background subtraction. All data were fitted to a 1:1 binding model using GraphPad Prism v.9.0.1.

**Structural analysis.** All structural figures were produced using PyMOL (PyMOL Molecular Graphic System, Delano Scientific) with the following structures: unliganded CTLA-4 (Protein Databank (PDB) accession no. 3OSK)[51], CTLA-4 complexed with CD80 (PDB accession no. 1I8L)[52] and CTLA-4 complexed with CD86 (PDB accession no. 1I85)[53].

**Statistical analysis.** Statistical analyses and significance were determined using GraphPad Prism v.9.02 software (GraphPad Software Inc). All analyses were performed in triplicate or greater and the means obtained were used for independent Student's $t$-tests or two-way analysis of variance (ANOVA) with Sidak's correction for multiple comparisons. No statistical methods were used to predetermine sample sizes. Data distribution was assumed to be normal, but this was not formally tested. Data collection and analysis were not performed blind to the conditions of the experiments. Asterisks denote statistical significance (nonsignificant or NS, $^*P < 0.05$, $^{**}P < 0.01$, $^{***}P < 0.005$ and $^{****}P < 0.001$). All the data are reported as mean ± s.d. Representative experiments shown were generally repeated independently on at least three occasions with similar results.

**Reporting summary.** Further information on research design is available in the Nature Research Reporting Summary linked to this article.

## Data availability
Data generated or analyzed in the present study are available upon reasonable request. Source data are provided with this paper and are available in the main text or the supplementary materials.

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

## Acknowledgements

A.K., E.W., C.H., N.H., T.A.F. and D.M.S. were funded by the Wellcome Trust (grant nos. 204798/Z/16/Z, 110297/Z/15/Z and 102186/B/13/Z). B.R. and B.S. were funded by the Biotechnology and Biological Sciences Research Council (BBSRC; grant no. BB/M009203/1). C.W. was funded by Versus Arthritis (grant no. 21147). O.S.Q. was funded by BBSRC (grant no. BB/H013598). D.J. was funded by a BBSRC Case studentship. C.B. was supported by Great Ormond Street Hospital National Institute for Health and Care Research (NIHR) Biomedical Research Centre (BRC). A.M.P. was funded by Versus Arthritis Career Development fellowship (no. ARUK CDF 21738). E.C.M. was supported by the NIHR University College London Hospitals BRC. L.S.K.W. and D.M.S. were funded by the Medical Research Council (grant no. MR/N001435/1). This research was funded in whole, or in part, by the Wellcome Trust (grant no. 204798/Z/16/Z). For the purpose of Open Access, the author has applied a CC BY public copyright license to any author-accepted manuscript version arising from this submission.

## Author contributions

D.M.S., L.S.K.W., E.B. and S.J.D. conceived the study. A.K., D.J., C.H., O.S.Q., S.K., B.S., C.B., A.M.P. and S.J.D. provided the methodology. A.K., E.W., B.R., D.J., C.W., N.H., T.A.F., J.H., C.P., C.H. and S.K. performed the investigations. A.K., E.W., B.R., S.I. and D.M.S. visualized the study. D.M.S., E.B., E.M. and L.S.K.W. acquired the funds. D.M.S., L.S.K.W., E.B. and S.J.D. supervised the study. D.M.S. wrote the original draft of the manuscript. A.K., E.W., B.R., A.M.P., C.H., C.W., S.J.D., L.S.K.W., E.B. and D.M.S. reviewed and edited the manuscript.

## Competing interests

O.S.Q. is an employee of Celentix Ltd. The other authors declare no competing interests

## Additional information

**Extended data** are available for this paper at https://doi.org/10.1038/s41590-022-01289-w.

**Correspondence and requests for materials** should be addressed to David M. Sansom.

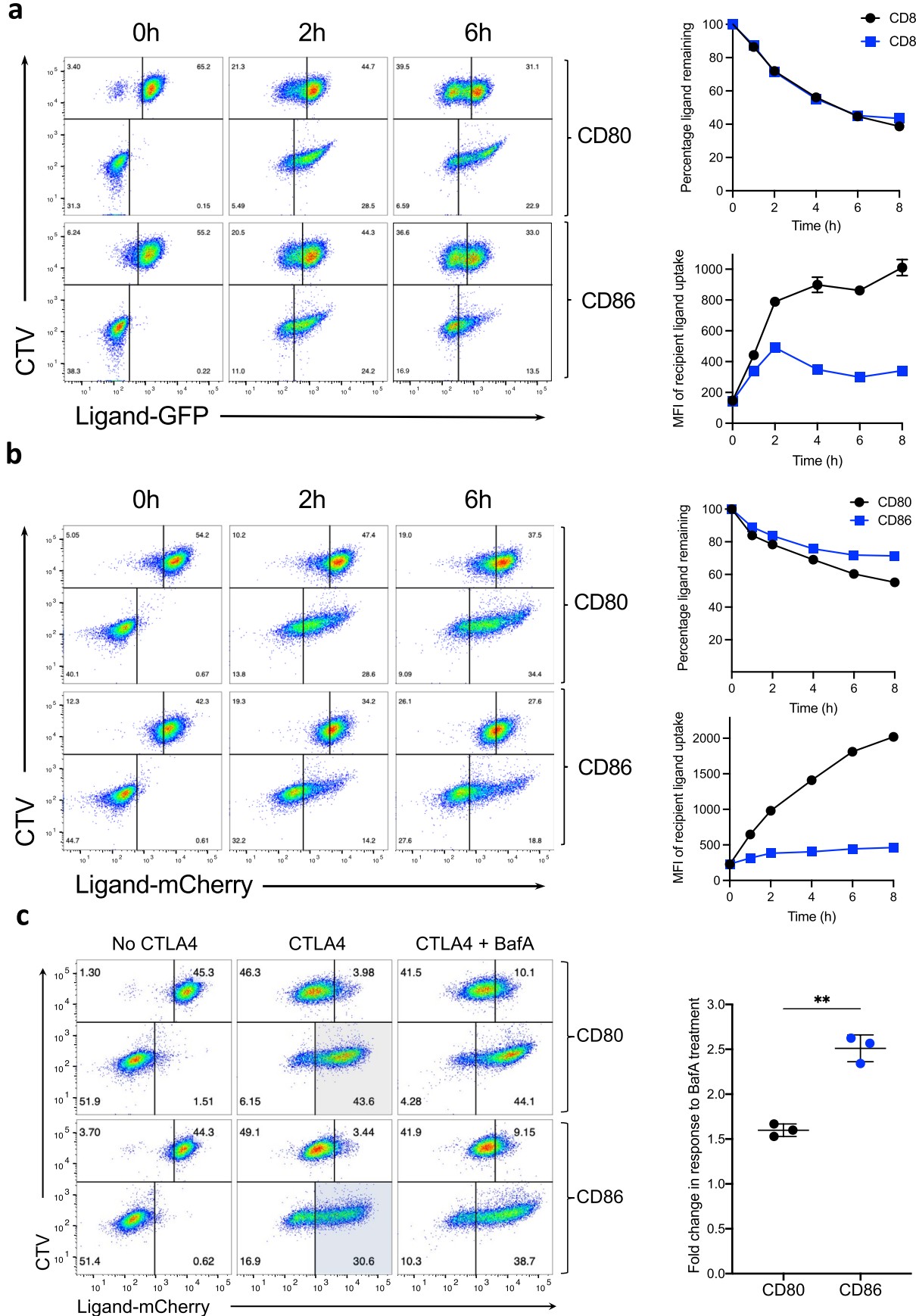

**Extended Data Fig. 1 |** See next page for caption.

**Extended Data Fig. 1 | Kinetic analysis of transendocytosis by flow cytometry.** Transendocytosis assays were performed with CTLA-4-CHO: ligand CHO cells **(a)** or CTLA-4-Jurkat: ligand DG75 B cells **(b)** for the times indicated. Representative FACS plots at time points indicated on the left, with full kinetic analysis of both ligand downregulation on donor cells and uptake by CTLA-4 recipient cells quantified on the right. **c)**. Example of transendocytosis using Jurkat cells without CTLA-4 (no CTLA-4) or expressing (CTLA-4) capturing ligand from CD80 or CD86-expressing DG75 B cells showing the impact of Bafilomycin A (+ BafA) treatment on detection of ligand in CTLA-4 expressing recipient cells. Detection of CD80 and CD86 acquisition is highlighted in the grey and blue shaded quadrants, respectively. Statistical significance was determined by paired t test, **P < 0.01. All data are presented as mean ± s.d. and show individual data points from 3 independent experiments.

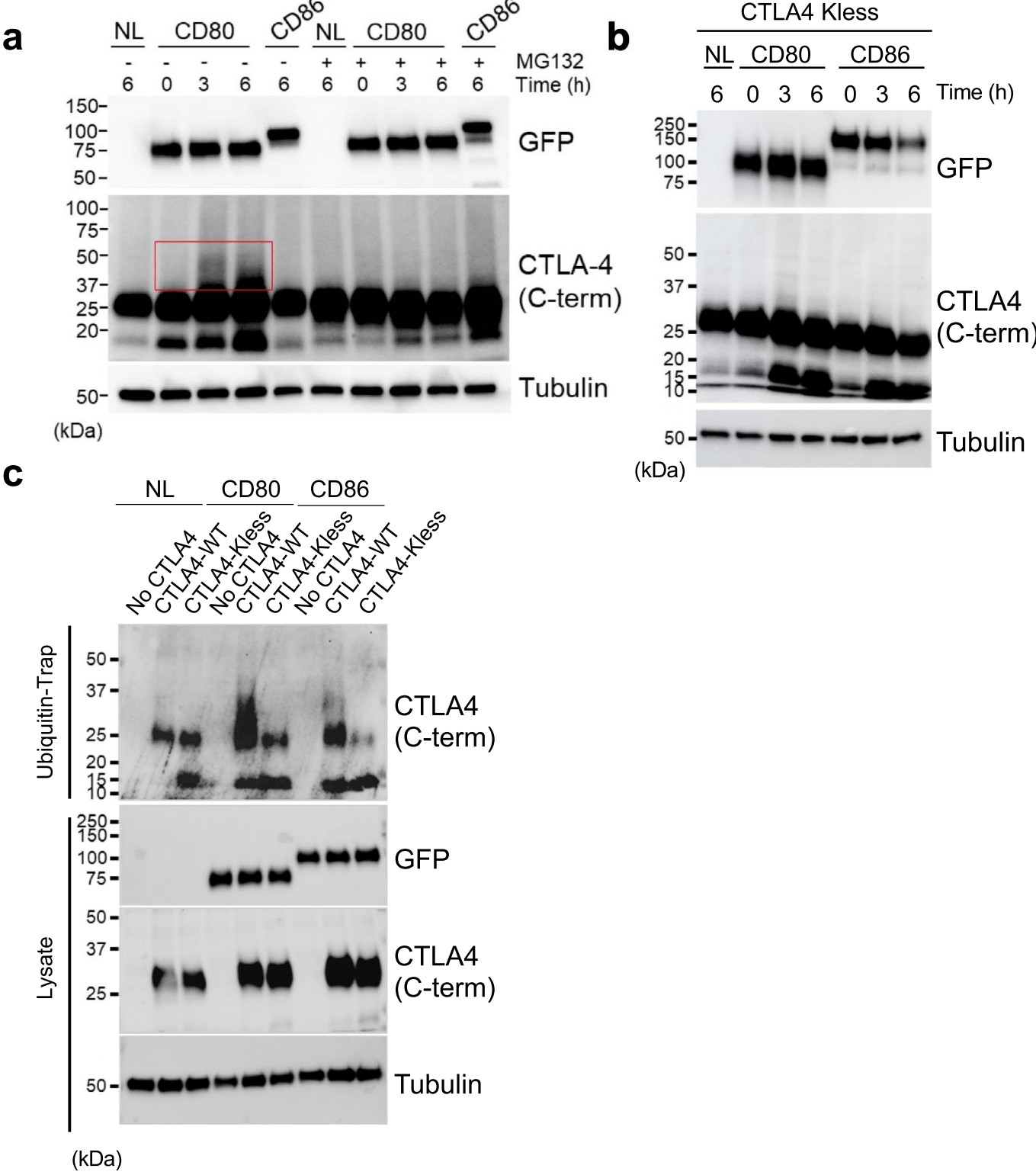

**Extended Data Fig. 2 | Impact of MG132 treatment and Kless mutation on CTLA-4 ubiquitylation. a)**. Transendocytosis was carried out using CD80-GFP or CD86-GFP expressing CHO cells or CHO cells with no ligand (NL) for the times shown and whole cell lysates were blotted for GFP (ligand), CTLA4 (C-term), with membranes stripped and reprobed for tubulin. CTLA-4 increases in Mw are highlighted (red box). **b)**. CTLA-4 lacking cytoplasmic lysine residues (CTLA-4 Kless) was used in transendocytosis assays with CHO cells expressing CD80, CD86 or CHO cells with no ligand (NL) for times indicated and whole cell lysates blotted for GFP (ligand), and CTLA-4. Lysates were blotted for tubulin as a sample processing control. **c)**. The impact of Kless mutation on ubiquitylation of CTLA-4 expressed in Jurkat T cells. Transendocytosis of DG75 B cells expressing CD80-GFP or CD86-GFP was carried out for 6 hours, followed by lysis and immunoprecipitation of total ubiquitin (ubiquitin trap). Blots were then probed for CTLA-4 expression using anti-CTLA4 antibody (C-term) and GFP (ligand). Whole cell lysates (WCL) were also blotted using anti-CTLA4, and tubulin to control for protein loading. All data is representative of at least 3 independent experiments.

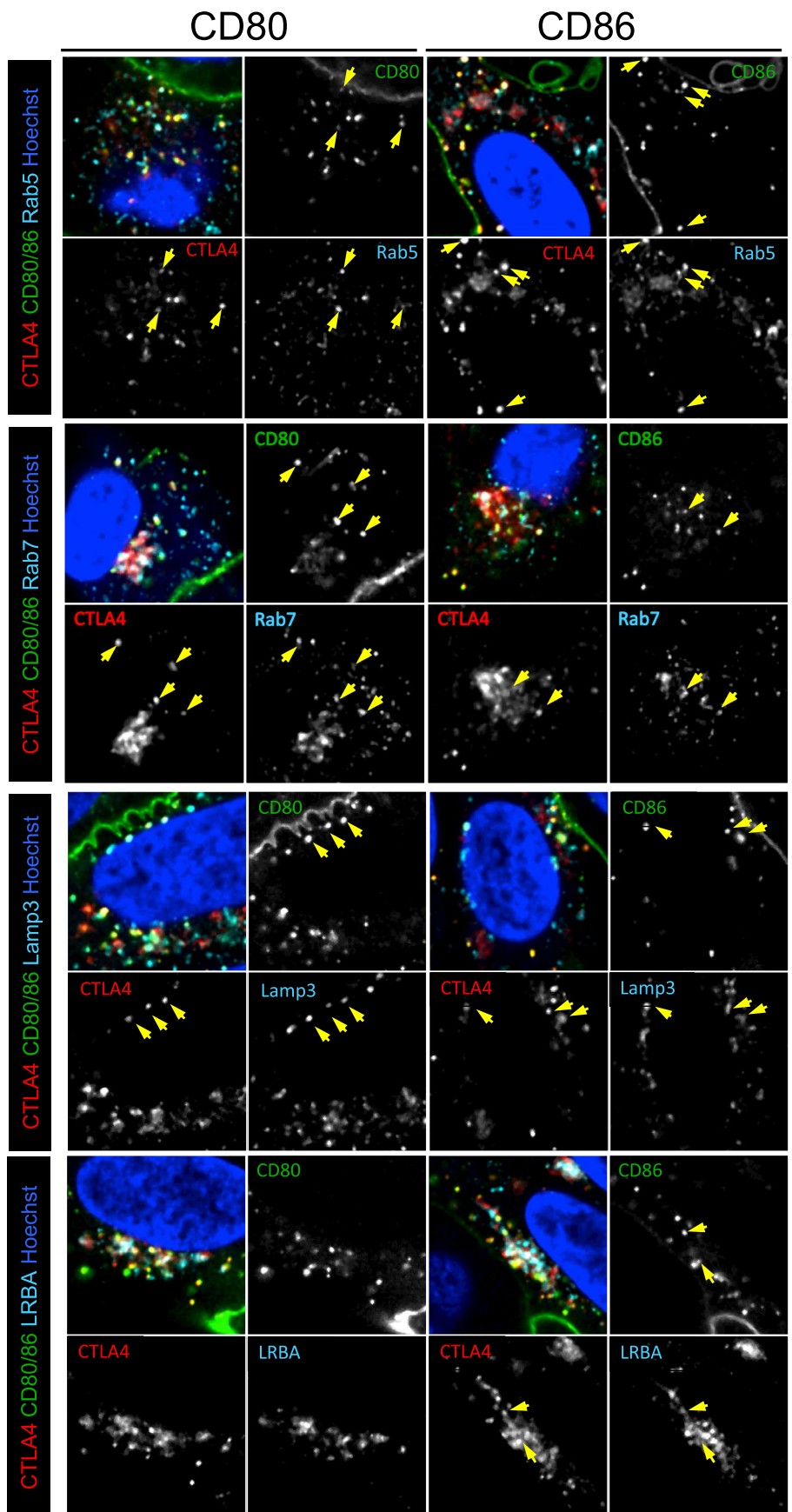

**Extended Data Fig. 3 | See next page for caption.**

**Extended Data Fig. 3 | Colocalization of CTLA-4, ligand and markers of intracellular trafficking.** Confocal analysis of overnight transendocytosis between CHO CD80GFP (left-hand panels) or CD86GFP (right-hand panels) and HeLa cells expressing CTLA-4. Following transendocytosis cells were fixed and stained for CTLA-4 (red), ligand (green) and components of the endosomal/lysosomal pathway (Rab5, Rab7, Lamp3 and LRBA) as indicated (cyan). Arrowheads indicate location of triple colocalization of ligand, CTLA-4 and indicated cellular component. Arrows are illustrative and do not show all colocalization events, which were determined automatically using Cell Profiler software and are quantified in **Figure 3b**.

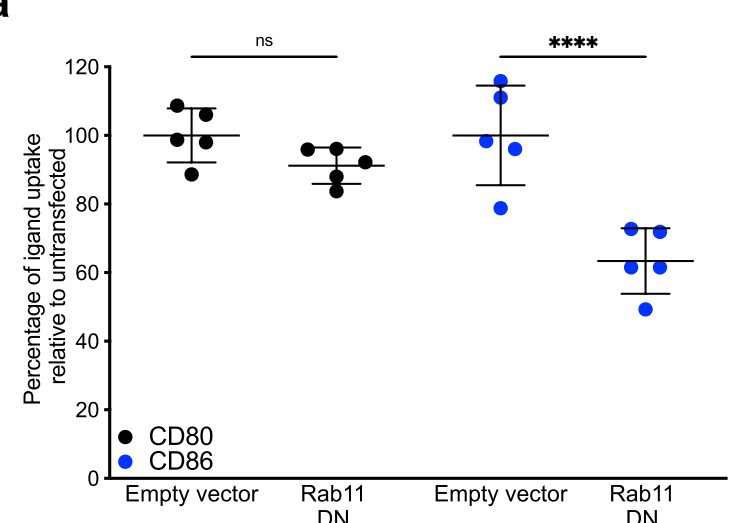

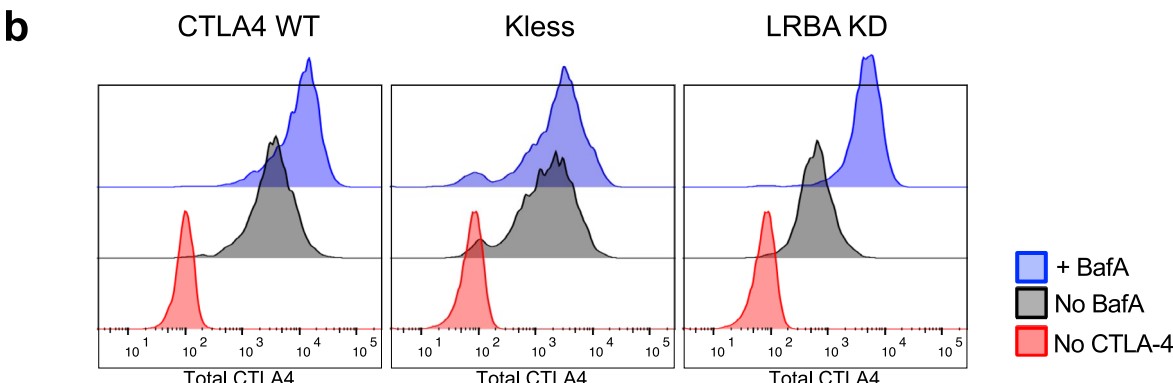

**Extended Data Fig. 4 | Inhibiting recycling using dominant negative Rab11 impairs CD86 transendocytosis. a).** CTLA-4+ HeLa cells were transiently transfected with dominant negative (DN) Rab11-GTPase or empty vector backbone and the impact on transendocytosis assessed using CHO mCherry-ligand uptake at 20h assessed by flow cytometry. Statistical significance was determined by 2-way ANOVA with Sidak's multiple comparison correction, ****P < 0.0001. All data are presented as mean ± s.d. and show individual data points from 5 independent experiments. **b). Impact of Kless and LRBA on CTLA-4 degradation**. Comparison of CTLA-4 sensitivity to BafA in WT, Kless and LRBA knockout Jurkat T cells. CTLA-4 degradation in the steady state was estimated by the impact of treatment with BafA on the staining for total CTLA-4 expression.

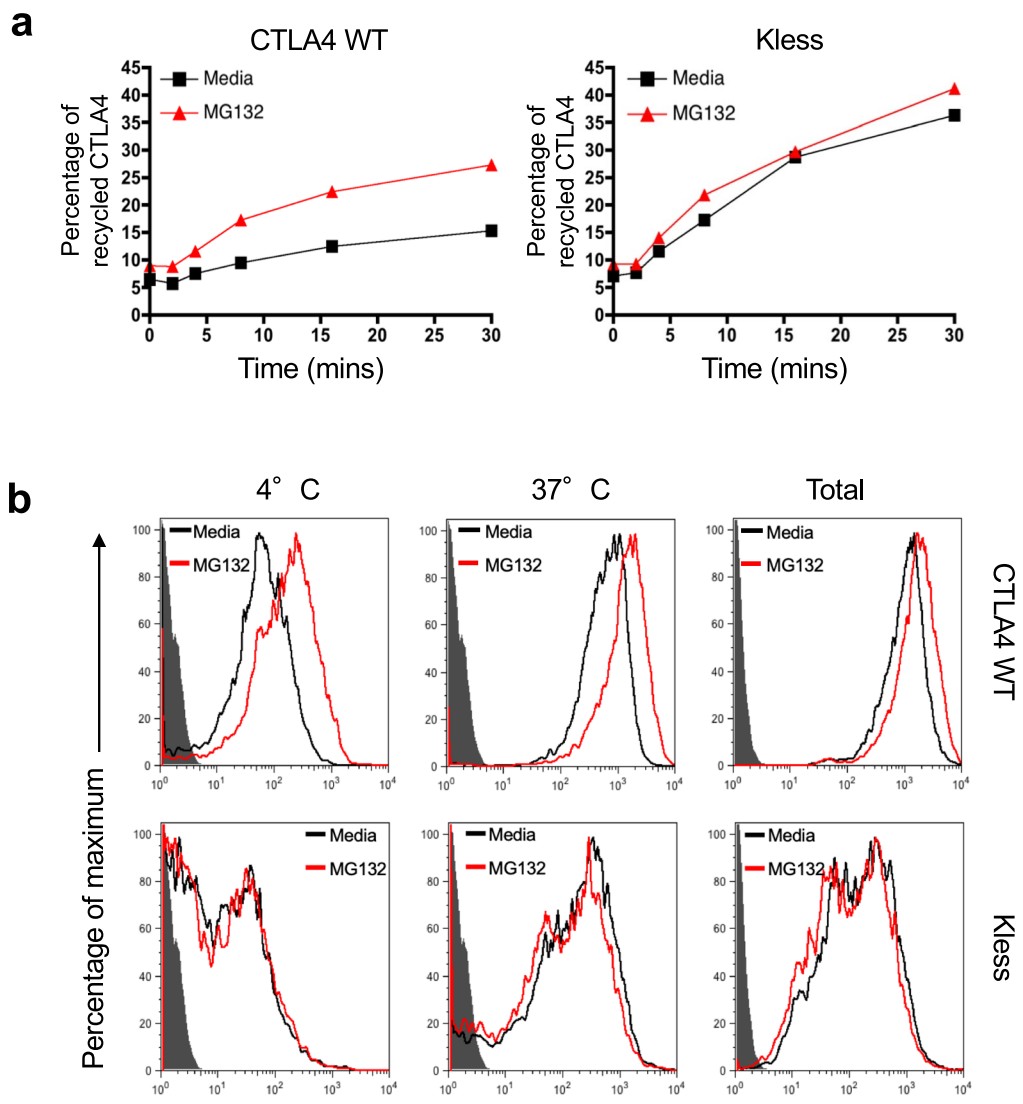

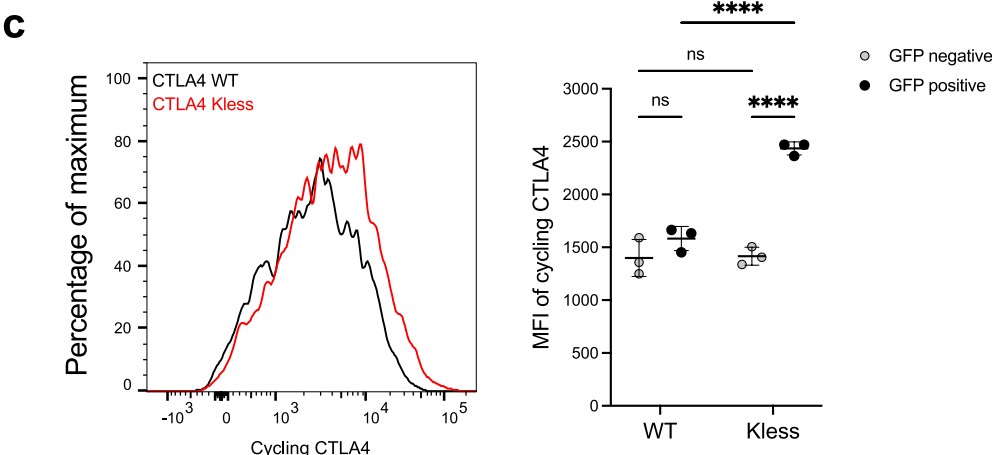

**Extended Data Fig. 5 | See next page for caption.**

**Extended Data Fig. 5 | Enhanced recycling and insensitivity to degradation in CTLA-4 Kless cells. a)**. CHO cells expressing WT or CTLA-4 lacking cytoplasmic lysine residues (Kless) were stained for CTLA-4 recycling using anti-CTLA-4 antibodies at 37 °C and detected by flow cytometry. The percentage of CTLA-4 recycled is shown in the presence and absence of MG132 to assess the impact of ubiquitylation. **b)**. Flow cytometry analysis of WT and Kless cells stained for CTLA-4 at 4°C (to stain cell surface), 37 °C (to stain cycling) and following fixation and permeabilisation (to stain total) in the presence or absence of MG132. **c)**. Impact of Kless mutation knocked in to the CTLA-4 locus in activated human CD4+ T cells using CRISPR-Cas9/AAV6 HDR. Cycling CTLA-4 was detected using anti-CTLA-4 antibody at 37°C for 1h in cells receiving either a WT or Kless CTLA-4 repair template (histograms). The amount of cycling CTLA-4 was compared between edited (GFP+) and non-edited (GFP−) cells in the same culture. Data shown is collated from 3 biologically independent samples. Statistical significance was determined by 2-way ANOVA with Sidak's multiple comparison correction, ****$P < 0.0001$. All data are presented as mean ± s.d. and show individual data points from 3 biologically independent samples.

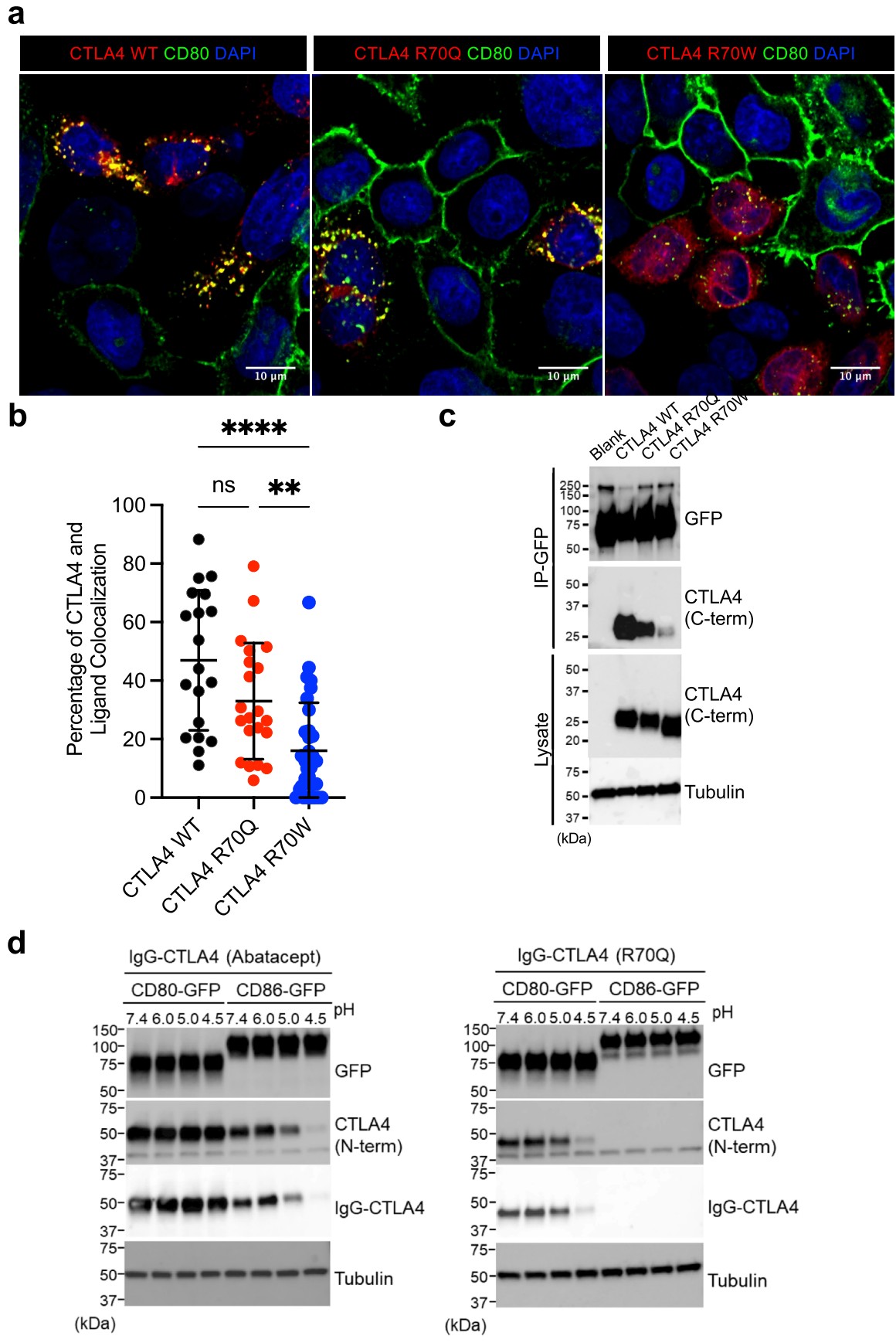

**Extended Data Fig. 6 | See next page for caption.**

**Extended Data Fig. 6 | R70 mutations in CTLA-4 cause CD80 to behave more like CD86. a)**. Confocal microscopy of CD80-GFP (green) with CTLA-4 WT, R70Q or R70W (red) following overnight transendocytosis in CHO cells. Scale bar, 10 μm. **b)**. Quantification of the experiment shown in **a**, showing the percentage of colocalization between CTLA-4 and CD80. Data shown is collated from a minimum of 4 images per condition. Statistical significance was determined by 2-way ANOVA with Sidak's multiple comparison correction, **P < 0.01, ****P < 0.0001. All data are presented as mean ± s.d. and show individual data points. n= 20-33 cells from 1 experiment, representative of 2 independent experiments c). Following transendocytosis by Jurkat T cells expressing CTLA-4 WT or R70 mutants, CD80 was immunoprecipitated via its GFP tag and blotted for the presence of co-precipitated CTLA-4. Data are representative of two similar experiments. **d)**. pH sensitivity of R70Q-Ig binding to CD80 expressing CHO cells. CHO cells expressing CD80 or CD86 were surface stained using CTLA-4 WT-Ig (Abatacept), or CTLA-4 R70Q-Ig and then washed at the pH shown. Cells were lysed and bound CTLA-4 was detected by Immunoblotting using anti-Human-Fc (anti-human-HRP) and anti-N terminal CTLA-4 (EPR1476). Data are representative of two similar experiments. Tubulin was used as a loading control in **c** and **d**.

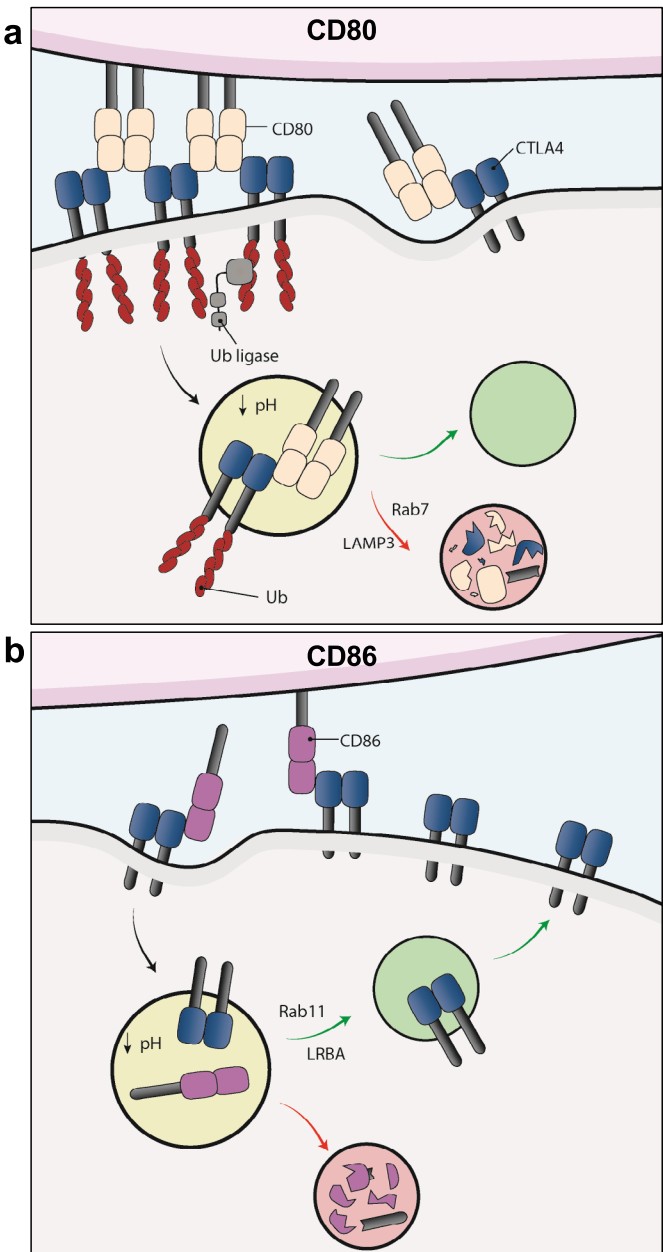

**Extended Data Fig. 7 | Proposed model of impact of CD80 and CD86 on CTLA-4 transendocytosis. a)**. CD80 forms a high avidity dimer-dimer lattice with CTLA-4, resulting in stable binding and increased CTLA-4 ubiquitylation. Ubiquitylation deviates CTLA-4 away from recycling by targeting of the CD80 and CTLA-4 complex to late endosomes/lysosomes marked by Rab7 and LAMP3. **b)**. Interaction of low affinity monomeric CD86 does not modify CTLA-4 with ubiquitin and results in pH-dependent separation of CTLA-4 and CD86. Unmodified CTLA-4 recycles back to the cell surface in an LRBA and Rab11-dependent manner. After detaching from CTLA-4, CD86 is rapidly degraded.

# Reporting Summary

Nature Research wishes to improve the reproducibility of the work that we publish. This form provides structure for consistency and transparency in reporting. For further information on Nature Research policies, see Authors & Referees and the Editorial Policy Checklist.

## Statistics

For all statistical analyses, confirm that the following items are present in the figure legend, table legend, main text, or Methods section.

| n/a | Confirmed | |
|---|---|---|
| ☐ | ☒ | The exact sample size (*n*) for each experimental group/condition, given as a discrete number and unit of measurement |
| ☒ | ☐ | A statement on whether measurements were taken from distinct samples or whether the same sample was measured repeatedly |
| ☐ | ☒ | The statistical test(s) used AND whether they are one- or two-sided<br>*Only common tests should be described solely by name; describe more complex techniques in the Methods section.* |
| ☒ | ☐ | A description of all covariates tested |
| ☐ | ☒ | A description of any assumptions or corrections, such as tests of normality and adjustment for multiple comparisons |
| ☐ | ☒ | A full description of the statistical parameters including central tendency (e.g. means) or other basic estimates (e.g. regression coefficient) AND variation (e.g. standard deviation) or associated estimates of uncertainty (e.g. confidence intervals) |
| ☐ | ☒ | For null hypothesis testing, the test statistic (e.g. *F*, *t*, *r*) with confidence intervals, effect sizes, degrees of freedom and *P* value noted<br>*Give P values as exact values whenever suitable.* |
| ☒ | ☐ | For Bayesian analysis, information on the choice of priors and Markov chain Monte Carlo settings |
| ☒ | ☐ | For hierarchical and complex designs, identification of the appropriate level for tests and full reporting of outcomes |
| ☒ | ☐ | Estimates of effect sizes (e.g. Cohen's *d*, Pearson's *r*), indicating how they were calculated |

*Our web collection on statistics for biologists contains articles on many of the points above.*

## Software and code

Policy information about availability of computer code

| Data collection | BD FACSDiva (BD Biosciences), Nikon elements. |
|---|---|
| Data analysis | FlowJo v10 (BD Biosciences), CellProfiler v2.2, Graphpad Prism |

For manuscripts utilizing custom algorithms or software that are central to the research but not yet described in published literature, software must be made available to editors/reviewers. We strongly encourage code deposition in a community repository (e.g. GitHub). See the Nature Research guidelines for submitting code & software for further information.

## Data

Policy information about availability of data

All manuscripts must include a data availability statement. This statement should provide the following information, where applicable:
- Accession codes, unique identifiers, or web links for publicly available datasets
- A list of figures that have associated raw data
- A description of any restrictions on data availability

The authors declare that the data supporting the findings of this study are available within the paper and its supplementary information files.

# Field-specific reporting

Please select the one below that is the best fit for your research. If you are not sure, read the appropriate sections before making your selection.

☒ Life sciences ☐ Behavioural & social sciences ☐ Ecological, evolutionary & environmental sciences

For a reference copy of the document with all sections, see nature.com/documents/nr-reporting-summary-flat.pdf

# Life sciences study design

All studies must disclose on these points even when the disclosure is negative.

| | |
|---|---|
| Sample size | No statistical methods were used to predetermine sample size. |
| Data exclusions | No data was excluded |
| Replication | Typically experiments were performed 2 or more times to confirm robustness of biological observations. Predictions based on conclusions from these experiments were then tested by further independent experiments. |
| Randomization | No randomisation was used |
| Blinding | Experiments were not blinded as the same operator performed all experimental steps for a given experiment. |

# Behavioural & social sciences study design

All studies must disclose on these points even when the disclosure is negative.

| | |
|---|---|
| Study description | *Briefly describe the study type including whether data are quantitative, qualitative, or mixed-methods (e.g. qualitative cross-sectional, quantitative experimental, mixed-methods case study).* |
| Research sample | *State the research sample (e.g. Harvard university undergraduates, villagers in rural India) and provide relevant demographic information (e.g. age, sex) and indicate whether the sample is representative. Provide a rationale for the study sample chosen. For studies involving existing datasets, please describe the dataset and source.* |
| Sampling strategy | *Describe the sampling procedure (e.g. random, snowball, stratified, convenience). Describe the statistical methods that were used to predetermine sample size OR if no sample-size calculation was performed, describe how sample sizes were chosen and provide a rationale for why these sample sizes are sufficient. For qualitative data, please indicate whether data saturation was considered, and what criteria were used to decide that no further sampling was needed.* |
| Data collection | *Provide details about the data collection procedure, including the instruments or devices used to record the data (e.g. pen and paper, computer, eye tracker, video or audio equipment) whether anyone was present besides the participant(s) and the researcher, and whether the researcher was blind to experimental condition and/or the study hypothesis during data collection.* |
| Timing | *Indicate the start and stop dates of data collection. If there is a gap between collection periods, state the dates for each sample cohort.* |
| Data exclusions | *If no data were excluded from the analyses, state so OR if data were excluded, provide the exact number of exclusions and the rationale behind them, indicating whether exclusion criteria were pre-established.* |
| Non-participation | *State how many participants dropped out/declined participation and the reason(s) given OR provide response rate OR state that no participants dropped out/declined participation.* |
| Randomization | *If participants were not allocated into experimental groups, state so OR describe how participants were allocated to groups, and if allocation was not random, describe how covariates were controlled.* |

# Ecological, evolutionary & environmental sciences study design

All studies must disclose on these points even when the disclosure is negative.

| | |
|---|---|
| Study description | *Briefly describe the study. For quantitative data include treatment factors and interactions, design structure (e.g. factorial, nested, hierarchical), nature and number of experimental units and replicates.* |
| Research sample | *Describe the research sample (e.g. a group of tagged Passer domesticus, all Stenocereus thurberi within Organ Pipe Cactus National Monument), and provide a rationale for the sample choice. When relevant, describe the organism taxa, source, sex, age range and any manipulations. State what population the sample is meant to represent when applicable. For studies involving existing datasets, describe the data and its source.* |
| Sampling strategy | *Note the sampling procedure. Describe the statistical methods that were used to predetermine sample size OR if no sample-size calculation was performed, describe how sample sizes were chosen and provide a rationale for why these sample sizes are sufficient.* |
| Data collection | *Describe the data collection procedure, including who recorded the data and how.* |
| Timing and spatial scale | *Indicate the start and stop dates of data collection, noting the frequency and periodicity of sampling and providing a rationale for these choices. If there is a gap between collection periods, state the dates for each sample cohort. Specify the spatial scale from which the data are taken* |

| Data exclusions | *If no data were excluded from the analyses, state so OR if data were excluded, describe the exclusions and the rationale behind them, indicating whether exclusion criteria were pre-established.* |
| --- | --- |
| Reproducibility | *Describe the measures taken to verify the reproducibility of experimental findings. For each experiment, note whether any attempts to repeat the experiment failed OR state that all attempts to repeat the experiment were successful.* |
| Randomization | *Describe how samples/organisms/participants were allocated into groups. If allocation was not random, describe how covariates were controlled. If this is not relevant to your study, explain why.* |
| Blinding | *Describe the extent of blinding used during data acquisition and analysis. If blinding was not possible, describe why OR explain why blinding was not relevant to your study.* |

Did the study involve field work? ☐ Yes ☐ No

## Field work, collection and transport

| Field conditions | *Describe the study conditions for field work, providing relevant parameters (e.g. temperature, rainfall).* |
| --- | --- |
| Location | *State the location of the sampling or experiment, providing relevant parameters (e.g. latitude and longitude, elevation, water depth).* |
| Access and import/export | *Describe the efforts you have made to access habitats and to collect and import/export your samples in a responsible manner and in compliance with local, national and international laws, noting any permits that were obtained (give the name of the issuing authority, the date of issue, and any identifying information).* |
| Disturbance | *Describe any disturbance caused by the study and how it was minimized.* |

# Reporting for specific materials, systems and methods

We require information from authors about some types of materials, experimental systems and methods used in many studies. Here, indicate whether each material, system or method listed is relevant to your study. If you are not sure if a list item applies to your research, read the appropriate section before selecting a response.

### Materials & experimental systems

| n/a | Involved in the study |
| --- | --- |
| ☐ | ☒ Antibodies |
| ☐ | ☒ Eukaryotic cell lines |
| ☒ | ☐ Palaeontology |
| ☒ | ☐ Animals and other organisms |
| ☒ | ☐ Human research participants |
| ☒ | ☐ Clinical data |

### Methods

| n/a | Involved in the study |
| --- | --- |
| ☒ | ☐ ChIP-seq |
| ☐ | ☒ Flow cytometry |
| ☒ | ☐ MRI-based neuroimaging |

## Antibodies

| Antibodies used | Anti-GFP (clones 7.1 and 13.1, Sigma-Aldrich, 11814460001)- dilution 1/1000<br>Anti-RFP mouse monoclonal {6G6} (Chromotek, AB_2631395)- dilution 1/1000<br>anti-CTLA-4 polyclonal (C-19) (Santa Cruz Biotechnology, sc-1628)- dilution 1/1000<br>anti-CTLA-4 rabbit monoclonal (EPR1476) (Abcam, ab134090)- dilution 1/1000<br>anti-CTLA-4 BNI3 (BD-557301)- dilution 1/50<br>anti-tubulin mouse monoclonal (DMIA) (Sigma-Aldrich, T6199)- dilution 1/1000<br>Mouse monoclonal anti-pan ubiquitin (FK2) (Sigma-Aldrich, ST1200) - dilution 1/1000<br>Anti-LRBA (Atlas Antibodies, HPA019366)- dilution 1/500<br>anti-Rab5 (C8B1) (Cell Signaling Technology, 3547)- dilution 1/250<br> anti-Rab7 (D95F2) (Cell Signaling Technology, 9367)- dilution 1/250<br>horse anti-mouse IgG HRP-linked (Cell Signaling, 7076)- dilution 1/1000<br>anti-goat/sheep IgG HRP-linked (GT-34) (Sigma-Aldrich, A9452) - dilution 1/1000<br>anti-human IgG HRP-linked (Sigma-Aldrich, 634400)<br>Rabbit Anti-Human IgG(H+L)-PE (Southern Biotech, 6140-09) |
| --- | --- |
| Validation | All antibodies were commercially available and validated by manufacturers. In the case of anti- CTLA-4, LRBA, CD80, CD86 antibodies were also validated on transfected cell lines and gene knockouts to ensure specificity. |

# Eukaryotic cell lines

Policy information about cell lines

| | |
|---|---|
| Cell line source(s) | Jurkat  E6 cells (TIB-152) cells were originally from ATCC .<br>CHO cells were originally from ATCC (CCL-61)<br>DG-75 B cells were from the University of Birmingham (available from ATCC- CRL 2625).<br>Hela cells were originally from  ATCC (CRL-1958) |
| Authentication | All cells were authenticated based on FACS phenotyping consistent with known properties of the cells and based on appropriate morphology. CHO cells were negative for human markers prior to transduction. |
| Mycoplasma contamination | Cells were confirmed mycoplasma negative. |
| Commonly misidentified lines<br>(See ICLAC register) | *Name any commonly misidentified cell lines used in the study and provide a rationale for their use.* |

# Palaeontology

| | |
|---|---|
| Specimen provenance | *Provide provenance information for specimens and describe permits that were obtained for the work (including the name of the issuing authority, the date of issue, and any identifying information).* |
| Specimen deposition | *Indicate where the specimens have been deposited to permit free access by other researchers.* |
| Dating methods | *If new dates are provided, describe how they were obtained (e.g. collection, storage, sample pretreatment and measurement), where they were obtained (i.e. lab name), the calibration program and the protocol for quality assurance OR state that no new dates are provided.* |

☐ Tick this box to confirm that the raw and calibrated dates are available in the paper or in Supplementary Information.

# Animals and other organisms

Policy information about studies involving animals; ARRIVE guidelines recommended for reporting animal research

| | |
|---|---|
| Laboratory animals | |
| Wild animals | *Provide details on animals observed in or captured in the field; report species, sex and age where possible. Describe how animals were caught and transported and what happened to captive animals after the study (if killed, explain why and describe method; if released, say where and when) OR state that the study did not involve wild animals.* |
| Field-collected samples | *For laboratory work with field-collected samples, describe all relevant parameters such as housing, maintenance, temperature, photoperiod and end-of-experiment protocol OR state that the study did not involve samples collected from the field.* |
| Ethics oversight | *Identify the organization(s) that approved or provided guidance on the study protocol, OR state that no ethical approval or guidance was required and explain why not.* |

Note that full information on the approval of the study protocol must also be provided in the manuscript.

# Human research participants

Policy information about studies involving human research participants

| | |
|---|---|
| Population characteristics | *Describe the covariate-relevant population characteristics of the human research participants (e.g. age, gender, genotypic information, past and current diagnosis and treatment categories). If you filled out the behavioural & social sciences study design questions and have nothing to add here, write "See above."* |
| Recruitment | *Describe how participants were recruited. Outline any potential self-selection bias or other biases that may be present and how these are likely to impact results.* |
| Ethics oversight | *Identify the organization(s) that approved the study protocol.* |

Note that full information on the approval of the study protocol must also be provided in the manuscript.

# Clinical data

Policy information about clinical studies

All manuscripts should comply with the ICMJE guidelines for publication of clinical research and a completed CONSORT checklist must be included with all submissions.

| | |
|---|---|
| Clinical trial registration | *Provide the trial registration number from ClinicalTrials.gov or an equivalent agency.* |

| Study protocol | *Note where the full trial protocol can be accessed OR if not available, explain why.* |
| Data collection | *Describe the settings and locales of data collection, noting the time periods of recruitment and data collection.* |
| Outcomes | *Describe how you pre-defined primary and secondary outcome measures and how you assessed these measures.* |

# ChIP-seq

## Data deposition

☐ Confirm that both raw and final processed data have been deposited in a public database such as GEO.

☐ Confirm that you have deposited or provided access to graph files (e.g. BED files) for the called peaks.

| Data access links
*May remain private before publication.* | *For "Initial submission" or "Revised version" documents, provide reviewer access links. For your "Final submission" document, provide a link to the deposited data.* |
| Files in database submission | *Provide a list of all files available in the database submission.* |
| Genome browser session
(e.g. UCSC) | *Provide a link to an anonymized genome browser session for "Initial submission" and "Revised version" documents only, to enable peer review. Write "no longer applicable" for "Final submission" documents.* |

## Methodology

| Replicates | *Describe the experimental replicates, specifying number, type and replicate agreement.* |
| Sequencing depth | *Describe the sequencing depth for each experiment, providing the total number of reads, uniquely mapped reads, length of reads and whether they were paired- or single-end.* |
| Antibodies | *Describe the antibodies used for the ChIP-seq experiments; as applicable, provide supplier name, catalog number, clone name, and lot number.* |
| Peak calling parameters | *Specify the command line program and parameters used for read mapping and peak calling, including the ChIP, control and index files used.* |
| Data quality | *Describe the methods used to ensure data quality in full detail, including how many peaks are at FDR 5% and above 5-fold enrichment.* |
| Software | *Describe the software used to collect and analyze the ChIP-seq data. For custom code that has been deposited into a community repository, provide accession details.* |

# Flow Cytometry

## Plots

Confirm that:

☒ The axis labels state the marker and fluorochrome used (e.g. CD4-FITC).

☒ The axis scales are clearly visible. Include numbers along axes only for bottom left plot of group (a 'group' is an analysis of identical markers).

☒ All plots are contour plots with outliers or pseudocolor plots.

☒ A numerical value for number of cells or percentage (with statistics) is provided.

## Methodology

| Sample preparation | Cells were cultured isolated and stained according to standard protocols described in the manuscript. |
| Instrument | BD LSRFortessa (BD Biosciences) |
| Software | BD FACSDiva (BD Biosciences) was used for data collection.
FlowJo v10 (BD Biosciences) was used for manual gating and analysis. |
| Cell population abundance | No cells were sorted during this study. |
| Gating strategy | Transendocytosis gating strategy: FSC/SSC> singlets> Plot CTV vs Cherry/GFP |

☒ Tick this box to confirm that a figure exemplifying the gating strategy is provided in the Supplementary Information.

# Magnetic resonance imaging

## Experimental design

**Design type**
> *Indicate task or resting state; event-related or block design.*

**Design specifications**
> *Specify the number of blocks, trials or experimental units per session and/or subject, and specify the length of each trial or block (if trials are blocked) and interval between trials.*

**Behavioral performance measures**
> *State number and/or type of variables recorded (e.g. correct button press, response time) and what statistics were used to establish that the subjects were performing the task as expected (e.g. mean, range, and/or standard deviation across subjects).*

## Acquisition

**Imaging type(s)**
> *Specify: functional, structural, diffusion, perfusion.*

**Field strength**
> *Specify in Tesla*

**Sequence & imaging parameters**
> *Specify the pulse sequence type (gradient echo, spin echo, etc.), imaging type (EPI, spiral, etc.), field of view, matrix size, slice thickness, orientation and TE/TR/flip angle.*

**Area of acquisition**
> *State whether a whole brain scan was used OR define the area of acquisition, describing how the region was determined.*

**Diffusion MRI** ☐ Used ☐ Not used

## Preprocessing

**Preprocessing software**
> *Provide detail on software version and revision number and on specific parameters (model/functions, brain extraction, segmentation, smoothing kernel size, etc.).*

**Normalization**
> *If data were normalized/standardized, describe the approach(es): specify linear or non-linear and define image types used for transformation OR indicate that data were not normalized and explain rationale for lack of normalization.*

**Normalization template**
> *Describe the template used for normalization/transformation, specifying subject space or group standardized space (e.g. original Talairach, MNI305, ICBM152) OR indicate that the data were not normalized.*

**Noise and artifact removal**
> *Describe your procedure(s) for artifact and structured noise removal, specifying motion parameters, tissue signals and physiological signals (heart rate, respiration).*

**Volume censoring**
> *Define your software and/or method and criteria for volume censoring, and state the extent of such censoring.*

## Statistical modeling & inference

**Model type and settings**
> *Specify type (mass univariate, multivariate, RSA, predictive, etc.) and describe essential details of the model at the first and second levels (e.g. fixed, random or mixed effects; drift or auto-correlation).*

**Effect(s) tested**
> *Define precise effect in terms of the task or stimulus conditions instead of psychological concepts and indicate whether ANOVA or factorial designs were used.*

**Specify type of analysis:** ☐ Whole brain ☐ ROI-based ☐ Both

**Statistic type for inference**
(See Eklund et al. 2016)
> *Specify voxel-wise or cluster-wise and report all relevant parameters for cluster-wise methods.*

**Correction**
> *Describe the type of correction and how it is obtained for multiple comparisons (e.g. FWE, FDR, permutation or Monte Carlo).*

## Models & analysis

| n/a | Involved in the study |
|-----|------------------------|
| ☐ | ☐ Functional and/or effective connectivity |
| ☐ | ☐ Graph analysis |
| ☐ | ☐ Multivariate modeling or predictive analysis |

**Functional and/or effective connectivity**
> *Report the measures of dependence used and the model details (e.g. Pearson correlation, partial correlation, mutual information).*

**Graph analysis**
> *Report the dependent variable and connectivity measure, specifying weighted graph or binarized graph, subject- or group-level, and the global and/or node summaries used (e.g. clustering coefficient, efficiency, etc.).*

Multivariate modeling and predictive analysis

*Specify independent variables, features extraction and dimension reduction, model, training and evaluation metrics.*

