## [Peer Review File · Nature Immunology]

Peer Review Information

Journal: Nature Immunology

Manuscript Title: Differences in CD80 and CD86 transendocytosis reveal CD86 as a key target for CTLA-4 immune regulation

Corresponding author name(s): Professor David Sansom

Editorial Notes:

**Redactions –
unpublished data**

Reviewer Comments & Decisions:

Decision Letter, initial version:

20th Jul 2021

Dear David,

We have now finished reviewing your manuscript entitled "Differences in CD80 and CD86 transendocytosis reveal CD86 as a key target for CTLA-4 immune regulation", reference number NI-A32355.

Although the editors thought that the manuscript was interesting enough to send out for in-depth review, the reviewers were not in favor of publishing the paper in Nature Immunology

We are therefore returning the reviews to you with the hope that you find them useful when you prepare the paper for another journal.

We realize that this is disappointing. I hope that you continue to consider Nature Immunology for your results most significant for the immunology community and wish you well in your future investigations.

Sincerely,

Laurie

Laurie A. Dempsey, Ph.D.
Senior Editor
Nature Immunology
l.dempsey@us.nature.com
ORCID: 0000-0002-3304-796X

Reviewers' comments:

Reviewer #1 (Remarks to the Author):

NI-A32355

Differences in CD80 and CD86 transendocytosis reveal CD86 as a key target for CTLA-4 immune regulation

This paper addresses the intriguing question of the differential fates of CD80 and CD86 when binding to CTLA-4. In brief the paper shows that CTLA-4 transendocytosis of CD86 results in recycling of CTLA-4 to the recipient cell surface while CD80 results in lysosomal degradation of CTLA-4. The paper suggests that, in this way, CD86 effectively controls CTLA-4 availability on the cell surface, while CD80 simply destroys CTLA-4.

This is a data-rich paper with 7 figures and 5 supplemental figures. The first 4 figures focus on differential fates of CTLA-4 after engagement with either CD80 or CD86, showing ubiquitination of CTLA-4 after CD80 (but not 86) engagement (Figure 2), resulting in a difference in intracellular recycling of CTLA-4. The authors suggest, but do not show primary data to support, the idea that CD80 and CD86 traffic via different intracellular compartments after transendocytosis (Figure 3). They then mutate the lysine targets of ubiquitination in CTLA-4 to show that this only affects transendocytosis of CD86 (Figure 4). To examine the functional consequences of differential regulation of CTLA-4 by CD80 v CD86 they make use of a patient mutation (R70) that they say binds CD80 but not CD86 (showing data for CD80 only, Figure 5D, E). Cells expressing R70 CTLA-4 are defective in transendocytosis of CD86 (Figure 6) and failed to trigger T cell proliferation (Figure 7). While this study addresses an interesting question, the very artificial system used and the lack of robust biochemical and cell biological data undermine the conclusions.

Specific comments:

1. The data and methods are mixed and often confusing. For example, the transendocytosis assay in Figure 1 needs quite a bit of thought to work out that CTV is simply labeling donor cells and rather than CD80 and CD86 as the labels next to the y axis infer. Perhaps the authors could label "donor" populations. The gates are rather confusing too as sometimes these appear to pass through the middle of a discrete population of cells (eg Fig 1B, 24h Treg for CD80).
2. The transendocytosis assays are all carried out in different combinations of cells over-expressing CTLA-4, CD80 and CD86, using either CHO cells expressing CTLA-4 and CHO cells expressing CD80/86 or B cells (DG75) expressing CD80/86 and Treg cells over-expressing CTLA-4 (see Figure 1 C and D). In order to understand the biological significance in the immune system these assays need to examine endogenous (rather than overexpressed) protein in the relevant immune cells. The problem with using over-expressed proteins is that over-expression is well known to result in mislocalization. In addition, trafficking pathways in non-immune cells can vary as sorting adapter expression and activation varies between different cells. Why use these very artificial systems to measure transendocytosis?
3. Although the paper suggests that CD80 and 86 drive differential trafficking of CTLA4 (Figures 1C, D and 3B) this data is weak with no primary data shown in 3B where Rab protein co-localization is shown as a bar chart. Given how troublesome antibodies against Rab proteins are known to be, it is essential to show this data. There are many more markers of robust endocytic compartments available. A time course may also be useful as well as knowledge of how signaling and phosphorylation of CTLA4 are affected, as these will affect the efficiency of endocytosis.
4. Figure 2 compares ubiquitination of CTLA-4 after ligand encounter with CD80 v CD86 (all expressed in CHO cells). Mutation of lysine residues prevents ubiquitination of CTLA-4 after interaction with CD80 as interpreted from higher molecular weight bands in Fig 2A and bands seen after IP for Ub in Fig 2D. It is puzzling that the sizes of these higher molecular bands do not match, with the highest signal in 2A below the 37kD marker, while the signal in 2D shows discrete bands of 37kD and ~ 50kD.
5. Fig 2B is used to suggest that CTLA-4 interaction with CD86 does not cause ubiquitination, however there is no Ub IP panel for CD86. Furthermore, Fig 2B shows uneven contrast between bands (apparent upon enlargement of the image) with what looks like a splice between lanes 3 and 4, suggestive of inappropriate processing. It is not clear that the very straight tubulin bands are from the same gel as lanes 8-10 curve right for the CTLA-4 signal. These biochemical experiments cannot be included given these problems.
6. The 'degradation bands' described as an indicator of ubiquitination driven degradation for CD80 are also evident for CD86 in panel 2E using Jurkat recipient cells. No blue box has been drawn around these bands for CD86. Why are the degradation bands seen here?
7. Figure 3A uses proximity ligation of CTLA-4 and ubiquitin to examine co-localization with CD80 and CD86. (I think that the figure should read CTLA4, not 44.) Although there is less co-localization of CD86 with the PLA signal the strong PLA signal suggests that CTLA4 is also ubiquitinated after interaction with CD86. This and the biochemical data showing degradation bands with CD86 (2E) beg the question of whether the difference is a result of decreased intracellular signaling after CD86 interaction with CTLA4 compared to CD80. This could be measured in Jurkat recipient cells and this

simple explanation needs to be explored. This would be consistent with the reduced transendocytosis of CD86 shown in Fig 4.

8. In Figure S5 the authors suggest that R70 mutations in CTLA4 cause CD80 to behave more like CD86. It is not clear what exactly they mean by this as the point of R70 seems to be that it does not interact with CD86. It is important that a Biacore assay showing the difference in binding of CD80 and CD86 is provided in the figures. This patient mutation is clearly interesting and the patient data would provide important insights into the biology.

In summary, using a more physiological assay with appropriate cells (rather than CHO cells) and improving the biochemical data would strengthen this paper. I wonder whether splitting off the R70 patient data would allow the authors to really pin down the questions left unanswered at the moment and describe the insights from the patient in more detail too. One significant difference between the cell systems used here and the patient is that CD28, the alternative ligand for CD80 and 86 is absent from the cell-cell systems used here, while CD28 will be present in the patient.

Reviewer #2 (Remarks to the Author):

In this manuscript, entitled "Differences in CD80 and CD86 transendocytosis reveal CD86 as a key target for CTLA-4 immune regulation," Kennedy et al proposed a very intriguing model in which, while both CD80 and CD86 are subjected to CTLA-4-mediated transendocytosis, each ligand triggers distinct molecular fates for CTLA-4 upon internalization, subsequently leading to different immune regulatory outcomes. Specifically, while CTLA-4 binds to CD80 with stronger affinity, the stronger interaction was also more stable after internalization into lysosomes, which promoted the ubiquitination of CTLA-4 and termination of CTLA-4 signals. On the contrary, CD86 lacks such affinity, thus allowing CTLA-4 to be recycled back to the cell surface. Thus, CTLA-4 engagement of CD80 versus CD86 may lead to very different expression kinetics for CTLA-4.

The authors first utilized their previously established, flow cytometry-based approach to track and compare the transendocytosis of fluorescently tagged CD80 or CD86, and found that the CD86:CTLA-4 interaction presented a different transendocytosis fluorescence pattern than the CD80 interaction. Microscopy further confirmed the difference, and laid the foundation for the authors' working model that the different affinities between CD80:CTLA-4 and CD86:CTLA-4 may directly affect signaling by altering the protein half-life of CTLA-4. Using a series of molecular studies, the authors further revealed the differential molecular fates of CTLA-4 after engagement with CD80 versus CD86. The former, but not the latter, led to CTLA-4 degradation through ubiquitination. Taking advantage of biochemical approaches, the authors proposed that the drop in pH inside lysosomes may function as a driving force to first disrupt the relatively weaker interaction between CTLA-4 and CD86, whereas the stronger affinity of CD80 for CTLA-4 made the interaction more resistant to the pH change. Last, the authors used mutagenesis approaches to modulate the lysosome trafficking of CTLA-4 and showed that CD86 transendocytosis is relied upon more heavily for CTLA-4 recycling than CD80 transendocytosis. Furthermore, disruption of CTLA-4 recycling via imposition of mutations identified in human autoimmune patients specifically affected CD86-driven T cell function.

Overall, this study is well designed and well thought-out, and proposed a very interesting perspective that invites future follow-up studies in immunology. These new data were of great interest, especially as recent studies have shown that CD80 and PD-L1 on antigen-presenting cells (APCs) can also interact with each other in cis and form CD80–PD-L1 heterodimers to restrict CTLA-4 and PD-1 inhibitory signaling. This newly added layer of interaction would provide a more complete picture of the unique roles of CD80 and CD86. Future studies would be warranted to further connect these differences to T cell behavior in the immunological challenge setting. I have only a few suggestions that may hopefully strengthen the manuscript:

- 1) The authors did not comment on the CD80- versus CD86-driven biological effects on and consequences for CD28 signaling until the discussion. It may be worth introducing these dynamics earlier in the manuscript to give readers a more complete picture and allow them to put every detail into context.
- 2) It may be helpful to illustrate the expression pattern of CD80/CD86 in a supplementary figure. Even some data points from the Immgen database would be helpful.
- 3) In Figure 1A/B, the authors focused on the graded/shaded quadrants of the flow plots. However, in Fig 1A, it might also be helpful to bring the upper quadrants to the readers' attention. In the upper quadrants, the loss of fluorescence in the donor cells suggests that the fluorescently tagged CD80/CD86 was acquired by CTLA-4, and the losses for both ligands were comparable; however, that did not lead to comparable gains of fluorescent signals in the CTLA-4-expressing (recipient) cells. These findings suggest that different cellular signaling pathways may be involved in CD80 versus CD86 transendocytosis. Adding one or two sentences to describe the data may set the stage for the next section where the differences in cellular signaling components are highlighted.
- 4) In the same section, the authors suggested the data are recapitulated in Jurkat cells (Figure S1B), but the numbers in the quadrants of the CD86 data didn't seem to strongly argue the same message. I think there is a slight change if we compare the ratio of bottom left:bottom right in the CTLA-4 column versus the same ratio in the + Bafilomycin column. It could be worth including a bar graph showing the relevant ratios to reinforce the point.
- 5) It might be interesting if the authors can devote some discussion to whether/how the low pH in certain microenvironments (i.e., the acidic niche in lymph nodes, the low-pH environment in certain tumors) may also influence the interplay among CD28/CTLA4 signals by differentially affecting CD86/CD80 ligand binding to CTLA-4.
- 6) In Figure 5B, I appreciate the authors' effort to show the effects of the R70Q/W mutants. The authors showed the A86V mutant in Figure 5B but didn't follow up with further studies as they did with the R70Q/W mutants, given that the mutation did not dramatically change the ligand binding. However, I think the A86V mutant may actually be interesting to follow up because 1) it didn't completely abolish the interaction with CD86 as did R70Q/W, and 2) it seems to me that the CD80-Ig staining signal for A86V mutants is weaker than that for R70Q/W mutants, suggesting that it may have a weaker ability than the R70Q/W mutant to interact with CD80. I wonder if the authors would speculate that A86V may potentially cause CD80 to behave like CD86.
- 7) In Figure S3B, the unstained data appear to be identical in all three plots. The panels were labeled to suggest that the data should present unstained controls for three different types of cells (WT, Kless,

and KO). Please confirm that the correct histograms for the unstained data are included.

Reviewer #3 (Remarks to the Author):

In this manuscript, Sansom and colleagues use different systems of overexpression of tagged CD80 and CD86 to show, using transfected non-T cells and the cell line Jurkat, that CTLA-4 captures both ligands from the membrane of artificial antigen presenting cells and that the intracellular fate of the taken CD80 and CD86 is different. The repercussion of such differential fate of CD80 and CD86 on the immune response is marginally assessed by analysing the impact of CTLA-4 mutations found in human patients on the capacity of CD25+CD4+ human Treg to uptake CD80 and CD86. In an attempt to explain the repercussions of differential CD80 and CD86 transendocytosis by CTLA-4 mutants, the authors finally carry out an experiment of depletion of those CD28 ligands from a B cell line to assess the effect of such depletion on costimulation of T cells stimulated to proliferate with anti-CD3 antibodies.

The systems used throughout the manuscript are far of physiological and from the point of view of an immunologist, the question about what is the relevance of those findings is still unanswered. Optimally, the effect of differential fate of CD80 and CD86 would need to be studied in an animal model in the context of the entire immune system but if that was not possible, at least strategies to measure the impact of mutations in primary T cells in response to antigen, in the differentiation of CD4 T cells and in the suppressor role of Treg should be a minimum. Without these more physiological approaches, one questions the relevance of the data and if all observations are just due to overexpression.

Author Rebuttal to Initial comments

NI-A32355 response to reviewers.

Kennedy et.al.,

Differences in CD80 and CD86 transendocytosis reveal CD86 as a key target for CTLA-4 immune regulation.

Reviewer#1 (Remarks to the Author):

NI-A32355

Differences in CD80 and CD86 transendocytosis reveal CD86 as a key target for CTLA-4 immune regulation

This paper addresses the intriguing question of the differential fates of CD80 and CD86 when binding to CTLA-4. In brief the paper shows that CTLA-4 transendocytosis of CD86 results in recycling of CTLA-4 to the recipient cell surface while CD80 results in lysosomal degradation of CTLA-4. The paper suggests that, in this way, CD86 effectively controls CTLA-4 availability on the cell surface, while CD80 simply destroys CTLA-4.

This is a data-rich paper with 7 figures and 5 supplemental figures. The first 4 figures focus on differential fates of CTLA-4 after engagement with either CD80 or CD86, showing ubiquitination of CTLA-4 after CD80 (but not 86) engagement (Figure 2), resulting in a difference in intracellular recycling of CTLA-4. The authors suggest, but do not show primary data to support, the idea that CD80 and CD86 traffic via different intracellular compartments after transendocytosis (Figure 3). They then mutate the lysine targets of ubiquitination in CTLA-4 to show that this only affects transendocytosis of CD86 (Figure 4). To examine the functional consequences of differential regulation of CTLA-4 by CD80 v CD86 they make use of a patient mutation (R70) that they say binds CD80 but not CD86 (showing data for CD80 only, Figure 5D, E). Cells expressing R70 CTLA-4 are defective in transendocytosis of CD86 (Figure 6) and failed to trigger T cell proliferation (Figure 7). While this study addresses an interesting question, the very artificial system used and the lack of robust biochemical and cell biological data undermine the conclusions.

1. The data and methods are mixed and often confusing. For example, the transendocytosis assay in Figure 1 needs quite a bit of thought to work out that CTV is simply labeling donor cells and rather than CD80 and CD86 as the labels next to the y axis infer. Perhaps the authors could label "donor" populations. The gates are rather confusing too as sometimes these appear to pass through the middle of a discrete population of cells (eg Fig 1B, 24h Treg for CD80).

We have re-labelled all our transendocytosis flow cytometry plots to try to make them more intuitive. We have color coded CD80 and CD86 and moved the labels to avoid the axis confusion. Combined with figure S2 we feel this provides a clear layout for understanding these assays. We have confirmed our quadrants are accurate and are set based on the relevant no TE controls. As the cells lose or gain ligands via TE they will move out of these control gates. As such the quadrant positions show the starting point for donor and recipient cells and provide a reference for the reader as to how TE has progressed.

2. The transendocytosis assays are all carried out in different combinations of cells over-expressing CTLA-4, CD80 and CD86, using either CHO cells expressing CTLA-4 and CHO cells expressing CD80/86 or B cells (DG75) expressing CD80/86 and Treg cells over-expressing CTLA-4 (see Figure 1 C and D). In order to understand the biological significance in the immune system these assays need to examine endogenous (rather than overexpressed) protein in the relevant immune cells. The problem with using over-expressed proteins is that over-expression is well known to result in mislocalization. In addition, trafficking pathways in non-immune cells can vary as sorting adapter expression and activation varies between different cells. Why use these very artificial systems to measure transendocytosis?

In the manuscript we make use of several different cellular systems, for clear reasons. CHO cells provide an initial expression platform for assessing behaviour of ligands and CTLA-4. They have significant advantages when it comes to looking at vesicle behaviour because they have a large and spread cytoplasm,

allowing us to make clear observations such as those in figure 1 (e.g co-localisation of CTLA-4 and ligands after TE) accurately. The Jurkat cell system is well established as a relevant T cell line that allows us to make convenient genetic manipulations in a T cell setting (e.g. evaluation of Kless, LRBA knockout etc.). Importantly the key features we describe in the manuscript are highly consistent between cell lines suggesting that ectopic expression is not a major issue and our observations are robust across different cell types. **However, we then further validate the key ideas we propose in primary human Treg, where CTLA-4 is not over expressed. These data are entirely consistent with the data from our model systems suggesting there are no artifacts that relate to expression in our work.** We have now made it clearer throughout the manuscript that all the key data are replicated using non-overexpressing, physiologically relevant systems. Importantly, all the data from the different systems support each other, demonstrating a high degree of robustness of the key ideas.

3. Although the paper suggests that CD80 and 86 drive differential trafficking of CTLA4 (Figures 1C, D and 3B) this data is weak with no primary data shown in 3B where Rab protein co-localization is shown as a bar chart. Given how troublesome antibodies against Rab proteins are known to be, it is essential to show this data. There are many more markers of robust endocytic compartments available. A time course may also be useful as well as knowledge of how signaling and phosphorylation of CTLA4 are affected, as these will affect the efficiency of endocytosis.

We have now included examples of Rab stains in figure S4 demonstrating the discrete punctate stains as expected for Rab proteins, addressing the referee's concerns about antibody specificity. In addition to this data arguing that CTLA-4 and ligand complexes are associated with differential trafficking, we also directly manipulate Rab 11 and LRBA, which show differential effects on CD80 and CD86 TE reinforcing this point. Further evidence of these differences is also found in figure 1 (e.g NH₄Cl sensitivity, co-localisation) and in our PLA studies. When taken together it is hard to suggest that the data supporting differential trafficking are weak.

A time course may also be useful as well as knowledge of how signaling and phosphorylation of CTLA4 are affected, as these will affect the efficiency of endocytosis.

The concepts behind CTLA-4 signalling, phosphorylation and endocytosis are still controversial and in our experience these experiments are not reproducible^{1, 2}. It is therefore not feasible for us to pursue such experiments.

4. Figure 2 compares ubiquitination of CTLA-4 after ligand encounter with CD80 v CD86 (all expressed in CHO cells). This is not the case. The ubiquitin signals are also analysed in Jurkat cells (Fig. 2E) and primary Treg (Fig. 2F), with consistent results in all 3 systems.

Mutation of lysine residues prevents ubiquitination of CTLA-4 after interaction with CD80 as interpreted from higher molecular weight bands in Fig 2A and bands seen after IP for Ub in Fig 2D. It is puzzling that the sizes of these higher molecular bands do not match, with the highest signal in 2A below the 37kD marker, while the signal in 2D shows discrete bands of 37kD and ~ 50kD .

We disagree and see no issue with the band sizing. In the total WB (Fig. 2A/B) the Ubiquitinated bands are faint as expected when a large fraction of CTLA-4 is not ubiquitinated –Nonetheless there is still evidence of an upward smear that is not present for CD86 or for Kless. In the Ubq IPs, these bands are enriched as would be expected making clear bands appear above the 37Kd marker, but all consistent in size overall. Indeed in figure 3C we clearly observe in the Ubq blot that Ubiquitinated bands appear only above the position of CTLA-4 (~25-30kd), which is the same location as the smearing seen after CD80 engagement. Across multiple independent experiments the fact that CTLA-4 is modified by ubiquitin in a manner dependent on cytoplasmic lysine residues, seems beyond doubt.

5. Fig 2B is used to suggest that CTLA-4 interaction with CD86 does not cause ubiquitination, however there is no Ub IP panel for CD86. Ubq IP's are shown for CD86 in panels 2E and 2F (in physiologically relevant cell types) and where CTLA-4 is not overexpressed (2F). The data are in complete agreement with our stated position that CD86 does not drive CTLA-4 Ubiquitination.

Furthermore, Fig 2B shows uneven contrast between bands (apparent upon enlargement of the image) with what looks like a splice between lanes 3 and 4, suggestive of inappropriate processing. It is not clear that the very straight tubulin bands are from the same gel as lanes 8-10 curve right for the CTLA-4 signal. These biochemical experiments cannot be included given these problems.

[REDACTED]. We have now replaced fig 2A and 2B with different examples of the same experiment in the hope that whatever it was that the referee found disturbing is no longer a problem.

6. The 'degradation bands' described as an indicator of ubiquitination driven degradation for CD80 are also evident for CD86 in panel 2E using Jurkat recipient cells. No blue box has been drawn around these bands for CD86. Why are the degradation bands seen here?

It is known that CTLA-4 is constitutively turned over and has a relatively short half life in the absence of ligation. Accordingly, there is always background CTLA-4 degradation (both ubiquitin driven and not) and we see these as low Mw fragments of CTLA-4 on our blots. In general when we ligate with CD80 we see

increases in these fragments and they are inhibited by bafilomycin treatment, however they are not exclusively generated by CD80 ligation and therefore we also observe them (albeit typically at lower levels) in CD86 experiments, hence their presence in our blots. We have reworked the text to make this background issue clearer.

7. Figure 3A uses proximity ligation of CTLA-4 and ubiquitin to examine co-localization with CD80 and CD86. (I think that the figure should read CTLA4, not 44.) Although there is less co-localization of CD86 with the PLA signal the strong PLA signal suggests that CTLA4 is also ubiquitinated after interaction with CD86. This and the biochemical data showing degradation bands with CD86 (2E) beg the question of whether the difference is a result of decreased intracellular signaling after CD86 interaction with CTLA4 compared to CD80. This could be measured in Jurkat recipient cells and this simple explanation needs to be explored. This would be consistent with the reduced transendocytosis of CD86 shown in Fig 4.

We have now corrected the typo. We have also tried to make the point clearer in the text that there is a background of CTLA-4 ubiquitination even in the absence of ligation, which is then enhanced by CD80 ligation. We have now split figure 3A to show more obviously the increase in CTLA-4 ubiquitination caused by CD80 (number of PLA puncta per cell) compared with CD86. In addition the significant colocalisation between CD80 and ubiquitinated CTLA-4 (% ligand and PLA co-localisation) is consistent with our hypothesis that CD80 drives CTLA-4 ubiquitination and that CD80 remains bound in this setting. Whilst it is possible to hypothesise that CD86 might cause decreased “CTLA-4 signalling” compared to CD80 (based on weaker affinity), a simple quantitative defect would not explain the fact that ubiquitinated CTLA-4 does not associate with CD86 or that CD86- in association with CTLA-4 is found in different cellular locations to CD80. Given that there is no agreement as to CTLA-4 signals¹ that are driven by ligands it is difficult to test the “simple explanation” suggested above.

8. In Figure S5 the authors suggest that R70 mutations in CTLA4 cause CD80 to behave more like CD86. It is not clear what exactly they mean by this as the point of R70 seems to be that it does not interact with CD86. It is important that a Biacore assay showing the difference in binding of CD80 and CD86 is provided in the figures. This patient mutation is clearly interesting and the patient data would provide important insights into the biology. We have re- worded the text to try and make our point clearer in regard to the previous figure S5 (now Fig. S7). We have added a new description of the R70-CD80 interaction and specify examples of why this is now more like a CD86-CTLA-4 interaction. For example, there is less colocalisation between R70Q-CTLA-4 and CD80 after TE and there is now increased pH sensitivity between R70Q and CD80 etc- these are features typical of CD86 interactions. In relation to the Biacore, we have generated increased quantities of purified proteins and re-performed the biacore experiments, in order to try and detect the weak binding of R70Q to CD86, which was previously not detectable. **These new data are presented in figure 5D and E** and now show a Kd of ~20µM for R70Q-CD86. This allows us to conclude that CD86 binding to R70Q is some 10-fold weaker

than WT interactions. Moreover it demonstrates that a monomeric affinity of 20 μ M is insufficient to support CTLA-4 TE.

In summary, using a more physiological assay with appropriate cells (rather than CHO cells) and improving the biochemical data would strengthen this paper.

Once again we reiterate that our CHO data provide initial characterisation. The key concepts are then reproduced and extended in a Jurkat T cell lines. Finally we also **knock-in the R70Q allele into human Treg and show the CD86-defective phenotype** is entirely consistent with the other systems. **This experiment is based on expression from the endogenous CTLA-4 promoter, in the normal, physiologically relevant cell type and in direct comparison to the WT allele in the same assay.** The data are clear, robust and physiologically relevant.

I wonder whether splitting off the R70 patient data would allow the authors to really pin down the questions left unanswered at the moment and describe the insights from the patient in more detail too.

The patient data has been reported elsewhere³. However, it should be noted that patient phenotypes are notoriously variable and it has not been possible to correlate disease phenotypes (e.g. different autoimmune conditions and different severity) with any features of the different CTLA-4 mutations. This is true even within families, where exactly the same mutation has markedly different presentations⁴. Moreover, the patients are heterozygous and have a confounding WT allele making any molecular analysis difficult. Given these issues it is not clear how we might present this in a cogent way. However, what we do provide is a precise and robust molecular characterisation of the problem underlying a patient-derived CTLA-4 mutation where we show clearly how it affects ligand interactions and TE.

One significant difference between the cell systems used here and the patient is that CD28, the alternative ligand for CD80 and 86 is absent from the cell-cell systems used here, while CD28 will be present in the patient.

This is incorrect. CD28 is present in both Jurkat and Treg systems we present and does not alter any of the key concepts presented.

Reviewer #2 (Remarks to the Author):

In this manuscript, entitled "Differences in CD80 and CD86 transendocytosis reveal CD86 as a key target for CTLA-4

immune regulation,” Kennedy et al proposed a very intriguing model in which, while both CD80 and CD86 are subjected to CTLA-4-mediated transendocytosis, each ligand triggers distinct molecular fates for CTLA-4 upon internalization, subsequently leading to different immune regulatory outcomes. Specifically, while CTLA-4 binds to CD80 with stronger affinity, the stronger interaction was also more stable after internalization into lysosomes, which promoted the ubiquitination of CTLA-4 and termination of CTLA-4 signals. On the contrary, CD86 lacks such affinity, thus allowing CTLA-4 to be recycled back to the cell surface. Thus, CTLA-4 engagement of CD80 versus CD86 may lead to very different expression kinetics for CTLA-4.

The authors first utilized their previously established, flow cytometry-based approach to track and compare the transendocytosis of fluorescently tagged CD80 or CD86, and found that the CD86:CTLA-4 interaction presented a different transendocytosis fluorescence pattern than the CD80 interaction. Microscopy further confirmed the difference, and laid the foundation for the authors’ working model that the different affinities between CD80:CTLA-4 and CD86:CTLA-4 may directly affect signaling by altering the protein half-life of CTLA-4. Using a series of molecular studies, the authors further revealed the differential molecular fates of CTLA-4 after engagement with CD80 versus CD86. The former, but not the latter, led to CTLA-4 degradation through ubiquitination. Taking advantage of biochemical approaches, the authors proposed that the drop in pH inside lysosomes may function as a driving force to first disrupt the relatively weaker interaction between CTLA-4 and CD86, whereas the stronger affinity of CD80 for CTLA-4 made the interaction more resistant to the pH change. Last, the authors used mutagenesis approaches to modulate the lysosome trafficking of CTLA-4 and showed that CD86 transendocytosis is relied upon more heavily for CTLA-4 recycling than CD80 transendocytosis. Furthermore, disruption of CTLA-4 recycling via imposition of mutations identified in human autoimmune patients specifically affected CD86-driven T cell function.

Overall, this study is well designed and well thought-out, and proposed a very interesting perspective that invites future follow-up studies in immunology. These new data were of great interest, especially as recent studies have shown that CD80 and PD-L1 on antigen-presenting cells (APCs) can also interact with each other in cis and form CD80–PD-L1 heterodimers to restrict CTLA-4 and PD-1 inhibitory signaling. This newly added layer of interaction would provide a more complete picture of the unique roles of CD80 and CD86. Future studies would be warranted to further connect these differences to T cell behavior in the immunological challenge setting. I have only a few suggestions that may hopefully strengthen the manuscript:

We very much appreciate the referees’ comments and analysis of our work. As they correctly observe, our work combined with the recent PD-L1–CD80 interaction data very much illustrate the fact that CD80 is a distinct immune regulator. These new concepts together invite the testing of many new immunological ideas.

1) The authors did not comment on the CD80- versus CD86-driven biological effects on and consequences for CD28 signaling until the discussion. It may be worth introducing these dynamics earlier in the manuscript to give readers a more complete picture and allow them to put every detail into context.

2) It may be helpful to illustrate the expression pattern of CD80/CD86 in a supplementary figure. Even some data points from the Immgen database would be helpful.

We have added a section to the introduction to try to provide some more perspective on ligand and CD28 interactions, albeit briefly. In addition we have extended the discussion to try and cover differences between CD80 and CD86, placing this in the context of CD80 or CD86 knockouts. In response to point 2, we have added a **new figure S1** to bring together some of the publically available data on CD80 and CD86 gene expression in immune cells. We hope this provides more perspective on how CD80 and CD86 expression varies across a variety of different immune cell types. The fact that they are often differentially expressed aligns with the concept that they have different functions as presented here. Certainly it has been useful to pull this data together for the first time.

*3) In Figure 1A/B, the authors focused on the graded/shaded quadrants of the flow plots. However, in Fig 1A, it might also be helpful to bring the upper quadrants to the readers' attention. In the upper quadrants, the loss of fluorescence in the donor cells suggests that the fluorescently tagged CD80/CD86 was acquired by CTLA-4, and the **losses for both ligands were comparable; however, that did not lead to comparable gains of fluorescent signals in the CTLA-4-expressing (recipient) cells**. These findings suggest that different cellular signaling pathways may be involved in CD80 versus CD86 transendocytosis. Adding one or two sentences to describe the data may set the stage for the next section where the differences in cellular signaling components are highlighted.*

We have tried to make this clearer in the text and now highlight that the upper quadrants reflect loss of ligand from the donor cells. We indicate, as suggested, that the disconnet between donor loss and recipient gain indicates CD80 and CD86 behave differently.

4) In the same section, the authors suggested the data are recapitulated in Jurkat cells (Figure S1B), but the numbers in the quadrants of the CD86 data didn't seem to strongly argue the same message. I think there is a slight change if we compare the ratio of bottom left:bottom right in the CTLA-4 column versus the same ratio in the + Bafilomycin column. It could be worth including a bar graph showing the relevant ratios to reinforce the point.

We have replaced Fig.S1B with a new panel (S2B in the new manuscript) that better illustrates the increased impact of Baf A on CD86 detection in Jurkat TE experiments. We have also provided a graph as suggested that hopefully reinforces this message. However, across the CHO, Jurkat and Treg systems we are confident in the observation that CD86 detection following in TE is more pH sensitive.

5) It might be interesting if the authors can devote some discussion to whether/how the low pH in certain microenvironments (i.e., the acidic niche in lymph nodes, the low-pH environment in certain tumors) may also influence the interplay among CD28/CTLA4 signals by differentially affecting CD86/CD80 ligand binding to CTLA-4. This is an interesting point, which we have now raised in the discussion.

*6) In Figure 5B, I appreciate the authors' effort to show the effects of the R70Q/W mutants. The authors showed the A86V mutant in Figure 5B but didn't follow up with further studies as they did with the R70Q/W mutants, given that the mutation did not dramatically change the ligand binding. However, I think the A86V mutant may actually be interesting to follow up because 1) it didn't completely abolish the interaction with CD86 as did R70Q/W, and 2) it seems to me that the **CD80-Ig staining signal for A86V mutants is weaker than that for R70Q/W mutants**, suggesting that it may*

have a weaker ability than the R70Q/W mutant to interact with CD80. I wonder if the authors would speculate that A86V may potentially cause CD80 to behave like CD86.

We appreciate the referees thoughts and have now commented on A86V more fully in the results and in the discussion. We have tried a number of different experiments with A86V none of which reveal a robust phenotype that we are confident in. It is our conclusion that A86V may be a variant that is not necessarily pathogenic, despite it being reported in a patient with some autoimmune features. So far we have functionally tested a large number of CTLA-4 patient mutations (>20) in the lab. In those where we see clearly identifiable defects (mis-folding, mis-localisation, loss of ligand binding etc), we do not find these mutations in population databases such as GnomAD (which aggregates over 250K exome and genome sequences). In contrast, for two different patient-associated mutations where we can find no obvious defects (A86V and G109E) both are found in GnomAD (some 7 and 71 times respectively) suggesting they occur at a low but detectable frequency in the general population. Thus at present we consider A86V to be a relatively rare variant, but one that is unlikely to be pathogenic.

7) In Figure S3B, the unstained data appear to be identical in all three plots. The panels were labeled to suggest that the data should present unstained controls for three different types of cells (WT, Kless, and KO). Please confirm that the correct histograms for the unstained data are included.

We thank the referee for spotting this. In fact the controls are Jurkat cells that don't express CTLA-4, which have been stained for CTLA-4 expression. This control is used for all 3 cell lines shown and is therefore very similar between cell lines. We have now changed the figure labelling to reflect this, although the interpretation of the data is not unaffected.

Reviewer #3 (Remarks to the Author):

In this manuscript, Sansom and colleagues use different systems of overexpression of tagged CD80 and CD86 to show, using transfected non-T cells and the cell line Jurkat, that CTLA-4 captures both ligands from the membrane of artificial antigen presenting cells and that the intracellular fate of the taken CD80 and CD86 is different. The repercussion of such differential fate of CD80 and CD86 on the immune response is marginally assessed by analysing the impact of CTLA-4 mutations found in human patients on the capacity of CD25+CD4+ human Treg to uptake CD80 and CD86. In an attempt to explain the repercussions of differential CD80 and CD86 transendocytosis by CTLA-4 mutants, the authors finally carry out an experiment of depletion of those CD28 ligands from a B cell line to assess the effect of such depletion on costimulation of T cells stimulated to proliferate with anti-CD3 antibodies. The systems used throughout the manuscript are far from physiological and from the point of view of an immunologist, the question about what is the relevance of those findings is still unanswered. Optimally, the effect of differential fate of CD80 and CD86 would need to be studied in an animal model in the context of the entire immune system but if that was not possible, at least strategies to measure the impact of mutations in primary T cells in response to antigen, in

the differentiation of CD4 T cells and in the suppressor role of Treg should be a minimum. Without these more physiological approaches, one questions the relevance of the data and if all observations are just due to overexpression.

Our manuscript dissects a disease phenotype seen in two independent human conditions (CTLA-4 and LRBA mutations that lead to autoimmunity)^{5,6,7}. This is by definition based on what happens "in the context of the entire immune system". In the process we describe how CD80 and CD86 differ and dissect in some detail how their molecular cell biology is affected by the above clinical mutations. We also re-validate some of our ideas using CRISPR in primary human Treg to express CTLA-4 mutants within their endogenous locus. This is not overexpression and clear evidence of this is within the paper. The comment that "all observations are just due to overexpression" simply cannot be justified.

Whilst the reviewer may feel that the only relevant (or manipulable) physiology is in a mouse, we would point out that it is not yet clear that the affinity of the interactions between CD80, CD86, CD28 and CTLA-4 are completely identical across species. Similarly the role of pH may differ because this is dependent on both affinity and sequence of the interacting partners. Furthermore, there is no evidence that the altered binding of R70Q to CD80 and CD86 would be equivalent in mice, making the direct transfer of ideas non-viable. Finally, mouse LRBA phenotypes do not overlap with human suggesting other differences in trafficking. Thus there is much to consider before taking *in vivo* approaches and we feel these are clearly beyond the scope of the present work.

We appreciate that we have not addressed in detail the functional consequences of the differences we report, using classical *in vivo* immunology experiments. Nonetheless, the concepts raised in our work are robust, important and relevant to immunology. We show clearly defined differences between CD80 and CD86 on CTLA-4 biology for the first time and discuss the implications of these. We believe the resulting ideas are important to immunology in a range of areas (e.g. autoimmunity, cancer therapy etc.) and that our findings deserve a broad immunology audience so they can then be further tested by other immunologists equipped to do so.

References

1. Walker, L.S. & Sansom, D.M. Confusing signals: Recent progress in CTLA-4 biology. *Trends in immunology* **36**, 63-70 (2015).

2. Qureshi, O.S. *et al.* Constitutive clathrin-mediated endocytosis of CTLA-4 persists during T cell activation. *J Biol Chem* **287**, 9429-9440 (2012).
3. Schwab, C. *et al.* Phenotype, penetrance, and treatment of 133 cytotoxic T-lymphocyte antigen 4-insufficient subjects. *J Allergy Clin Immunol* **142**, 1932-1946 (2018).
4. Hou, T.Z. *et al.* Study of an extended family with CTLA-4 deficiency suggests a CD28/CTLA-4 independent mechanism responsible for differences in disease manifestations and severity. *Clin Immunol* **188**, 94-102 (2018).
5. Schubert, D. *et al.* Autosomal dominant immune dysregulation syndrome in humans with CTLA4 mutations. *Nat Med* **20**, 1410-1416 (2014).
6. Lo, B. *et al.* Patients with LRBA deficiency show CTLA4 loss and immune dysregulation responsive to abatacept therapy. *Science* **349**, 436-440 (2015).
7. Lo, B. *et al.* CHAI and LATAIE: new genetic diseases of CTLA-4 checkpoint insufficiency. *Blood* **128**, 1037-1042 (2016).

Decision Letter, first revision:

31st Oct 2021

Dear David,

Thank you for providing your point-by-point response to the referees' comments on your revised manuscript entitled "Differences in CD80 and CD86 transendocytosis reveal CD86 as a key target for CTLA-4 immune regulation". As noted earlier, the original referee #2 was positive about the study, but the other two referees less so. As noted previously in our e-mail correspondence after review of the first version of the manuscript, many ambiguities - mainly from the lack of precision in describing the experiments and their results, unfortunately, has led to misunderstandings as to how the conclusions are being drawn from the data shown. Hence I suggest a thorough re-write of the manuscript is necessary. I have taken the unusual step at this stage of manuscript review to perform a partial line-by-line edit of the current manuscript in an exercise to show examples of how to clarify the description of the results. (Please see the attached Word document where I have line-edited the Abstract, Introduction and first two Results section). Clarity and accuracy are needed in describing the experimental logic to set up the experiments, describing the experimental data obtained, then whatever conclusions that can be drawn from those data. I am also attached an annotated version of your point-by-point response. I will send a marked-up version of the manuscript PDF in a separate message, as the file size might be too large here.

We invite you to submit a substantially revised manuscript, however please bear in mind that we will

be reluctant to approach the referees again in the absence of major revisions.

Specifically, the revision should include new experiments to address:

- (1) Ref 4 point 5 - Stain for CLTA-4 after CD80/86 engagement in Treg cells.
- (2) Ref 4 point 6 - Perform transendocytosis experiment showing time course (with earlier time points) of GFP loss on donor cells and accumulation in recipient cells (ideally in primary Treg cells).
- (3) Ref 4 point 7 - enumerate proliferation data in Fig. 7 and show summary graph that includes data from multiple independent experiments. Similar requests was made by Ref 5 and editor.

Please include the additional textual clarifications as indicated in your response letter.

When you revise your manuscript, please take into account all reviewer and editor comments, please highlight all changes in the manuscript text file in Microsoft Word format.

- * Include a "Response to referees" document detailing, point-by-point, how you addressed each referee comment. If no action was taken to address a point, you must provide a compelling argument. This response will be sent back to the referees along with the revised manuscript.
- * If you have not done so already please begin to revise your manuscript so that it conforms to our Article format instructions at <http://www.nature.com/ni/authors/index.html>. Refer also to any guidelines provided in this letter.
- * Include a revised version of any required reporting checklist. It will be available to referees (and, potentially, statisticians) to aid in their evaluation if the manuscript goes back for peer review. A revised checklist is essential for re-review of the paper.

The Reporting Summary can be found here:

[Redacted]

If you wish to submit a suitably revised manuscript we would hope to receive it within 6 months. If you cannot send it within this time, please let us know. We will be happy to consider your revision so long as nothing similar has been accepted for publication at Nature Immunology or published elsewhere.

Nature Immunology is committed to improving transparency in authorship. As part of our efforts in this direction, we are now requesting that all authors identified as 'corresponding author' on published papers create and link their Open Researcher and Contributor Identifier (ORCID) with their account on the Manuscript Tracking System (MTS), prior to acceptance. ORCID helps the scientific community achieve unambiguous attribution of all scholarly contributions. You can create and link your ORCID from the home page of the MTS by clicking on 'Modify my Springer Nature account'. For more information please visit www.springernature.com/orcid.

Thank you for the opportunity to review your work.

Kind regards,

Laurie

Laurie A. Dempsey, Ph.D.
Senior Editor
Nature Immunology
l.dempsey@us.nature.com
ORCID: 0000-0002-3304-796X

Reviewers' Comments:

Reviewer #2:

Remarks to the Author:

In this revised manuscript by Kennedy et al., entitled "Differences in CD80 and CD86 transendocytosis reveal CD86 as a key target for CTLA-4 immune regulation," my questions were appropriately addressed. I thank the authors for their thoughtful responses. Although the revised sections of text were not highlighted, which made it difficult to track what had been changed, I did get the impression that the story (especially the Introduction/Results sections) flows better than in the previous version. My enthusiasm for the manuscript remains. I have only a few additional suggestions, offered in the hope of further strengthening the story; as they say, writing is never done, only due (or submitted).

1. Regarding Fig 2A/B, the authors might consider mentioning in the Methods section (or figure legend) that both ligand donor and recipient CHO cells were analyzed (unlike for the flow-based experiments in Fig 1, in which the donors and recipients can be easily separated).
2. Also in Fig 2A, I assume that the decreased GFP expression observed by immunoblotting reflects the loss of the ligand on ligand-donor CHO cells. However, it is puzzling why a decrease in GFP signal was not observed in Fig 2B (CTLA-4 WT + CD86 lanes) at a comparable level to that observed in Fig 2A, especially as the data in 3E seem to imply that CD86 possesses comparable ligand-depleting capability as CD80 at the 6-hour time point.
3. In the Discussion, the authors comment that the LRBA pathway seems to be the default mechanism for CTLA-4 regulatory trafficking: the high-affinity interaction between CTLA-4:CD80 leads to ubiquitination of CTLA-4, causing ubiquitinated CTLA-4 to deviate from LRBA routes and eventually end up in the lysosome for degradation. In Fig 8, it may be helpful to include the LRBA pathway in panel A (to illustrate how the default pathway is bypassed) and the endosome/lysosome pathway in panel B (to illustrate how CTLA-4 recycling occurs before the early endosome matures into the late endosome) to strengthen the point.

A clear illustration of this temporospatial relationship could help readers process the data in Fig 2. For example, it helped me make sense of Fig 2E, where I would intuitively expect Bafilomycin A treatment to alter the ubiquitin pattern of CD86 to be like that of CD80, similar to Bafilomycin A's effect in Fig 1A/B by flow analysis. A better introduction to Fig 2 may be helpful.
4. Statistical analysis should be included in Fig S2. Similarly, statistical analysis of the data comparing DN and EV, and CA and EV, should also be included in Fig 4A, rather than simply comparing loss of function with gain of function.

Reviewer #4:

Remarks to the Author:

In this manuscript, Kennedy et al. investigate the different fates of CTLA-4 following engagement by CD80 or CD86. They propose that CTLA-4 is ubiquitinated and degraded following engagement with CD80 but not CD86. They additionally identify mutations in patients with autoimmunity affecting the binding of CTLA-4 to CD86 more than CD80. While the proposed mechanism is potentially intriguing, the physiological relevance remains rather unclear, and the main conclusions are not fully supported by the available data. The main concerns and important questions that need to be addressed are as follows:

1. While the authors show that CD80 engagement leads to CTLA-4 ubiquitination, they do not provide sufficient evidence that CD86 engagement does not. For example, figures 2C and S3A lack a crucial control with CD86. Further, the blots shown in figures 2A and B appear to have different exposure times or were loaded with different amounts of protein, and in figure 2B (CD86), the bands in lane 4 appear to show signs of ubiquitination (increased molecular weight and degradation bands) that could be interpreted to be similar to what is seen with CD80 in 2A. Therefore, it is critical to use same amounts of proteins and exposure times in these and other similar specific comparisons, as well as to include actin, tubulin or similar controls in all blots in Figure 2. Similar signs of ubiquitination are also seen in figure 2E. This all points to the possibility that engagement of CD86 by CTLA-4 also leads to its ubiquitination and degradation. The lack of co-precipitation of ubiquitin with GFP following CD86 engagement (figure 2E) does not exclude the possibility that CTLA-4 is being ubiquitinated following CD86 engagement. The authors could IP with CTLA-4 at different timepoints following CD80 or CD86 engagement and then directly stain for ubiquitin to formally assess this possibility.
2. In regard to the differential intracellular trafficking of CD80 and CD86, only about 30% of CD86+/CTLA-4+ vesicles are in early endosomes (LRBA and Rab5). The majority of such vesicles are in the late endosomal compartments, so the conclusion that “CD86- CTLA-4 interactions were more biased towards early endosomal compartments” is not entirely true. Further, the primary microscopy data (for LRBA) in figure S4 appears to show some areas of co-localization that are not quantified.
3. The experiment with CA or DN Rab11 (figure 4A) show similar trends for both CD80 and CD86, although significance is reached only for CD86. It is possible that this experiment is under-powered and that the trend would become significant with more replicates. The authors should include additional data points.
4. The pH-dependent dissociation of CD80/CD86 from CTLA-4 also does not appear to be CD86-specific. Instead, in the presence of NH₄Cl, the downregulation of both CD80 and CD86 are affected (figure 3E).
5. Staining of CTLA-4 after CD80 or CD86 engagement (as shown in figure S5A using Jurkat cells) should be performed with Tregs to confirm that this effect in recycling of CTLA-4 is also observed in the endogenous system. These results should be expanded and placed in the main figures as they are key to support the authors’ model.
6. If engagement of CD80 by CTLA-4 leads to its ubiquitination and degradation, why does the associated GFP not get degraded (figure 1A for example)? There are also major differences in the three cell types used for these assays. Further, figure 1A should show earlier time points to demonstrate the initial presence of CD86-GFP in the recipient cells prior to its degradation.
7. Finally, the results of the sole functional assay (figure 7) are over-interpreted. The authors state that the CTLA-4 R70Q suppresses CD80-driven T cell responses similarly to CTLA-4 WT, but there is increased proliferation, showing that the CTLA-4 R70Q mutation affects both CD80 and CD86, bringing the conclusions regarding the clinical significance of the differences of the binding of CTLA-4 to CD80 or CD86 into question. Further, the data shown in plots is not quantified to distinguish between the number of cell divisions and the number of cells that have divided.

Other concerns:

1. Flow data is shown as representative plots. The authors should include quantification in all figures in order to ensure the robustness of these experiments.
2. The quality of figure 2F is not good and should be replaced with a clearer picture.
3. In figure 4B, why do the donor cells not become negative for mCherry like in other similar experiments?

Reviewer #5:

Remarks to the Author:

The manuscript "Differences in CD80 and CD86 transendocytosis reveal CD86 as a key target for CTLA-4 immune regulation" is challenging to assess the difference between CD80-CTLA4 and CD86-CTLA4 interactions and proposes an impressive new model that CTLA-4 interacted with CD86 is recycled whereas CTLA-4 interacted with CD80 is degraded due to the high affinity. As authors describe, T cell regulation via CD80, CD86 and CTLA-4 is important but contains unclear open questions. The phenomenon of transendocytosis is assessed by multiple methods. The manuscript includes important data, however, the model is not well supported by the data.

Major comment

1. The lack of direct evidence for the model.

The model is clearly shown in Fig 8. It indicates two interesting phenomena.

1-1. CTLA-4 after binding with CD86 is reused for the down-regulation of CD86.

1-2. The interaction with low affinity may have paradoxically high activity.

However, I could not find out clear evidence that directly support these predictions.

As for 1-1, data in "Figure 3 E No NH₄Cl" indicates no difference or slightly less reduction of CD86 than CD80. If the Fig 8 model is correct, CD86 should decrease more remarkably, especially at the later time point, than CD80. This is the most critical data to assess the hypothesis, and the result denies the hypothesis.

Fig S5A is important Figure to show the amount of CTLA-4 after ligand binding. However, this difference can be explained by higher affinity of CD86.

As for 1-2, CTLA-4 mutants are analyzed in Figure 5-7. R70Q/W mutants has the decreased affinity to CD80 (Fig 5), and showed nice transendocytosis activity against CD80 (Fig 6). If the R70Q/W – CD80 interaction has good regulatory function as WT CTLA-4 – CD86, the 1-2 prediction is clearly demonstrated. However, Figure 7 shows low affinity mutant of CTLA-4 has lower regulatory function. They cause autoimmune disease in patients. I understand that there are alternative explanations, but anyway, the result does not support the model.

Minor comments

1. Generally speaking, it is difficult to compare two different molecules with a similar property.

I do not want to say such a small difference is not important, but the authors need to be more careful to demonstrate the difference than in the case to show the effect of a particular reagent. The difference between CD80 and CD86 in Figure 1, Figure 2, Figure 3A, C D can be explained by slightly higher expression or higher affinity of CD80 than CD86. Ideally, dose-dependent effect, which is lack in this manuscript, may strengthen the results more clearly. Instead, the authors show time-

dependency and different effects of reagents in Figure 3B E and Figure 4, which may be enough to show the difference of CD 80 and CD86.

2. Figure 1 CTLA4-ve is not explained.

Although the Fig S2 clearly explain which are Donor and Recipient, Fig 1 is a bit hard to understand. I have confused CTV-ve in the text and the legend and CTLA4-ve in the Figure 1. The text in line 32 in page 4 writes "CTLA-4 expressing (CTV-ve)" although CTLA4-ve, which means "no-expression of CTLA-4", but not CTV-ve is written in Figure 1A.

3. Figure2A indicates several findings that are ignored in the text.

In Figure 2A, I find (1) the amount of GFP decreases after 6 hours even by CTLA-4 Kless, (2) the amount of CTLA-4 at the main band (25kDa) does not decrease after CD80 interaction, (3) smaller fragment of CTLA-4 (15kDa) is generated after CD80 interaction (which disappears by adding Bafilomycin A in Fig 2E), whereas authors only describe the increased molecular weight of the CTLA-4 band. The results indicate that a part of CTLA-4 is degraded or cleaved after CD86 and CD80 interaction possibly via a ubiquitin-independent but Bafilomycin A related mechanism. Some description and discussion are required.

I found the authors are trying to address a process that is very important but hard to demonstrate. My sole critical concern is the comment written as major comment 1-1 (Figure 3E). This contradictory data may be explained by higher expression of CD80 than CD86, but if CD80 expression is too high to compare the effects, most data must be re-considered as I describe in minor comment 1.

Author Rebuttal, first revision:

Decision Letter, second revision:

11th Mar 2022

Dear David,

We have received back the comments on your revised manuscript entitled "Differences in CD80 and CD86 transendocytosis reveal CD86 as a key target for CTLA-4 immune regulation", which has now been re-reviewed by the 3 referees who had seen the previous version of the manuscript. Once again, we received mixed reports for this study. You will see from their comments below that while they find your work of interest, some important points are raised. We are interested in the possibility of publishing your study in Nature Immunology, but would like to ask you to address these concerns in the form of a revised manuscript before we make a final decision on publication.

We invite you to submit a substantially revised manuscript, however please bear in mind that we will be reluctant to approach the referees again in the absence of major revisions.

Specifically, the revision should include new experiments to address:

- (1) repeat the immunoblot experiments shown in Figure 2, (please note that in accordance with SN policies for publication standards, we routinely ask authors to supply full blots for all experiments). Referee #4 is still expressing concerns about these data (see their point 1 below) and the normalization of the gel loading by showing the tubulin blots for the gel lanes.
- (2) increase the numbers of independent assays for the experiments shown in Extended Data Figure 4a,
- (3) tone down or more accurately describe what is being shown in the figures (as requested by both referees #4 & #5).

As noted by referee #5, substantial re-writing of the title & abstract is also required.

- * Include a "Response to referees" document detailing, point-by-point, how you addressed each referee comment. If no action was taken to address a point, you must provide a compelling argument. This response will be sent back to the referees along with the revised manuscript.
- * If you have not done so already please begin to revise your manuscript so that it conforms to our Article format instructions at <http://www.nature.com/ni/authors/index.html>. Refer also to any guidelines provided in this letter.
- * Include a revised version of any required reporting checklist. It will be available to referees (and,

potentially, statisticians) to aid in their evaluation if the manuscript goes back for peer review. A revised checklist is essential for re-review of the paper.

The Reporting Summary can be found here:

[Redacted]

If you wish to submit a suitably revised manuscript we would hope to receive it within 6 months. If you cannot send it within this time, please let us know. We will be happy to consider your revision so long as nothing similar has been accepted for publication at Nature Immunology or published elsewhere.

Nature Immunology is committed to improving transparency in authorship. As part of our efforts in this direction, we are now requesting that all authors identified as 'corresponding author' on published papers create and link their Open Researcher and Contributor Identifier (ORCID) with their account on the Manuscript Tracking System (MTS), prior to acceptance. ORCID helps the scientific community achieve unambiguous attribution of all scholarly contributions. You can create and link your ORCID from the home page of the MTS by clicking on 'Modify my Springer Nature account'. For more information please visit www.springernature.com/orcid.

Thank you for the opportunity to review your work.

Sincerely,

Laurie

Laurie A. Dempsey, Ph.D.
Senior Editor
Nature Immunology
l.dempsey@us.nature.com
ORCID: 0000-0002-3304-796X

Reviewers' Comments:

Reviewer #2:

Remarks to the Author:

The authors have appropriately addressed my concerns. Again, I thank the authors for the discussion and intriguing findings.

Reviewer #4:

Remarks to the Author:

While this revised manuscript by Kennedy et al. has somewhat improved, our crucial concerns have not been appropriately addressed:

1. Figure 2: In Figure 2e, there are degradation bands seen for both CD80 and CD86 at 3h and 5h that are not seen without ligand and these bands are not seen after the addition of BafA. This suggests less significant differences between CD80 and CD86 functions regarding the degradation of CTLA-4 than those suggested by the authors. Most importantly, there is still no CD86 control (for 2c and 2d), as asked for in our previous review. The authors' suggestion that this control is not necessary because it is excluded by their proposed model is preposterous. Further, extended data 2a must also include this CD86 control as suggested before. Also, tubulin loading controls should also be included for ALL gels as requested previously.
2. Figure 3b: The differences between CD80 and CD86 for Rab5 and LRBA are not significant and should not be discussed as such. Further, the majority of the CTLA-4+/CD86+ vesicles are still within late endosomal compartments (Rab7 and LAMP3), so the conclusion that "Since, relative to CD80-CTLA-4, interactions CD86 and CTLA-4 were biased towards early endosomal compartments (Rab5+ and LRBA+) this potentially indicated that they separate as intracellular pH decreases" is still not accurate.
3. Extended data 4a: It is not clear that "CD80 and CD86 respond differently to these manipulations in terms of magnitude of effect." There are clearly similar trends for both CD80 and CD86, and given the slight differences in baseline, the magnitude of effect is not very different. Given the variation seen in the data points, the study is underpowered and should be expanded.

4. The phrase “CTLA-4-mediated regulation of CD86 is required to prevent autoimmunity” is not supported by the data included in this manuscript. While the mutations of CTLA-4 that are investigated here clearly disrupt binding to CD86 more, there is still an effect for CD80 that could contribute to the autoimmunity seen in patients with such mutations. There may also be effects on the other functions of CTLA-4 that are not investigated here. Therefore, this cannot be stated in the abstract or anywhere else in the text.

5. There is also an overall concern that the authors overstate the biological significance and impact of their findings in the absence of more definitive in vivo functional studies. The manuscript appears to include two separate concepts. One, focused on the impact of binding of CD80 or CD86 to CTLA-4 on the availability of CTLA-4 on cells such as Treg cells. Second, the role of a specific mutation in CTLA-4 on its ability to regulate CD80 and CD86. These two concepts are not effectively linked in this manuscript.

Minor points:

1. Extended Data 2b: Are the tubulin bands from the same gel? The GFP and CTLA-4 stains clearly slope down to the right, but the tubulin stain does not.

2. Figure 3a: There should be a control with no ligand in order to be determine if the amount of PLA signal observed with CD86 is truly “background.”

3. Lack of consistency for SD and SEM make it difficult to properly interpret some data. Additionally, the methods say that all graphs include SD, which does not seem to be the case. For example, Figure 4c appear to contain SEM (left) and SD (right). Further, the right graph for CD86 contains data points with either no error bars, error bars going in one direction, or both directions. Figure 1d (SD) and 1e (SEM) are another example. Also the methods section should include what types of post-hoc tests were done for one- and two-way ANOVAs.

4. The quality of Figures 2c and 2f are not good.

5. Figure legends should indicate whether excess ligand is used or not to help readers who may have similar questions regarding the downregulation of ligand.

Reviewer #5:

Remarks to the Author:

The revised manuscript “Differences in CD80 and CD86 transendocytosis reveal CD86 as a key target for CTLA-4 immune regulation” is assessing the difference between CD80-CTLA4 and CD86-CTLA4 interactions. The manuscript is substantially improved. Data are now presented clearly and reliably, and with appropriate statistics, except for the following two concerns. Authors have responded to majority of my concerns. I have two comments for minor revision.

1. Their title is “Differences in CD80 and CD86 transendocytosis reveal CD86 as a key target for CTLA-4 immune regulation”. Therefore, I thought that authors want to say that CD86 removal is “better” than CD80 because of recycling even though CD86 has lower affinity for CTLA-4 compared to CD80. According to the point-by-point response, it was a misconception. The title and sentences in lines 53-54 (Abstract), 123-124(Introduction) might be misleading. Further, in line 49 (Abstract), “In contrast,

in the presence of CD86, CTLA-4 detached ----" can be interpreted as the effect of CD86 becomes dominant over CD80 if they are co-expressed.

The data can be interpreted as follows. "CD86 has lower affinity for CTLA-4 compared to CD80. The lower affinity induces easier dissociation and reduces the degradation rate of CTLA-4. Consequently, the reduced affinity is almost fully compensated as for the function of B7. The quantitative difference of affinity explains the qualitative differences in transendocytosis." I prefer these explanations than a claim that "we find out a novel distinct function of CD86!". If CD80 and CD86 are different in transendocytosis, researchers must find out the cause. It is explained by the low affinity by the data using the interaction between CD80 and R70Q/W mutant. As a general tendency in published papers, new specific difference may be too much emphasized.

The presented data is very interesting, even if there is no distinct novel function in CD86. I will accept the title if editors agree with such a slightly exaggerated presentation.

2. The CTLA-4 degradation bands (15kDa in Figure2) in Western blotting should be described in the text. I think the results well support the notion that CD80 induces CTLA-4 degradation. Although there are some amounts of 15kDa degradation CTLA-4 bands in the presence of CD86 interaction, the result can be explained without confusing, because authors have added descriptions about background of CTLA-4 ubiquitylation in the revised manuscript. Minor modification may be required in the description that "CTLA-4 undergoes a constitutive low level of ubiquitylation and that this process is stimulated by ligation with CD80 but not CD86" in line 195.

Author Rebuttal, second revision:

Nature Immunology (NI-A32355B "Differences in CD80 and CD86 transendocytosis reveal CD86 as a key target for CTLA-4 immune regulation")

Response to referees

Reviewer #4:

Remarks to the Author:

While this revised manuscript by Kennedy et al. has somewhat improved, our crucial concerns have not been appropriately addressed:

1. Figure 2: In Figure 2e, there are degradation bands seen for both CD80 and CD86 at 3h and 5h that are not seen without ligand and these bands are not seen after the addition of BafA. This suggests less

significant differences between CD80 and CD86 functions regarding the degradation of CTLA-4 than those suggested by the authors.

In response to previous comments by the referee, we re-ran CD80 and CD86 transendocytosis experiments on the same gel and quantified the extent to which an increase CTLA-4 Mw (ubiquitylation) was seen (fig. 2a and b). This shows clear differences between the two ligands in terms of CTLA-4 modification. We did not focus on the degradation fragment of CTLA-4 that the reviewer refers to, because this is not specific for CTLA-4 degradation via ubiquitylation. Its presence or absence does not seem to be a reliable indicator of whether CTLA-4 was ubiquitylated or not. For example, it is observed to some extent in settings where there is no ligand, in settings where CD86 is present and even in experiments where we use the Kless mutant (Extended Data 2). This does not diminish our core observation, that CD80 increases the Mw of CTLA-4 via ubiquitylation and CD86 does not. Therefore, the presence of the degradation band cannot easily be used to infer whether CD80 and CD86 are having similar effects. We have now added a discussion on this point to the main text (lines 197-201).

Most importantly, there is still no CD86 control (for 2c and 2d), as asked for in our previous review. The authors' suggestion that this control is not necessary because it is excluded by their proposed model is preposterous. Further, extended data 2a must also include this CD86 control as suggested before. Also, tubulin loading controls should also be included for ALL gels as requested previously.

We have now replaced Fig.2c and 2d with new experiments to include the CD86 controls as requested. As previously indicated, using CD86 as bait (IP -GFP) does not precipitate significant amounts of CTLA-4 in contrast to CD80 precipitation (figure 2c). Moreover, ubiquitin smears are only associated with CD80 interacting with WT CTLA-4 and are not seen where the CTLA-4 tail contains lysine mutations (Kless) or is deleted ($\Delta 36$) or in any condition using CD86. Similarly, in figure 2d, when we IP ubiquitin we pull down much more CTLA-4 and associated CD80 than with CD86. In this experiment there is no a priori selection for CD80 or CTLA-4 indicating it is the CD80-CTLA-4 interaction that drives increased precipitation via ubiquitin. Increases in CTLA-4 ubiquitylation are seen in the presence of CD80 but not CD86 and are abrogated by removing lysine residues from the CTLA-4 tail (figure 2d).

We have also replaced ED 2a with a new experiment as requested and show that ubiquitin smearing of CTLA-4 is CD80 driven and inhibited by MG132. No such effect is seen for CD86. The "degradation band" is also reduced by MG132, but low levels remain, consistent with an additional, non-ubiquitin, pathway seen with CD80, CD86 and no ligand. We have included tubulin controls as requested.

2. Figure 3b: The differences between CD80 and CD86 for Rab5 and LRBA are not significant and should not be discussed as such. Further, the majority of the CTLA-4+/CD86+ vesicles are still within late endosomal compartments (Rab7 and LAMP3), so the conclusion that “Since, relative to CD80-CTLA-4, interactions CD86 and CTLA-4 were biased towards early endosomal compartments (Rab5+ and LRBA+) this potentially indicated that they separate as intracellular pH decreases” is still not accurate. *We have now re-phrased the text so as to avoid any potential overinterpretation.*

3. Extended data 4a: It is not clear that “CD80 and CD86 respond differently to these manipulations in terms of magnitude of effect.” There are clearly similar trends for both CD80 and CD86, and given the slight differences in baseline, the magnitude of effect is not very different. Given the variation seen in the data points, the study is underpowered and should be expanded. *We have replaced this figure with a new series of experiments using DN Rab11. These indicate that CD86 transendocytosis is more sensitive to DN Rab11 than CD80 transendocytosis (Extended Data Fig. 4a).*

4. The phrase “CTLA-4-mediated regulation of CD86 is required to prevent autoimmunity” is not supported by the data included in this manuscript. While the mutations of CTLA-4 that are investigated here clearly disrupt binding to CD86 more, there is still an effect for CD80 that could contribute to the autoimmunity seen in patients with such mutations. There may also be effects on the

other functions of CTLA-4 that are not investigated here. Therefore, this cannot be stated in the abstract or anywhere else in the text. *We agree the text is overstated and have re-phrased so as to avoid this issue. Nonetheless, since both CTLA-4 R70 and LRBA mutations independently lead to a defect of CD86 transendocytosis and are both associated with autoimmunity, the concept that CTLA-4 control of CD86 is more important is not entirely unreasonable.*

5. There is also an overall concern that the authors overstate the biological significance and impact of their findings in the absence of more definitive in vivo functional studies. The manuscript appears to include two separate concepts. One, focused on the impact of binding of CD80 or CD86 to CTLA-4 on the availability of CTLA-4 on cells such as Treg cells. Second, the role of a specific mutation in CTLA-4 on its ability to regulate CD80 and CD86. These two concepts are not effectively linked in this manuscript. *We have re-worked the manuscript in order to avoid overstating the data.*

Minor points:

1. Extended Data 2b: Are the tubulin bands from the same gel? The GFP and CTLA-4 stains clearly slope down to the right, but the tubulin stain does not. *In some early experiments, tubulin controls were run on duplicate gels that were loaded from the same sample, on the same day and processed alongside the CTLA-4 blot. Hence the difference in gel appearance here.*

2. Figure 3a: There should be a control with no ligand in order to be determine if the amount of PLA signal observed with CD86 is truly “background.” *We have performed additional experiments and now show the data with no ligand, CD80 and CD86 together as requested. This supports the idea that CD86 does not affect CTLA-4 ubiquitylation above background levels.*

3. Lack of consistency for SD and SEM make it difficult to properly interpret some data. Additionally, the methods say that all graphs include SD, which does not seem to be the case. For example, Figure 4c appear to contain SEM (left) and SD (right). Further, the right graph for CD86 contains data points with either no error bars, error bars going in one direction, or both directions. Figure 1d (SD) and 1e (SEM) are another example. Also, the methods section should include what types of post-hoc

tests were done for one- and two-way ANOVAs.. *This has now been corrected. The error bars were present on Figure 4c right graph but are quite tight, hence the appearance. We used Sidak's correction, and this has now been stated in the methods section.*

4. The quality of Figures 2c and 2f are not good. *We have now replaced 2c with a new experiment. We had previously replaced 2f with a new version as requested by the referee. The data clearly make the point that CD80 induces CTLA-4 ubiquitylation and CD86 does not. Despite the increased technical difficulty of performing biochemistry experiments with small numbers of primary Treg cells, Fig 2f provides evidence that aligns completely with the rest of the biochemistry shown. We have now provided the cleanest blot available in Fig.2f to make this point.*

5. Figure legends should indicate whether excess ligand is used or not to help readers who may have similar questions regarding the downregulation of ligand. *This has been done.*

Reviewer #5:

Remarks to the Author:

The revised manuscript “Differences in CD80 and CD86 transendocytosis reveal CD86 as a key target for CTLA-4 immune regulation” is assessing the difference between CD80-CTLA4 and CD86-CTLA4 interactions. The manuscript is substantially improved. Data are now presented clearly and reliably, and with appropriate statistics, except for the following two concerns. Authors have responded to majority of my concerns. I have two comments for minor revision.

1. Their title is “Differences in CD80 and CD86 transendocytosis reveal CD86 as a key target for CTLA-4 immune regulation”. Therefore, I thought that authors want to say that CD86 removal is "better" than CD80 because of recycling even though CD86 has lower affinity for CTLA-4 compared to CD80. According to the point-by-point response, it was a misconception. The title and sentences in lines 53-54 (Abstract), 123-124(Introduction) might be misleading. Further, in line 49 (Abstract), “In contrast, in the presence of CD86, CTLA-4 detached ----” can be interpreted as the effect of CD86 becomes dominant over CD80 if they are co-expressed.

The data can be interpreted as follows. “CD86 has lower affinity for CTLA-4 compared to CD80. The lower affinity induces easier dissociation and reduces the degradation rate of CTLA-4. Consequently, the reduced affinity is almost fully compensated as for the function of B7. The quantitative difference of affinity explains the qualitative differences in transendocytosis.” I prefer these explanations than a claim that “we find out a novel distinct function of CD86!”. If CD80 and CD86 are different in transendocytosis, researchers must find out the cause. It is explained by the low affinity by the data using the interaction between CD80 and R70Q/W mutant. As a general tendency in published papers, new specific difference may be too much emphasized. The presented data is very interesting, even if there is no distinct novel function in CD86. I will accept the title if editors agree with such a slightly exaggerated presentation.

This is a complex area of biology, and it is sometimes hard to generate simple take-home messages while avoiding any possible misunderstandings. However, with respect, at no point have we claimed to have found a "novel distinct function of CD86" or suggested that CD86 will be "dominant" when the ligands are co-expressed. We would argue CD80 is dominant over CD86 upon co-expression, however we do not touch on this issue in the manuscript. We do not believe such messages can be ascertained from the title or abstract. The sentence in the abstract (lines 53-54) "In contrast, in the presence of CD86, CTLA-4 detached in a pH-dependent manner and recycled back to the cell surface to permit further transendocytosis, allowing CD86 on opposing cells to be continuously targeted." seems reasonable to us and is supported by the data. Nonetheless, in response to the referee we have now modified the abstract and introduction to try and avoid any perceived issues.

To clarity, we re-state our position below:

There are two elements to the CD86 data. Firstly, during transendocytosis itself CD86 does not ubiquitylate CTLA-4, it detaches in a pH dependent manner and permits CTLA-4 recycling. Our data show that the recycling of CTLA-4 is important for CD86 transendocytosis to be effective. However, we make no suggestion that CD86 will be "better," merely that CD86 transendocytosis relies on the recycling process, to increase effectiveness. In contrast, CD80 modifies CTLA-4 via ubiquitin and directs it away from recycling. So, in terms of transendocytosis we show that CD80 and CD86 direct different CTLA-4 fates, hence "Differences in CD80 and CD86 transendocytosis..." in our title.

*The second part of the title relates to the importance of CD86 TE for immune regulation. This is based on our analysis of mutations found in humans with autoimmunity. It is known that patients with LRBA deficiency have problems with CTLA-4 homeostasis and suffer severe autoimmunity. Here we show using CRISPR generated LRBA knockout cells, that this selectively impacts CD86 transendocytosis. Since CD80 transendocytosis is relatively unaffected, we conclude that CD86 is a more important CTLA-4 target in control of autoimmunity. The concept that CD86 is a key target is then independently reinforced by a second set of mutations, this time in CTLA-4 itself. Here we show R70Q mutations ablate transendocytosis of CD86 but not CD80. Because there is an impact on CTLA-4 affinity, we cannot rule out effects on CD80 outside of transendocytosis, however this is not the case with the LRBA mutation. Therefore, together we think the data makes a reasonable case that CD86 is a more important target for CTLA-4. Hence we feel the title "**Differences in CD80 and CD86 transendocytosis reveal CD86 as a key target for CTLA-4 immune regulation**" is justified and does not exaggerate our findings.*

2. The CTLA-4 degradation bands (15kDa in Figure2) in Western blotting should be described in the text. I think the results well support the notion that CD80 induces CTLA-4 degradation. Although there are some amounts of 15kDa degradation CTLA-4 bands in the presence of CD86 interaction, the result can be explained without confusing, because authors have added descriptions about background of CTLA-4 ubiquitylation in the revised manuscript. Minor modification may be required in the description that "CTLA-4 undergoes a constitutive low level of ubiquitylation and that this process is stimulated by ligation with CD80 but not CD86" in line 195.

We have now modified the text to discuss this issue in more detail (lines 197-201).

Decision Letter, third revision:

Our ref: NI-A32355C

28th Jun 2022

Dear David,

Thank you for submitting your revised manuscript "Differences in CD80 and CD86 transendocytosis reveal CD86 as a key target for CTLA-4 immune regulation" (NI-A32355C). We have looked over the revised manuscript and determined that these final revisions have addressed the previous requests from the referees, therefore we'll be happy in principle to publish it in Nature Immunology, pending minor revisions to comply with our editorial and formatting guidelines.

We will now perform detailed checks on your paper and will send you a checklist detailing our editorial and formatting requirements in about a week. Please do not upload the final materials and make any revisions until you receive this additional information from us.

If you had not uploaded a Word file for the current version of the manuscript, we will need one before beginning the editing process; please email that to immunology@us.nature.com at your earliest convenience.

Thank you again for your interest in Nature Immunology Please do not hesitate to contact me if you have any questions.

Kind regards,

Laurie

Laurie A. Dempsey, Ph.D.
Senior Editor
Nature Immunology
l.dempsey@us.nature.com
ORCID: 0000-0002-3304-796X

Decision Letter, final checks:

Our ref: NI-A32355C

6th Jul 2022

Dear Dr. Sansom,

Thank you for your patience as we've prepared the guidelines for final submission of your Nature Immunology manuscript, "Differences in CD80 and CD86 transendocytosis reveal CD86 as a key target for CTLA-4 immune regulation" (NI-A32355C). Please carefully follow the step-by-step instructions provided in the attached file, and add a response in each row of the table to indicate the changes that you have made. Please also check and comment on any additional marked-up edits we have proposed within the text. Ensuring that each point is addressed will help to ensure that your revised manuscript can be swiftly handed over to our production team.

We would like to start working on your revised paper, with all of the requested files and forms, as soon as possible (preferably within one week, by Thursday, July 15th). Please get in contact with us if you anticipate delays.

When you upload your final materials, please include a point-by-point response to any remaining reviewer comments and please make sure to upload your checklist.

If you have not done so already, please alert us to any related manuscripts from your group that are under consideration or in press at other journals, or are being written up for submission to other journals (see: <https://www.nature.com/nature-portfolio/editorial-policies/plagiarism#policy-on-duplicate-publication> for details).

In recognition of the time and expertise our reviewers provide to Nature Immunology's editorial process, we would like to formally acknowledge their contribution to the external peer review of your manuscript entitled "Differences in CD80 and CD86 transendocytosis reveal CD86 as a key target for CTLA-4 immune regulation". For those reviewers who give their assent, we will be publishing their names alongside the published article.

Nature Immunology offers a Transparent Peer Review option for new original research manuscripts submitted after December 1st, 2019. As part of this initiative, we encourage our authors to support increased transparency into the peer review process by agreeing to have the reviewer comments, author rebuttal letters, and editorial decision letters published as a Supplementary item. When you submit your final files please clearly state in your cover letter whether or not you would like to participate in this initiative. Please note that failure to state your preference will result in delays in accepting your manuscript for publication.

Cover suggestions

As you prepare your final files we encourage you to consider whether you have any images or illustrations that may be appropriate for use on the cover of Nature Immunology.

If your image is selected, we may also use it on the journal website as a banner image, and may need

to make artistic alterations to fit our journal style.

Nature Immunology has now transitioned to a unified Rights Collection system which will allow our Author Services team to quickly and easily collect the rights and permissions required to publish your work. Approximately 10 days after your paper is formally accepted, you will receive an email in providing you with a link to complete the grant of rights. If your paper is eligible for Open Access, our Author Services team will also be in touch regarding any additional information that may be required to arrange payment for your article.

Please note that *Nature Immunology* is a Transformative Journal (TJ). Authors may publish their research with us through the traditional subscription access route or make their paper immediately open access through payment of an article-processing charge (APC). Authors will not be required to make a final decision about access to their article until it has been accepted. [Find out more about Transformative Journals](https://www.springernature.com/gp/open-research/transformative-journals).

If you have any questions about costs, Open Access requirements, or our legal forms, please contact ASJournals@springernature.com.

[Redacted]

Best regards,

Elle Morris
Senior Editorial Assistant
Nature Immunology
Phone: 212 726 9207
Fax: 212 696 9752

E-mail: immunology@us.nature.com

On behalf of

Laurie A. Dempsey, Ph.D.
Senior Editor
Nature Immunology
l.dempsey@us.nature.com
ORCID: 0000-0002-3304-796X

Final Decision Letter:

In reply please quote: NI-A32355D

Dear David,

I am delighted to accept your manuscript entitled "Differences in CD80 and CD86 transendocytosis reveal CD86 as a key target for CTLA-4 immune regulation" for publication in an upcoming issue of Nature Immunology.

Over the next few weeks, your paper will be copyedited to ensure that it conforms to Nature Immunology style. Once your paper is typeset, you will receive an email with a link to choose the appropriate publishing options for your paper and our Author Services team will be in touch regarding any additional information that may be required.

Please note that *Nature Immunology* is a Transformative Journal (TJ). Authors may publish their research with us through the traditional subscription access route or make their paper immediately open access through payment of an article-processing charge (APC). Authors will not be required to make a final decision about access to their article until it has been accepted. [Find out more about Transformative Journals](https://www.springernature.com/gp/open-research/transformative-journals).

Your paper will be published online soon after we receive your corrections and will appear in print in the next available issue. Content is published online weekly on Mondays and Thursdays, and the embargo is set at 16:00 London time (GMT)/11:00 am US Eastern time (EST) on the day of publication. Now is the time to inform your Public Relations or Press Office about your paper, as they might be interested in promoting its publication. This will allow them time to prepare an accurate and satisfactory press release. Include your manuscript tracking number (NI-A32355D) and the name of the journal, which they will need when they contact our office.

About one week before your paper is published online, we shall be distributing a press release to news organizations worldwide, which may very well include details of your work. We are happy for your institution or funding agency to prepare its own press release, but it must mention the embargo date and Nature Immunology. Our Press Office will contact you closer to the time of publication, but if you or your Press Office have any enquiries in the meantime, please contact press@nature.com.

Also, if you have any spectacular or outstanding figures or graphics associated with your manuscript - though not necessarily included with your submission - we'd be delighted to consider them as candidates for our cover. Simply send an electronic version (accompanied by a hard copy) to us with a possible cover caption enclosed.

If you have not already done so, we strongly recommend that you upload the step-by-step protocols used in this manuscript to the Protocol Exchange. Protocol Exchange is an open online resource that allows researchers to share their detailed experimental know-how. All uploaded protocols are made freely available, assigned DOIs for ease of citation and fully searchable through nature.com. Protocols can be linked to any publications in which they are used and will be linked to from your article. You can also establish a dedicated page to collect all your lab Protocols. By uploading your Protocols to Protocol Exchange, you are enabling researchers to more readily reproduce or adapt the methodology

you use, as well as increasing the visibility of your protocols and papers. Upload your Protocols at www.nature.com/protocolexchange/. Further information can be found at www.nature.com/protocolexchange/about .

Please note that we encourage the authors to self-archive their manuscript (the accepted version before copy editing) in their institutional repository, and in their funders' archives, six months after publication. Nature Portfolio recognizes the efforts of funding bodies to increase access of the research they fund, and strongly encourages authors to participate in such efforts. For information about our editorial policy, including license agreement and author copyright, please visit www.nature.com/ni/about/ed_policies/index.html

Sincerely,

Laurie

Laurie A. Dempsey, Ph.D.
Senior Editor
Nature Immunology
l.dempsey@us.nature.com
ORCID: 0000-0002-3304-796X

Click here if you would like to recommend Nature Immunology to your librarian
<http://www.nature.com/subscriptions/recommend.html#forms>